# TIER BALANCING: TOWARDS DYNAMIC FAIRNESS OVER UNDERLYING CAUSAL FACTORS

**Zeyu Tang**[1], **Yatong Chen**[2], **Yang Liu**[2], and **Kun Zhang**[1,3]

[1]Department of Philosophy, Carnegie Mellon University
[2]Computer Science and Engineering Department, University of California, Santa Cruz
[3]Machine Learning Department, Mohamed bin Zayed University of Artificial Intelligence
`zeyutang@cmu.edu, ychen592@ucsc.edu, yangliu@ucsc.edu, kunz1@cmu.edu`

## ABSTRACT

The pursuit of long-term fairness involves the interplay between decision-making and the underlying data generating process. In this paper, through causal modeling with a directed acyclic graph (DAG) on the decision-distribution interplay, we investigate the possibility of achieving long-term fairness from a dynamic perspective. We propose *Tier Balancing*, a technically more challenging but more natural notion to achieve in the context of long-term, dynamic fairness analysis. Different from previous fairness notions that are defined purely on observed variables, our notion goes one step further, capturing behind-the-scenes situation changes on the unobserved latent causal factors that directly carry out the influence from the current decision to the future data distribution. Under the specified dynamics, we prove that in general one cannot achieve the long-term fairness goal only through one-step interventions. Furthermore, in the effort of approaching long-term fairness, we consider the mission of "getting closer to" the long-term fairness goal and present possibility and impossibility results accordingly.

## 1 INTRODUCTION

The long-term fairness endeavor inevitably involves the interplay between decision policies and the underlying data generating process: when deriving a decision-making system, one usually makes use of data at hand; when we deploy such a system, the decision would impact how data will look in the future (Perdomo et al., 2020; Liu et al., 2021). To understand why and how a data distribution responds to decision-making strategies, the investigation has to resort to causal modeling. The pursuit of long-term fairness, in turn, should also consider the changes in the underlying causal factors.

Various fairness notions with different flavors have been proposed in the literature: associative fairness notions that capture the correlation or dependence between variables, e.g., *Demographic Parity* (Calders et al., 2009), *Equalized Odds* (Hardt et al., 2016); causal fairness notions that involve modeling causal relations between variables, e.g., *Counterfactual Fairness* (Kusner et al., 2017; Russell et al., 2017), *Path-Specific Counterfactual Fairness* (Chiappa, 2019; Wu et al., 2019), *Causal Multi-Level Fairness* (Mhasawade & Chunara, 2021). The previously proposed fairness notions are with respect to a snapshot of the static reality, and do not have a built-in capacity to model the distribution-decision interplay in the long-term fairness pursuit.

In the effort of enforcing fairness in the dynamic setting, researchers have approached the problem from different angles: they provide causal modeling for fairness notions (Creager et al., 2020), analyze the delayed impact or downstream effect on utilities (Liu et al., 2018; Heidari et al., 2019; Kannan et al., 2019; Nilforoshan et al., 2022), enforce fairness in sequential or online decision-making (Joseph et al., 2016; Liu et al., 2017; Hashimoto et al., 2018; Heidari & Krause, 2018; Bechavod et al., 2019), investigate the relation between the long-term population qualification and fair decisions (Zhang et al., 2020), take into consideration the user behavior/action when deriving a decision policy (Zhang et al., 2019; Ustun et al., 2019; Miller et al., 2020; von Kügelgen et al., 2022), provide fairness transferability guarantee across domains (Schumann et al., 2019; Singh et al., 2021), or derive robust fair predictors (Coston et al., 2019; Rezaei et al., 2021). The proposed dynamic fairness enforcing

procedures usually limit their scope of consideration to only observed variables, and the fairness audit is performed directly on the decision or statistics defined on observed data.

In order to have a built-in capacity to capture the influence from the current decision to future data distributions, and more importantly, to induce a fair future in the long run, in this paper, we propose *Tier Balancing*, a long-term fairness notion that characterizes the interplay between decision-making and data dynamics through a detailed causal modeling with a directed acyclic graph (DAG). For example, the latent socio-economic status (whose estimation can be the output of a FICO credit score model), although not directly measurable, plays an important role in credit applications. We are motivated by the goal of inducing a fair future by actually balancing the inherent socio-economic status, i.e., the "tier", of agents from different groups. We summarize our contributions as follows:

- We formulate *Tier Balancing*, a fairness notion from the dynamic and long-term perspective that characterizes the decision-distribution interplay with a detailed causal modeling over both observed variables and latent causal factors.

- Under the specified data dynamics, we prove that in general, one cannot directly achieve the long-term fairness goal only through a one-step intervention, i.e., static decision-making.

- We consider the possibility of getting closer to the long-term fairness goal through a sequence of algorithmic interventions, and present possibility and impossibility results derived from the one-step analysis of the decision-distribution interplay.

## 2   PROBLEM SETUP

In this section, we present the formulation of the problem of interest. We first demonstrate in Section 2.1 a detailed causal modeling of the interplay between decision-making and data dynamics. Then in Section 2.2, we formulate *Tier Balancing*, a long-term fairness notion that captures the decision-distribution interplay with the presented causal modeling.

### 2.1   CAUSAL MODELING OF DECISION-DISTRIBUTION INTERPLAY ON DAG

Let us denote the time step as $T$ with domain of value $\mathbb{N}^+$. At time step $T$, let us denote the protected feature as $A_T$ with domain of value $\mathcal{A} = \{0, 1\}$, additional feature(s) as $X_{T,i}$ with domain of value $\mathcal{X}_i$, the (unmeasured) underlying causal factor $H_T$ (we call it "tier") with domain of value $\mathcal{H} = (0, 1]$, the (unobserved) ground truth label $Y_T^{(\text{ori})}$ and the observed label $Y_T^{(\text{obs})}$, with domain of value $\mathcal{Y} = \{0, 1\}$, and the decision $D_T$ with domain of value $\mathcal{D} = \{0, 1\}$. Figure 1 shows the causal modeling of the interplay between decision-making and underlying data generating processes, which involves multiple dynamics (from $T = t$ to $T = t + 1$).[1]

**Underlying data dynamics (stationary components)**   Considering the fact that the underlying data dynamics are relatively stable with respect to the timescale of decision-making (e.g., the societal changes happen at a much larger time scale compared to a particular credit application decision), we assume that processes governing how $(Y_t^{(\text{ori})}, X_{t,i})$ are generated from $(H_t, A_t)$ for each individual in the population are stationary and do not change over different $T = t$. We also assume that the underlying data generating process that governs how $H_{t+1}$ is updated from $(H_t, Y_t^{(\text{ori})}, D_t)$ across time steps is stationary, and so are the process governing the observation of $Y_{t+1}^{(\text{obs})}$ given $(D_t, Y_{t+1}^{(\text{ori})})$ and the process governing the update of $A_{t+1}$ from $A_t$.

The tier $H_t$ fully captures the individual's key property that is directly relevant to the scenario of interest, and therefore is the cause of $Y_t^{(\text{ori})}$ and $X_{t,i}$'s instead of the other way around. For example, the improvement in the socio-economic status can be reflected through an increase in income, while manipulating one's income only by changing the recorded number does not affect the actual ability to repay the loan. The determination of causal direction aligns with causal modelings in previous literature (see, e.g., Zhang et al. 2020).

**Decision-making dynamics (non-stationary components)**   The institution (decision maker) assigns decision $D_t$ to each individual according to the observed features $(A_t, X_{t,i})$ and the outcome

---

[1]Due to the space limit, we provide additional discussions on decision-distribution interplay in Appendix B.1.

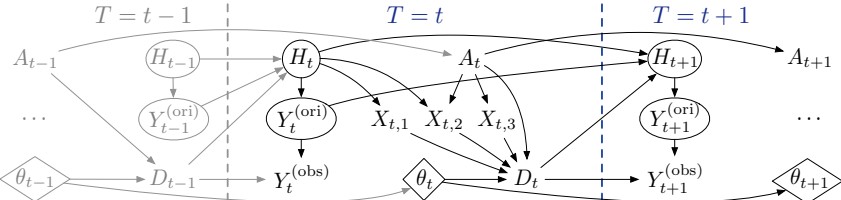

Figure 1: The causal modeling of the decision-distribution interplay. The circle (diamond) indicates that the corresponding variable (underlying factor) is unobservable.

record $Y_t^{(\text{obs})}$. The interpretation of the aforementioned variables depends on the problem at hand. For instance, in the credit application scenario where $D_t$ denotes the application decision (approval or denial), we can interpret the latent tier $H_t$ as the underlying socio-economic status of an individual. Since the decision-making strategy can vary across different time steps, we explicitly introduce the underlying factor $\theta_t$ (e.g., a hyperparameter or an auxiliary variable) to indicate such (possible) non-stationary property of decision-making. The causal path from $\theta_t$ to $\theta_{t+1}$ indicates the similarity between decision-making strategies as time goes by (e.g., the continuing interest on utility), although strategies themselves are not necessarily identical across different time steps.

We interpret the variable $Y_t^{(\text{ori})}$ as "whether or not one would repay the loan were he/she approved the credit at $T = t - 1$ (which might not be the case in reality)". The variable $Y_t^{(\text{obs})}$ is observed only if this individual actually got approved at $T = t - 1$, i.e., $D_{t-1} = 1$. We distinguish between the underlying ground truth $Y_t^{(\text{ori})}$ and the observed record $Y_t^{(\text{obs})}$ because of their different roles in the decision-distribution interplay. On the one hand, only $Y_t^{(\text{obs})}$ is observed and therefore accessible to the decision maker (e.g., for training and evaluation). On the other, since the potential outcome $Y_t^{(\text{ori})}$ reflects individual's inherent characteristic, which is not directly relevant to whether it is observable, $Y_t^{(\text{ori})}$ is utilized when the underlying data generating process specifies the update from $H_t$ to $H_{t+1}$.

## 2.2 THE NOTION OF TIER BALANCING

**Definition 2.1 ($K$-Step Tier Balancing).** Under the specified dynamics, starting from any time step $T$ and a given $K \geq 0$, let us denote a sequence of $K$ decision-making strategies as $D_{T:T+K-1} := \{D_T, ..., D_{T+K-1}\}$ (an empty set if $K = 0$), and the latest hidden tier after $K$-step decision-making as $H_{T+K}$. We say $D_{T:T+K-1}$ satisfies $K$-*Step Tier Balancing*, if at $T + K$ the following condition holds true (where "⊥⊥" denotes statistical independence):

$$H_{T+K} \perp\!\!\!\perp A_{T+K}, \text{ where } H_{T+K} \text{ is updated from } (H_T, Y_{T:T+K-1}^{(\text{ori})}, D_{T:T+K-1}). \tag{1}$$

Equation 1 captures the statistical consequence in the future (in the form of an associative relationship) induced by the interplay between the underlying data dynamics and decision-making policies along the way. The causal modeling is essential in capturing our long-term fairness goal, since the attainment of *Tier Balancing* is an induced outcome of a sequence of $K$ decision-making strategies $D_{T:T+K-1}$ (which are indispensable although the fairness notion itself is not explicitly defined on decisions).

Our *Tier Balancing* notion of algorithmic fairness is distinguished from previously proposed fairness notions in several important ways.[2] To begin with, *Tier Balancing* has a built-in dynamic flavor, whose definition involves variables that span across multiple time steps. Therefore the audit of *Tier Balancing* inevitably requires long-term and dynamic analysis, which is very different from previously proposed (both associative and causal) fairness notions defined with respect to a static snapshot of reality (e.g., Calders et al., 2009; Hardt et al., 2016; Kusner et al., 2017; Chiappa, 2019).

Besides, considering the fact that the decision-distribution interplay often involves situation changes in the hidden causal factors, *Tier Balancing* extends the scope of fairness consideration beyond only observed variables to hidden causal factors, which makes our notion a technically more challenging but more natural long-term fairness goal to achieve. The endeavor to explore the possibility of defining fairness in terms of latent causal factors is not an unrealistic fantasy. Recent advances in causal discovery literature have established identifiability results (under certain assumptions) on

---

[2]Due to the space limit, we provide detailed discussions on related works in Appendix A.

causal structures among latent variables (Xie et al., 2020; Adams et al., 2021; Kivva et al., 2021; Xie et al., 2022), which provide not only a theoretical justification, but also an indication of the potential, for our effort in exploring long-term fairness endeavor through modeling latent causal factors.

Furthermore, although *Tier Balancing* is not directly defined in terms of the decisions themselves, *Tier Balancing* is characterized with a detailed causal modeling that involves both decision-making and data dynamics. The explicit causal modeling of the decision-distribution interplay offers both challenges and opportunities for more principled fairness inquiries in the long-term, dynamic context (Hu & Chen, 2018; Liu et al., 2018; Mouzannar et al., 2019; Heidari et al., 2019; Zhang et al., 2020).

In Definition 2.1, we specify the step $K$ at which *Tier Balancing* is evaluated. If $K = 0$, i.e., $H_T \perp\!\!\!\perp A_T$ happens to be attained initially (although in general it may not be the case), *Tier Balancing* is attained at the beginning.[3] When $H_T \not\perp\!\!\!\perp A_T$ and $K \geq 1$, *K-Step Tier Balancing* is achieved with respect to the underlying causal factor $H_{T+K}$.

# 3 CHARACTERIZING TIER BALANCING

In Section 2, we propose a detailed causal modeling of the interplay between decision-making and underlying data generating processes, based on which we formulate a novel long-term fairness notion *Tier Balancing*. Our model is applicable to a wide range of resource allocation scenarios, e.g., hiring practice (Hu & Chen, 2018; Kannan et al., 2019), credit application (Liu et al., 2018), predictive policing (Ensign et al., 2018; Elzayn et al., 2019).

For the clarity of discussion, in this section we consider a running example of credit application where agents in a fixed population repeatedly apply for credit. We first demonstrate how one can apply the proposed causal modeling in the credit application scenario in Section 3.1. Then in Section 3.2, we characterize the *Single-step Tier Imbalance Reduction (STIR)* term for the purpose of conducting one-step analysis on the *Tier Balancing* notion of long-term fairness.

## 3.1 MODELING DETAIL OF DECISION-DISTRIBUTION INTERPLAY

As shown in Figure 1, the unmeasured latent causal factor $H_t$ (the hidden socio-economic status) is the actual root cause of the ground truth label $Y_t^{(\text{ori})}$ as well as the (possibly) observed label $Y_t^{(\text{obs})}$ (the repayment record). For any given tier $H_t = h_t$, let us assume that the unobserved ground truth $Y_t^{(\text{ori})}$ is sampled from a Bernoulli distribution with $h_t$ as the success probability, and that the observed repayment record $Y_t^{(\text{obs})}$ depends on both the ground truth label $Y_t^{(\text{ori})}$ and the previously received decision $D_{t-1}$:

$$Y_t^{(\text{obs})} \begin{cases} = Y_t^{(\text{ori})} & \text{if } D_{t-1} = 1, \\ \textit{is undefined} & \text{if } D_{t-1} = 0, \end{cases} \text{ where } Y_t^{(\text{ori})} \sim \text{Bernoulli}(h_t). \tag{2}$$

From Equation 2 we can see that $Y_t^{(\text{obs})}$ is a masked copy of $Y_t^{(\text{ori})}$ (masked by $D_{t-1}$), and we have the following proposition capturing the property of the marginal distribution of $Y_t^{(\text{obs})}$:

**Proposition 3.1.** *At time step $T = t$, for any $H_t = h_t \in (0, 1]$, under the specified dynamics, among the population where ground truth is actually observable, i.e., $Y_t^{(obs)}$ is not undefined, we have:*

$$Y_t^{(obs)} \sim \text{Bernoulli}(h_t).$$

Proposition 3.1 captures the fact that among the population where repayment record $Y_t^{(\text{obs})}$ is observed, the marginal distributions of $Y_t^{(\text{obs})}$ and $Y_t^{(\text{ori})}$ are actually identical. This property indicates that although one does not have access to the unobserved tier, i.e., the socio-economic status $H$, one can still use the observed $Y^{(\text{obs})}$ as a bridge to infer its behavior.

**Fact 3.2.** Let $\mathcal{A}$ be the domain of value for the protected feature $A_t$, and $\mathcal{E}$ be the domain of value for all other exogenous noise terms $E_t$. For each time step $T = t$, we can represent $D_t$, $Y_t^{(\text{ori})}$, and $H_t$ via functions $g_t^D$, $g_t^{Y^{(\text{ori})}}$, and $f_t$ respectively, where $g_t^D : \mathcal{A} \times \mathcal{E} \to \{0, 1\}$, $g_t^{Y^{(\text{ori})}} : \mathcal{A} \times \mathcal{E} \to \{0, 1\}$, and $f_t : \mathcal{A} \times \mathcal{E} \to (0, 1]$, i.e., $D_t = g_t^D(A_t, E_t)$, $Y_t^{(\text{ori})} = g_t^{Y^{(\text{ori})}}(A_t, E_t)$, and $H_t = f_t(A_t, E_t)$.

---

[3]In Appendix B.2, we analyze the scenario in which *Tier Balancing* is initially satisfied.

Notice that Fact 3.2 represents variables $D_t$, $Y_t^{(\text{ori})}$, and $H_t$ with functions of *all* root causes (including the protected feature $A_t$ and exogenous noise terms $E_t$) in the system without explicitly specifying the respective functional forms, which may depend on further assumptions on the joint distribution and the time step $T = t$.[4] Fact 3.2 is a direct result of representing causal relations with a functional causal model (FCM) (Spirtes et al., 1993; Pearl, 2009), and is for the purpose of notational convenience in later analysis.[5]

**Assumption 3.3** (**Multiplicative update of underlying tier**). Let $\alpha_D, \alpha_Y \in [0, \frac{1}{2})$ be the parameters that capture the influences from current decision $D_t$ and ground truth $Y_t^{(\text{ori})}$ to next step:

$$H_{t+1} = \min\left\{1, H_t \cdot \left[1 + \alpha_D(2D_t - 1) + \alpha_Y(2Y_t^{(\text{ori})} - 1)\right]\right\}. \tag{3}$$

Assumption 3.3 states that $H_{t+1}$ treats $H_t$ as a baseline, with increase or decrease in a multiplicative form based on agent's received decision and repayment information. We are inspired by the evolution theory where multiplicative updates have been a common modeling choice to capture updates in relevant statistics (Friedman & Sinervo, 2016; Dawkins & Davis, 2017). The explicit dependency on the update parameters, $\alpha_D$ and $\alpha_Y$, related to $D_t$ and $Y_t^{(\text{ori})}$ respectively, characterizes the two important aspects of our model: the update in individuals' tier potentially depends on the received decision $D_t$, as well as the ground truth $Y_t^{(\text{ori})}$ (even if unobserved).[6] The condition $\alpha_D, \alpha_Y \in [0, \frac{1}{2})$ makes sure that $H_{t+1} > 0$, and the $\min\{\cdot, 1\}$ operation makes sure that $H_{t+1}$ is upper-capped by 1.

In practical scenarios where agents repetitively apply for resource (e.g., in our running example of credit application) at each time step $T = t$, with the entire group remains unchanged, we have:

**Assumption 3.4.** The protected feature at time step $T = t + 1$ is an identical copy of that at $T = t$:

$$\forall T = t : \; A_{t+1} = A_t. \tag{4}$$

## 3.2 One-step Analysis Towards Tier Balancing

In Definition 2.1 we established the long-term fairness goal. Considering the dynamic property of this fairness notion, apart from defining "what exactly is fair in the long run", in order to bridge the cognitive gap we also need to clarify the meaning of "getting closer to the long-term fairness goal", i.e., achieving the long-term fairness goal through a sequence of algorithmic interventions. In this section, we present the one-step theoretical analysis framework and characterize the *Single-step Tier Imbalance Reduction (STIR)* term $\Delta_{\text{STIR}}|_t^{t+1}$ for the purpose of investigating *Tier Balancing*.

### 3.2.1 Single-step Tier Imbalance Reduction (STIR)

Recall that the long-term fairness objective is the independence between the protected feature $A_T$ and the hidden tier $H_T = f_T(A_T, E_T)$ (at a time step $T$). Equivalently, we would like $H_T$ to *not* be a function of $A_T$. We can view $f_T(0, E_T)$ and $f_T(1, E_T)$ as two dependent random variables, and quantify the amount of "getting closer to the long-term fairness goal" by comparing the absolute difference between $f_T(0, E_T)$ and $f_T(1, E_T)$ before (when $T = t$) and after (when $T = t + 1$) one-step update, and see if the gap decreases. Since the individual-level exogenous noise term $E_T$ is the input, this comparison of absolute differences is on the individual level. Therefore in order to quantitatively characterize the overall amount of "getting closer to the long-term fairness goal", we need to take into account different possible combinations of decision $D_t$ and outcome $Y_t^{(\text{ori})}$ (when $T = t$) for each individual, and aggregate the individual-level comparisons over the population.

Given combinations of $D_t$ and $Y_t^{(\text{ori})}$, when $E_t = \epsilon$, we denote the conditional joint probability density of $\left(f_t(0, E_t), f_t(1, E_t)\right)$ as $q_t\left(f_t(0, \epsilon), f_t(1, \epsilon) \mid d, d', y, y'\right) := q_t\left(f_t(0, \epsilon), f_t(1, \epsilon) \mid g_t^D(0, \epsilon) = d, g_t^D(1, \epsilon) = d', g_t^{Y^{(\text{ori})}}(0, \epsilon) = y, g_t^{Y^{(\text{ori})}}(1, \epsilon) = y'\right)$, and calculate the *Single-step Tier Imbalance Reduction (STIR)* term from $T = t$ to $T = t + 1$, denoted by $\Delta_{\text{STIR}}|_t^{t+1}$, as following:[7]

---

[4] We implicitly adopt the assumption that the protected feature itself is not caused by other variables.

[5] In Appendix B.3, we discuss the role of exogenous terms $E_t$ in Fact 3.2.

[6] In Assumption 3.3 we explicitly specify that we are considering $Y_t^{(\text{ori})}$ instead of $Y_t^{(\text{obs})}$. At $T = t$, every individual has a binary ground truth $Y_t^{(\text{ori})}$. However, it might not be the case that everyone has an observable $Y_t^{(\text{obs})}$ (since $Y_t^{(\text{obs})}$ is undefined for an individual if its $D_{t-1} = 0$).

$$\Delta_{\text{STIR}}|_t^{t+1} := \mathbb{E}\left[\,|f_{t+1}(0, E_{t+1}) - f_{t+1}(1, E_{t+1})|\,\right] - \mathbb{E}\left[\,|f_t(0, E_t) - f_t(1, E_t)|\,\right]$$

$$= \sum_{d,d',y,y' \in \{0,1\}} P_t(d, d', y, y') \cdot \int_{\epsilon \in \mathcal{E}} \int_{\xi \in \mathcal{E}} q_t\big(f_t(0, \epsilon), f_t(1, \epsilon) \mid d, d', y, y'\big) \tag{5}$$

$$\cdot \big(|\varphi_{t+1}(\xi)| - |\varphi_t(\epsilon)|\big) \cdot \mathbb{1}\{\varphi_{t+1}(\xi) = G_t(f_t, g_t^D, g_t^{Y^{(\text{ori})}}; d, d', y, y', \epsilon, \alpha_D, \alpha_Y)\} d\xi d\epsilon,$$

where $\varphi_t(\epsilon) := f_t(0, \epsilon) - f_t(1, \epsilon)$, $G_t$ is a function whose value *only* relies on the information available at time step $T = t$, and $P_t(d, d', y, y')$ is the joint distribution of $(d, d', y, y')$:

$$P_t(d, d', y', y') := P_t\big(g_t^D(0, E_t) = d, g_t^D(1, E_t) = d', g_t^{Y^{(\text{ori})}}(0, E_t) = y, g_t^{Y^{(\text{ori})}}(1, E_t) = y'\big).$$

We can then characterize "getting closer to long-term fairness goal" via the inequality $\Delta_{\text{STIR}}|_t^{t+1} < 0$.

### 3.2.2 SIMPLIFICATION ASSUMPTIONS

From Equation 5 we can see that the calculation of $\Delta_{\text{STIR}}|_t^{t+1}$ requires knowledge about the gap comparison for each individual $|\varphi_{t+1}(e_{t+1})| - |\varphi_t(e_t)|$, the conditional joint density $q_t\big(f_t(0, \epsilon), f_t(1, \epsilon) \mid d, d', y, y'\big)$, and the joint probability for different combinations of decision and ground truth before one-step dynamics $P_t(d, d', y, y')$. In order to quantitatively analyze the property of $\Delta_{\text{STIR}}|_t^{t+1}$, it is essential that we have access to all three aforementioned quantities.

To begin with, we need to know the instantiations of $\varphi_{t+1}(e_{t+1})$ given $\varphi_t(e_t)$ under the specified dynamics. Luckily, as we have illustrated in Table 4, Table 5, and Table 6 of Appendix, under the specified dynamics we can list all possible cases of the term $\varphi_{t+1}(e_{t+1})$ given $\varphi_t(e_t)$.

Besides, we need additional knowledge on the conditional joint density $q_t\big(f_t(0, \epsilon), f_t(1, \epsilon) \mid d, d', y, y'\big)$. For the purpose of better elaboration, we present two assumptions on the behavior of this conditional joint density – a qualitative assumption and a quantitative assumption:

**Assumption 3.5** (**Qualitative assumption**). For any time step $T = t$ and any exogenous term $E_t = \epsilon \in \mathcal{E}$, let us denote $y = g_t^{Y^{(\text{ori})}}(0, \epsilon)$ and $y' = g_t^{Y^{(\text{ori})}}(1, \epsilon)$. The following inequalities hold:

$$P_t\big(f_t(0, \epsilon) > f_t(1, \epsilon) \mid y > y'\big) > P_t\big(f_t(0, \epsilon) < f_t(1, \epsilon) \mid y > y'\big),$$
$$P_t\big(f_t(0, \epsilon) < f_t(1, \epsilon) \mid y < y'\big) > P_t\big(f_t(0, \epsilon) > f_t(1, \epsilon) \mid y < y'\big). \tag{6}$$

**Assumption 3.6** (**Quantitative assumption**). On top of Assumption 3.5, let us further assume that the conditional joint density $q_t\big(f_t(0, \epsilon), f_t(1, \epsilon) \mid d, d', y, y'\big)$ satisfies the following condition:

$$q_t\big(f_t(0, \epsilon), f_t(1, \epsilon) \mid d, d', y, y'\big) = \begin{cases} \gamma_{dd'yy'}^{(\text{up})} & \text{if } f_t(0, \epsilon) \leq f_t(1, \epsilon) \\ \gamma_{dd'yy'}^{(\text{low})} & \text{if } f_t(0, \epsilon) > f_t(1, \epsilon) \end{cases}, \tag{7}$$

where $\gamma_{dd'yy'}^{(\text{low})} + \gamma_{dd'yy'}^{(\text{up})} = 2$, $\gamma_{dd'yy'}^{(\text{low})} < \gamma_{dd'yy'}^{(\text{up})}$ when $y < y'$, and $\gamma_{dd'yy'}^{(\text{low})} > \gamma_{dd'yy'}^{(\text{up})}$ when $y > y'$.

Assumption 3.5 is rather mild, stating that for any given exogenous noise term (of an individual) $E_t = e_t$, whenever the ground truth $Y_t^{(\text{ori})}$ favors certain demographic group, it is more likely that the underlying tier also favors the same group. Assumption 3.6 is just a special case of Assumption 3.5, with quantitative characteristics built-in for technical purposes.[8]

Lastly, we need to know the (behavior of) joint probability density $P_t(d, d', y, y')$ for all combinations of $(d, d', y, y')$. In fact, as we shall see in Section 4, when taking into consideration certain characteristics of the predictor, the joint probability $P_t(d, d', y, y')$ would follow some patterns that can simplify the analysis.

## 4 ONE-STEP ANALYSIS TOWARDS THE LONG-TERM FAIRNESS GOAL

In this section, we consider the possibility of attaining the long-term fairness goal with (a sequence of) one-step interventions. We first present a negative result that in general one cannot hope to achieve the long-term fairness goal through a single one-step intervention, i.e., static decision-making. In light of this result, we further investigate the possibility of getting closer to the long-term fairness goal with a sequence of one-step interventions, and present possibility and impossibility results accordingly.

---

[7] The details of the derivation can be found in Appendix B.4.

[8] In Appendix B.5, we present illustrative figures to demonstrate the connection between qualitative and quantitative assumptions.

### 4.1 THE GENERAL IMPOSSIBILITY OF ACHIEVING 1-STEP TIER BALANCING

In the following theorem, we prove that it is in general impossible to achieve $H_{t+1} \perp\!\!\!\perp A_{t+1}$ when initially $H_t \not\perp\!\!\!\perp A_t$, i.e., in general we cannot achieve *Tier Balancing* with a single one-step intervention.

**Theorem 4.1.** *Let us consider the general situation where both $D_t$ and $Y_t^{(ori)}$ are dependent with $A_t$, i.e., $D_t \not\perp\!\!\!\perp A_t, Y_t^{(ori)} \not\perp\!\!\!\perp A_t$. Then under Fact 3.2, Assumption 3.3, and Assumption 3.4, as well as the specified dynamics, when $H_t \not\perp\!\!\!\perp A_t$, only if at least one of the following conditions holds true for all $e_t \in \mathcal{E}$ can we possibly attain $H_{t+1} \perp\!\!\!\perp A_{t+1}$:*

*(1) The ratio $\frac{f_t(0,e_t)}{f_t(1,e_t)}$ has a specific domain of value $\frac{f_t(0,e_t)}{f_t(1,e_t)} = \frac{1 \pm \alpha_D \pm \alpha_Y}{1 \pm \alpha_D \pm \alpha_Y}$;*

*(2) Positive (negative) label is exclusive to the advantaged (disadvantaged) group, and everyone receives a positive decision (if $\alpha_D > \alpha_Y$);*

*(3) Negative (positive) labels only appear in the advantaged (disadvantaged) group, and everyone receives a positive decision (if $\alpha_D > \alpha_Y$);*

*(4) Everyone has a positive label, but the positive decision is exclusive to the advantaged group (if $\alpha_D < \alpha_Y$);*

*(5) Everyone has a positive label, but the positive decision is exclusive to the disadvantaged group (if $\alpha_D < \alpha_Y$).*

Theorem 4.1 lists all possible ways to directly achieve the long-term fairness goal (under the specified condition). As we can see that all of these conditions are rather restrictive: Condition (1) imposes strong conditions on the functional form of $f(\cdot)$. In particular, when $f_t(0, e_t), f_t(1, e_t)$ are both continuous random variables with non-zero density everywhere on the support $(0, 1]$, the ratio is still a continuous random variable (because the density is simply an integral over positive multiplications). We can see that the event specified in Condition (1) is a zero-measure one. Conditions (2-5) all require trivial decision-making policies. Therefore, in general one cannot directly achieve the long-term fairness goal. We need to consider the possibility of approaching the goal step-by-step.

### 4.2 POSSIBILITY OF GETTING CLOSER TO TIER BALANCING VIA ONE-STEP INTERVENTIONS

In Section 4.1, we have seen that under the specified dynamics, single one-step intervention is in general not enough in order to achieve long-term fairness. In this section, we investigate if certain strategy can get closer to the long-term fairness goal through a sequence of algorithmic interventions. If we follow the same principle to derive the decision-making policy from the data at each time step, one-step analysis suffices for the purpose of studying the interplay between decision and distribution.

#### 4.2.1 ONE-STEP ANALYSIS ON PERFECT PREDICTOR

In this section, we consider the perfect predictor, where the predicted output equals to the underlying ground truth, i.e., $D_t = Y_t^{(ori)}$. The output of the perfect predictor $D_t$ is fully specified by the value of ground truth $Y_t^{(ori)}$, and therefore is conditionally independent from the protected feature $A_t$ given $Y_t^{(ori)}$. Based on the definition of *Equalized Odds* (Hardt et al., 2016), the prefect predictor is also the best possible Equalized Odds predictor (at time step $t$) with an accuracy of $100\%$.

Furthermore, the joint probability $P_t(d, d', y, y')$ for a perfect predictor is not always positive for any possible combination of $(d, d', y, y')$. We have the following sufficient condition for this joint probability to be zero:

$$d \neq y, \text{ or } d' \neq y' \implies P_t(d, d', y, y') = 0. \tag{8}$$

With the help of this additional knowledge on the joint probability $P_t(d, d', y, y')$, we can quantitatively analyze the *Single-step Tier Imbalance Reduction (STIR)* term $\Delta_{\text{STIR}}|_t^{t+1}$ and present the following impossibility result for the perfect predictor:

**Theorem 4.2.** *Let us consider the general situation where both $D_t$ and $Y_t^{(ori)}$ are dependent with $A_t$, i.e., $D_t \not\perp\!\!\!\perp A_t, Y_t^{(ori)} \not\perp\!\!\!\perp A_t$. Under Fact 3.2, Assumption 3.3, and Assumption 3.4, and Assumption 3.6, as well as the specified dynamics, when $H_t \not\perp\!\!\!\perp A_t$, the perfect predictor does not have the potential to get closer to the long-term fairness goal after one-step intervention, i.e.,*

$$D_t = Y_t^{(ori)} \implies \Delta_{\text{STIR}}^{(Perfect\ Predictor)}|_t^{t+1} > 0. \tag{9}$$

Compared to Theorem 4.1 that one in general cannot directly attain the long-term fairness goal (balanced tier) through a one-step intervention, Theorem 4.2 provides further insights regarding

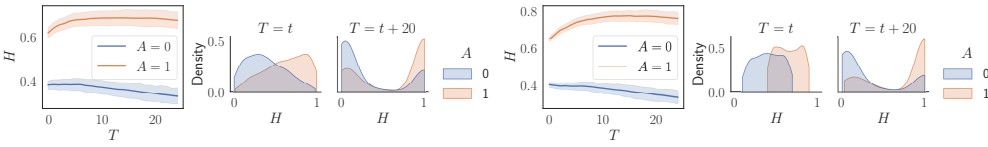

(a) the latent causal factor $H$ is sampled from truncated-Gaussian distributions

(b) the latent causal factor $H$ is sampled from Uniform distributions

Figure 2: Illustration of the interplay between decision with perfect predictors and data dynamics (20 steps) on simulated data, with different initialization of tier $H_t$.

what mission is possible through repetitive one-step interventions. In particular, under relatively mild assumptions, Theorem 4.2 establishes the impossibility of even getting closer to the long-term fairness goal through one-step interventions with the perfect predictor.

### 4.2.2 ONE-STEP ANALYSIS ON COUNTERFACTUAL FAIR PREDICTOR

In this section, we consider the *Counterfactual Fair* (Kusner et al., 2017) predictor. Similar to the one-step analysis on perfect predictors, we need to make use of the characteristic of Counterfactual Fair predictors to simplify the quantitative analysis on the term $\Delta_{\text{STIR}}|_t^{t+1}$.

The definition of Counterfactual Fairness requires the predictor to satisfy $g_t^D(0, E_t) = g_t^D(1, E_t)$ within each time step $T = t$. Therefore, we have the following sufficient condition for the joint probability $P_t(d, d', y, y')$ to be zero:

$$d \neq d' \implies P_t(d, d', y, y') = 0. \tag{10}$$

**Theorem 4.3.** *Let us consider the general situation where both $D_t$ and $Y_t^{(ori)}$ are dependent with $A_t$, i.e., $D_t \not\perp\!\!\!\perp A_t, Y_t^{(ori)} \not\perp\!\!\!\perp A_t$. Let us further assume that the data dynamics satisfies $\alpha_D \in (0, \frac{1}{2}), \alpha_Y = 0$. Then under Fact 3.2, Assumption 3.3, Assumption 3.4, and Assumption 3.6, as well as the specified dynamics, when $H_t \not\perp\!\!\!\perp A_t$, it is possible for the Counterfactual Fair predictor to get closer to the long-term fairness goal after one-step intervention, if certain properties of the data dynamics and the predictor behavior are satisfied simultaneously, i.e.,*

$$\begin{cases} g_t^D(0, E_t) = g_t^D(1, E_t), \frac{P_t(1,1,0,1)+P_t(1,1,1,0)}{P_t(0,0,0,1)+P_t(0,0,1,0)} < \frac{27}{8} \\ \alpha_D \in \left( \left( \frac{P_t(1,1,0,1)+P_t(1,1,1,0)}{P_t(0,0,0,1)+P_t(0,0,1,0)} \right)^{\frac{1}{3}} - 1, \frac{1}{2} \right), \alpha_Y = 0 \end{cases} \implies \Delta_{\text{STIR}}^{(Counterfactual\ Fair)}\Big|_t^{t+1} < 0. \tag{11}$$

Theorem 4.3 demonstrates the possibility (not guarantee) of getting closer to the long-term fairness goal through one-step interventions (under certain conditions) with Counterfactual Fair predictors. Compared to the general impossibility results for perfect predictors (Theorem 4.2), there are additional requirements (on both data dynamics and $P_t(d, d', y, y')$) accompanying the possibility result for Counterfactual Fair predictors. That being said, Theorem 4.3 clearly illustrates that the understanding of data dynamics through a detailed causal modeling, combined with a suitable decision-making strategy, can provide us with a promising way to approach the long-term dynamic fairness goal (*K-Step Tier Balancing* with $K > 1$), step by step.

## 5 EXPERIMENTS

In this section, we present experimental results on both simulated and real-world FICO data set from Board of Governors of the Federal Reserve System (US) (2007).[9] In the sequence of decision-distribution interplay, the latent causal factor $H_T$ is updated according to the specified dynamics (Equation 3) at each time step. The output of the decision policy (at each time step) depends on the specific scenario. In particular, we consider perfect predictors and Counterfactual Fair (Kusner et al., 2017, Level 1 implementation) predictors.

### 5.1 DECISION WITH PERFECT PREDICTORS ON SIMULATED DATA

In Figure 2 we present the 20-step interplay between decision and the underlying data generating process on the simulated data. The distributions of $H_t$ for different groups are initialized with

---

[9]Our code repository is available on Github: https://github.com/zeyutang/TierBalancing.

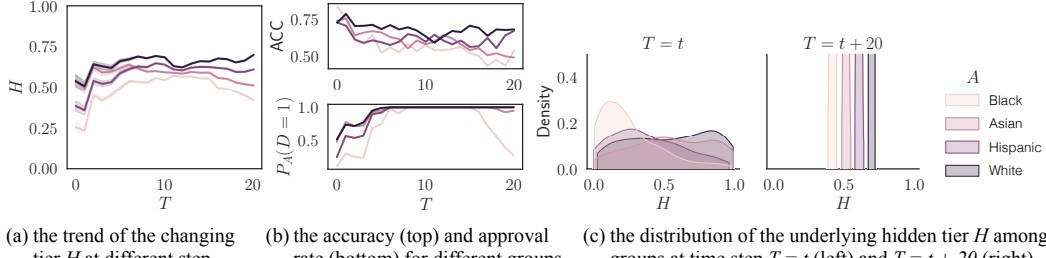

(a) the trend of the changing tier $H$ at different step

(b) the accuracy (top) and approval rate (bottom) for different groups

(c) the distribution of the underlying hidden tier $H$ among groups at time step $T = t$ (left) and $T = t + 20$ (right)

Figure 3: Illustration of the interplay between decision with Counterfactual Fair predictors and the data dynamics (20 steps) on the credit score data set. Panel (a) and (b) present the step-by-step tracks of update in tier, accuracy, and approval rates for different groups; panel (c) presents group-conditioned distributions of tier before (left) and after (right) 20 steps of interventions. The legend is shared across panel (a), (b), and (c).

truncated Gaussian distributions and Uniform distributions, respectively. During each time step $T = t$ we generate ground truth labels $Y_t^{(\text{ori})}$ according to data dynamics specified in Section 3.1 and set the decision $D_t$ to be equal to the ground truth $Y_t^{(\text{ori})}$ (perfect predictor as the decision-making policy); then the pair of $(Y_t^{(\text{ori})}, D_t)$ are utilized by the data dynamics to determine the tier $H_{t+1}$ for next step. As we can see from the left-hand-side figures in panels (a) and (b), the gap between tier for different groups is enlarged as the time goes by. This indicates that interventions through decision with perfect predictors did not get closer to the long-term fairness goal.

## 5.2 DECISION WITH COUNTERFACTUAL FAIR PREDICTORS ON CREDIT SCORE DATA

The FICO credit score data set contains 301,536 records of TransUnion credit score from 2003 (Board of Governors of the Federal Reserve System (US), 2007). In the preprocessed credit score data set (Hardt et al., 2016), we convert the cumulative distribution function (CDF) of TransRisk score among different groups into group-wise density distributions of the credit score, and use them as the initial tier distributions for different groups.

In Figure 3 we present the summary of a 20-step interplay between decision with Counterfactual Fair predictors and the underlying data generating process on the credit score data set. The Counterfactual Fair decision-making strategy is retrained after each one-step data dynamics. From Figure 3(a) we can observe that the gap between step-by-step tracks of tiers for different groups actually decreases before increasing again (around step 12). This indicates that decision with Counterfactual Fair predictors does have the potential to get closer to the long-term fairness goal, if the data dynamics and the initial condition (for each one-step analysis at different time step) satisfy certain properties. Figure 3(a) also highlight the importance of our dynamic fairness analyzing framework: if one does not model the interplay between decision and data dynamics, one may well deviate from long-term fairness goal even after making some progress by getting closer to the fairness objective.

## 6 CONCLUSION AND FUTURE WORK

In this paper, we propose *Tier Balancing*, a dynamic fairness notion that characterizes the decision-distribution interplay through a detailed causal model over both observed variables and underlying causal factors. We characterize *Tier Balancing* in terms of a one-step analysis framework on *Single-step Tier Imbalance Reduction (STIR)*. We show that in general one cannot directly achieve the long-term fairness goal only through a one-step intervention, i.e., static decision-making. We further show that under certain conditions it is possible (but not guaranteed) for one to get closer to the long-term fairness goal with (a sequence of) Counterfactual Fair decisions.

Our results highlight the challenges and opportunities of enforcing a fairness notion that has built-in capacity to model decision-distribution interplay over underlying causal factors. Future works naturally include developing algorithms to effectively and efficiently enforce the *Tier Balancing* notion of long-term, dynamic fairness for various practical scenarios.

ETHICS STATEMENT

The motivation of our work is to pursue long-term fairness. The conduct of research is under full awareness of, and with adherence to, ICLR Code of Ethics. We focus on the decision-distribution interplay and present a detailed causal modeling on both observed variables and latent causal factors. We present challenges and opportunities offered by this new type of fairness endeavor, and hope our work can inspire further research to promote fairness in the long run.

ACKNOWLEDGEMENT

This project was partially supported by the National Institutes of Health (NIH) under Contract R01HL159805, by the NSF-Convergence Accelerator Track-D award #2134901, by a grant from Apple Inc., a grant from KDDI Research Inc., and generous gifts from Salesforce Inc., Microsoft Research, and Amazon Research. YC and YL are partially supported by the National Science Foundation (NSF) under grants IIS-2143895 and IIS-2040800, and CCF-2023495.

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

# SUPPLEMENT TO "TIER BALANCING: TOWARDS DYNAMIC FAIRNESS OVER UNDERLYING CAUSAL FACTORS"

**Zeyu Tang**[1], **Yatong Chen**[2], **Yang Liu**[2], and **Kun Zhang**[1,3]

[1]Department of Philosophy, Carnegie Mellon University
[2]Computer Science and Engineering Department, University of California, Santa Cruz
[3]Machine Learning Department, Mohamed bin Zayed University of Artificial Intelligence
`zeyutang@cmu.edu, ychen592@ucsc.edu, yangliu@ucsc.edu, kunz1@cmu.edu`

## TABLE OF CONTENTS: APPENDIX

## LIST OF TABLES

# A    DETAILED DISCUSSIONS ON RELATED WORKS

In this section, we provide detailed discussions on related works. In particular, considering our focus on providing a novel long-term fairness notion with the help of the detailed causal modeling of involved dynamics, we compare our work with previous literature on causal notions of fairness, as well as fairness inquiries in dynamic settings.

## A.1    CAUSAL NOTIONS OF FAIRNESS

Various causal notions of algorithmic fairness have been proposed in the literature, for instance, fairness notions defined in terms of the (non-)existence of certain causal paths in the graph (Kamiran et al., 2013; Kilbertus et al., 2017; Zhang et al., 2017), fairness notions defined through estimating or bounding causal effects (Kusner et al., 2017; Chiappa, 2019; Wu et al., 2019; Mhasawade & Chunara, 2021), fairness notions defined with respect to statistics on certain factual/counterfactual groups (Imai & Jiang, 2020; Coston et al., 2020; Mishler et al., 2021). The proposed causal notions audit fairness in an instantaneous manner, i.e., the fairness inquires are with respect to a snapshot of reality, and the scope of consideration is limited to observed variables only. Our *Tier Balancing* notion has a built-in capacity to inquire fairness in the long-term and dynamic setting, which is very different from instantaneous causal fairness notions (beyond the fact that our notion encompasses latent causal factors).

While we can detect and measure discrimination based on previous (instantaneous) causal notions of fairness (e.g., the existence of certain causal paths or causal effects), eliminating such existence of causal paths or causal effects is a valid goal to achieve but might not be the means one should opt for. To begin with, there is no guarantee that eliminating a causal path or effect results in non-existence of such causal path or effect in the future under the interplay between decision-making and data dynamics. Furthermore, the data generating processes represented in the causal model might not be easily manipulable under the same timescale of decision-making (e.g., the ones governed by nature and/or the mode and structure of a society). One cannot expect that the manipulation on the causal model (for the purpose of enforcing fairness notions) directly translate to real-world changes in the underlying data generating processes.

Different from previous causal fairness notions, instead of directly "going against" the underlying data generating process (e.g., by eliminating certain causal path or causal effect), our *Tier Balancing* notion encourages "working with" the underlying data generating processes. With a detailed causal modeling of the decision-distribution interplay, *Tier Balancing* emphases on the possibility of inducing a future data distribution that is fair in the long run.

## A.2    FAIRNESS INQUIRES IN DYNAMIC SETTINGS

Previous literature have considered dynamic fairness in specific practical scenarios, for instance, opportunity allocation in labor market (Hu & Chen, 2018), a pipeline consisting of college admission followed by hiring (Kannan et al., 2019), opportunity allocation in credit application (Liu et al., 2018), and resource allocation in predictive policing (Ensign et al., 2018). Different from previous literature, we present a detailed causal modeling of the decision-distribution interplay that is general enough to be applicable in various resource allocation problems (e.g., loan applications, hiring practices) while also being specific enough to encompass nuances in data dynamics for the particular practical scenario of interest.

In terms of the analyzing framework, closely related works have considered the one-step analysis (Liu et al., 2018; Kannan et al., 2019; Mouzannar et al., 2019; Zhang et al., 2019). However, previous works focus on the long-term effect of imposing certain fairness notions that are readily available, for example, *Demographic Parity* (Calders et al., 2009; Liu et al., 2018; Mouzannar et al., 2019) and *Equal Opportunity* (Hardt et al., 2016; Liu et al., 2018). In our work, we formulate a novel notion of long-term fairness, namely, *Tier Balancing*, and explore the possibility of providing a fairness notion that characterizes the dynamic nature of decision-distribution interplay through detailed causal modeling on both observed variables and latent causal factors.

In terms of the modeling choice for data dynamics, most closely related works model data dynamics using variants of Markov Decision Processes (MDPs) (Jabbari et al., 2017; Siddique et al., 2020;

Zhang et al., 2020; D'Amour et al., 2020; Wen et al., 2021; Zimmer et al., 2021; Ge et al., 2021). For example, Zhang et al. (2020) consider the partially observed Markov decision process (POMDP) model, and conduct evolution and equilibrium analysis with respect to *Demographic Parity* and *Equal Opportunity* notions of fairness. The dynamics are modeled through transition matrices on group-level qualification rates. Compared to the modeling of transition matrices in MDPs, our model is more fine-grained and on the individual level, answering the call for "richer and more complex modelings [of involved dynamics]" in previous literature (Hu & Chen, 2018).

Another closely related work is the one-step analysis on the impact of causal fairness notions on downstream utilities conducted by Nilforoshan et al. (2022). They consider a detailed causal modeling on the college admission running example and analyze previously proposed (instantaneous) causal fairness notions, namely, *Counterfactual Predictive Parity* (Coston et al., 2020), *Counterfactual Equalized Odds* (Coston et al., 2020), and *Conditional Principal Fairness* (Imai & Jiang, 2020). Our work is different in several ways: instead of utilizing a static graph, we focus on the decision-distribution interplay and explicitly capture both observed and latent variables along the temporal axis; different from analyzing one-step downstream consequence in terms of utility, we formulate a long-term fairness goal and investigate the challenges and opportunities revealed by the notion.

# B    ADDITIONAL RESULTS, TECHNICAL DETAILS, AND DISCUSSIONS

In this section, we provide additional results, technical details, and discussions of our work. In Appendix B.1, we provide additional discussions on our causal modeling of the decision-distribution interplay; in Appendix B.2, we analyze the situation where *Tier Balancing* is initially attained; in Appendix B.3, we discuss the role of exogenous terms and provide a remark on Fact 3.2; in Appendix B.4, we present the detailed derivation of *Single-step Tier Imbalance Reduction* (STIR) term $\Delta_{\mathrm{STIR}}|_t^{t+1}$; in Appendix B.5, we illustrate the connection between Assumption 3.5 and Assumption 3.6; in Appendix B.6, we present additional experimental results; in Appendix B.7, we discuss potential limitations of our work.

## B.1    DISCUSSIONS ON THE CAUSAL MODELING OF DECISION-DISTRIBUTION INTERPLAY

In Appendix B.1.1, we provide additional details of the involved dynamics in the causal modeling of the decision-distribution interplay. In Appendix B.1.2, we discuss the relation between the practical scenarios and the modeled dynamics.

### B.1.1    ADDITIONAL MODELING DETAILS OF THE INVOLVED DYNAMICS

We use $X_{t,i}$'s to represent three different patterns (instead of the number of count) of variables with respect to how observed features are caused by the protected feature $A_t$ and the latent causal factor $H_t$. There are three types of observed features: (1) features that only have the latent causal factor $H_t$ as the case, e.g., $X_{t,1}$, (2) features that have both the latent causal factor $H_t$ and the protected feature $A_t$ as cause, e.g., $X_{t,2}$, and (3) features that only have the protected feature $A_t$ as the cause, e.g,. $X_{t,3}$. For conciseness, we omit features that are not relevant to the practical scenario of interest, i.e., variables that are not causally relevant to $(H_t, A_t)$. One can replace $X_{t,i}$'s with the actual number of additional features together with the causal relations among them in specific practical scenarios.

At every time step $T = t$, the decision-making strategy $D_t$ is trained on the joint distribution $(A_t, X_{t,i}, Y_t^{(\mathrm{obs})})$. However, when making the decision, $D_t$ only takes $(A_t, X_{t,i})$ as input. Since we are modeling causal relations in data generating processes, we only include a directed edge in the DAG if there is a causal relation between variables. Therefore, the data generating process of $D_t$ does not involve an edge between $Y_t^{(\mathrm{obs})}$ and $D_t$.

### B.1.2    THE PRACTICAL SCENARIOS OF INTEREST

As we can see from previous literature (discussed in Appendix A.2), the modeling choices are closely related to the practical scenarios of interest, and therefore, can be very different in terms of modeling details of the involved dynamics in long-term and dynamic settings.

Our causal modeling of repetitive resource application and allocation keeps track of individual-level situation changes, and enables informative and principled analysis on the decision-distribution interplay in different practical scenarios. For example, in credit application (e.g., Liu et al. 2018), the agents are clients and the latent causal factor (tier) can be individual's socio-economic status or creditworthiness; in predictive policing (e.g., Ensign et al. 2018), the agents are neighborhoods and the latent tier can be neighborhood's safety ratings; in the dual market pipeline (e.g., temporary labor markets followed by the permanent labor market considered in Hu & Chen 2018) or the admission-followed-by-hiring pipeline (e.g., Kannan et al. 2019), the agents are applicants who subject to a sequence of decisions and the latent tier can be the relevant qualification for the school program and the job.

However, when the decision received by the individual is once in a lifetime (or at least very long time compared to the timescale of the decision-making), repeated application and allocation of resource may not be a suitable modeling choice. For example, college admission decisions are made on a yearly basis but an individual does not repeatedly apply for college every single year (Mouzannar et al., 2019). In this case, if we focus on the decision made by a specific college, it is more natural to study changes in the population in terms of the group-level qualification profiles (Mouzannar et al., 2019). As another example, in the context of health care (e.g., Mhasawade & Chunara 2021), when the resource takes the form of the medical treatment for the purpose of improving health outcome, not all treatment requires regular doses and therefore, repeated allocation modeled on the individual-level may not be an optimal choice. One can, for instance, resort to the modeling at the level of subgroups as an alternative (Mhasawade & Chunara, 2021).

Considering the difference in semantics of fairness in various practical scenarios, previous literature has pointed out that there is in general no one-size-fits-all solution for algorithmic fairness (e.g., Kearns & Roth 2019). By presenting a detailed causal modeling for the decision-distribution interplay, we do not intend to provide a general framework to encompass long-term fairness considerations in all practical scenarios. Instead, we would like to demonstrate the opportunities and challenges and hope our work can inspire further research.

## B.2 WHEN TIER BALANCING IS INITIALLY SATISFIED

In the paper we have presented possibility and impossibility results to achieve, or get closer to, the long-term fairness goal when *Tier Balancing* is not initially satisfied. It is natural to wonder what we should do if we find out that *Tier Balancing* happen to be satisfied during fairness audit. In fact, as we shall see in Proposition B.1, if *Tier Balancing* is satisfied as the initial condition, under the specified dynamics, one can use *Demographic Parity* (Calders et al., 2009) decision-making strategy to maintain the status of satisfying *Tier Balancing*. This indicates that when *Tier Balancing* is satisfied (as a lucky initial condition, or as a result of $K$-step interventions), we have at least one way to maintain the fair state of affairs.

**Proposition B.1.** *When Tier Balancing is initially satisfied, i.e., $H_t \perp\!\!\!\perp A_t$, under Fact 3.2, Assumption 3.3, and Assumption 3.4, as well as the specified dynamics, the Demographic Parity decision-making strategy, i.e., $D_t \perp\!\!\!\perp A_t$, can ensure Tier Balancing still holds true for the next time step, i.e., $H_{t+1} \perp\!\!\!\perp A_{t+1}$.*

*Proof.* To begin with, since $H_t \perp\!\!\!\perp A_t$, by Fact 3.2, $H_t$ is not a function of $A_t$. As a direct result, $Y_t^{(\text{ori})}$ is also not a function of $A_t$ (since the distribution of $Y_t^{(\text{ori})}$ is fully determined by the value of $H_t = h_t$). Besides, since $D_t$ satisfies *Demographic Parity*, $D_t \perp\!\!\!\perp A_t$, and therefore by Fact 3.2, $D_t$ is not a function of $A_t$.

According to Assumption 3.3, under the specified dynamics, $H_{t+1}$ is fully determined by $(H_t, D_t, Y_t^{(\text{ori})})$, among which none of them is a function of $A_t$. Then, we have $H_{t+1}$ is not a function of $A_t$.

Recall that in the specified dynamics, the same group of agents repetitively apply for credit with the entire group unchanged. According to Assumption 3.4, $A_{t+1}$ is an identical copy of $A_t$. Therefore we have $H_{t+1}$ is not a function of $A_{t+1}$, i.e., $H_{t+1} \perp\!\!\!\perp A_{t+1}$. □

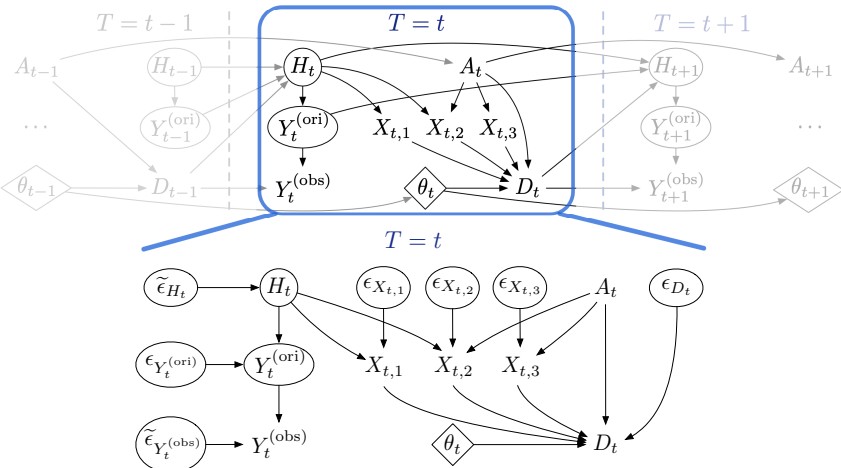

Figure 4: The causal modeling of the decision-distribution interplay. The circle indicates that the corresponding variable is unobserved. We use diamond to denote the underlying causal factor and explicitly indicate the (potential) non-stationary nature of the decision-making strategies across time.

### B.3 A REMARK ON FACT 3.2

Let us first present the definition of a functional causal model (Spirtes et al., 1993; Pearl, 2009):

**Definition B.2 (Functional Causal Model).** We can represent a causal model with a tuple $(E, V, \mathbf{F})$ such that:

(1) $V$ is a set of observed variables involved in the system of interest;

(2) $E$ is a set of exogenous variables that we cannot directly observe but contains the background information representing all other causes of $V$ and jointly follows a distribution $P(E)$;

(3) $\mathbf{F}$ is a set of functions (also known as structural equations) $\{f_1, f_2, \ldots, f_n\}$ where each $f_i$ corresponds to one variable $V_i \in V$ and is a mapping $E \cup V \setminus \{V_i\} \to V_i$.

The triplet $(E, V, \mathbf{F})$ is known as the functional causal model (FCM). We can also capture causal relations among variables via a directed acyclic graph (DAG), where nodes (vertices) represent variables and edges represent functional relations between variables and the corresponding direct causes (i.e., observed parents and unobserved exogenous terms).

For the purpose of illustration, in Figure 4 we present the DAG (at time step $T = t$) with the exogenous terms $E_t$ explicitly modeled, where $E_t$ is the concatenation of individual exogenous terms:

$$E_t = \left(\widetilde{\epsilon}_{H_t}, \widetilde{\epsilon}_{Y_t^{(\text{obs})}}, \epsilon_{Y_t^{(\text{ori})}}, \epsilon_{X_{t,i}}, \epsilon_{D_t}\right). \tag{B.1}$$

We use the $\widetilde{\cdot}$ symbol on certain exogenous noise terms, e.g., $\widetilde{\epsilon}_{H_t}$ and $\widetilde{\epsilon}_{Y_t^{(\text{obs})}}$, to denote the fact that the corresponding variables are affected by previous time step ($T = t - 1$), and such influence are encapsulated into exogenous terms from the standpoint of current time step ($T = t$). For example, the influence from the randomness in $D_{t-1}$ (when $T = t - 1$) on current $Y_t^{(\text{obs})}$ is encapsulated into an exogenous term $\widetilde{\epsilon}_{Y_t^{(\text{obs})}}$ when $T = t$.

As we can see from Figure 4, $(A_t, E_t)$ are root causes of all other variables $(H_t, X_{t,i}, Y_t^{(\text{ori})}, Y_t^{(\text{obs})}, D_t)$. Applying Definition B.2, we can utilize the functional causal model and represent each variable with a function (the structural equation) of its direct causes (including observed parents and unobserved exogenous terms). Then, we can iteratively replace variables with its corresponding structural equation and eventually represent variables in $(H_t, X_{t,i}, Y_t^{(\text{ori})}, Y_t^{(\text{obs})}, D_t)$ with functions of *only* root causes $(A_t, E_t)$, as summarized in Fact 3.2.

The noise terms $E_t$ are the unobserved exogenous terms that signify the unique characteristics of an individual. The utilization of such uniqueness of individual can be found in the estimation of counterfactual causal effect by making use of the posterior distribution of exogenous noise terms conditioning on the observed features, e.g., Kusner et al. (2017).

## B.4 Detailed Derivation of $\Delta_{\text{STIR}}|_t^{t+1}$ (Section 3.2.1)

In this section, we provide the derivation detail of the *Single-step Tier Imbalance Reduction* (STIR) term:

$$\Delta_{\text{STIR}}|_t^{t+1} := \mathbb{E}\left[\,|f_{t+1}(0, E_{t+1}) - f_{t+1}(1, E_{t+1})|\,\right] - \mathbb{E}\left[\,|f_t(0, E_t) - f_t(1, E_t)|\,\right] \quad \text{(B.2)}$$

Firstly, in Appendix B.4.1, we characterize the conditional joint density of $\big(f_T(0, E_T), f_T(1, E_T)\big)$. Then, in Appendix B.4.2, we focus on the situation changes of each individual from $T = t$ to $T = t + 1$ induced by the specified dynamics. Finally, in Appendix B.4.3, we can calculate the expectation in Equation B.2 by aggregating situation changes for each individual from $T = t$ to $T = t + 1$.

### B.4.1 Characterizing Conditional Joint Density

We can view $f_t(0, E_t)$ and $f_t(1, E_t)$ as two dependent random variables. Given combinations of $D_t$ and $Y_t$, we can define their conditional joint probability density when $E_t = \epsilon$ as $q_t\big(f_t(0, \epsilon), f_t(1, \epsilon) \mid d, d', y, y'\big)$ and calculate it as following:

$$
\begin{aligned}
&q_t\big(f_t(0, \epsilon), f_t(1, \epsilon) \mid d, d', y, y'\big) \\
&:= q_t\big(f_t(0, \epsilon), f_t(1, \epsilon) \mid g_t^D(0, \epsilon) = d, g_t^D(1, \epsilon) = d', g_t^{Y^{(\text{ori})}}(0, \epsilon) = y, g_t^{Y^{(\text{ori})}}(1, \epsilon) = y'\big) \\
&= \int_{\xi \in \mathcal{E}} \mathbb{1}\{f_t(0, \xi) = f_t(0, \epsilon), f_t(1, \xi) = f_t(1, \epsilon)\} \\
&\qquad \cdot p_t\big(E_t = \xi \mid g_t^D(0, \epsilon) = d, g_t^D(1, \epsilon) = d', g_t^{Y^{(\text{ori})}}(0, \epsilon) = y, g_t^{Y^{(\text{ori})}}(1, \epsilon) = y'\big) d\xi,
\end{aligned}
\quad \text{(B.3)}
$$

where $\mathbb{1}\{\cdot\}$ is the indicator function, and the subscript $t$ of the conditional probability densities (e.g., $q_t(\cdot)$ and $p_t(\cdot)$) indicates that they (might) change over time with different time step $T = t$. The functional form of $f_t$ can be convoluted and it is not necessarily the case that $f_t(0, \cdot)$ and $f_t(1, \cdot)$ are injective mappings $\mathcal{E} \to (0, 1]$. Therefore, for the purpose of generality, in Equation B.3 we explicitly introduce the identity function $\mathbb{1}\{f_t(0, \xi) = f_t(0, \epsilon), f_t(1, \xi) = f_t(1, \epsilon)\}$ when characterizing the conditional joint density $q_t$.

### B.4.2 Capturing Situation Changes for An Individual

For a specific individual $(j)$, given the value of individual's exogenous terms $E_t^{(j)} = e_t^{(j)}$, let us denote the difference between $f_t(0, e_t^{(j)})$ and $f_t(1, e_t^{(j)})$ as $\varphi_t(e_t^{(j)}) := f_t(0, e_t^{(j)}) - f_t(1, e_t^{(j)})$, and the sum of $f_t(0, e_t^{(j)})$ and $f_t(1, e_t^{(j)})$ as $\eta_t(e_t^{(j)}) := f_t(0, e_t^{(j)}) + f_t(1, e_t^{(j)})$. We introduce $\varphi_t(\cdot)$ and $\eta_t(\cdot)$ for the conciseness of notation, and we can always map $\big(\varphi_t(\cdot), \eta_t(\cdot)\big)$ back to $\big(f_t(0, \cdot), f_t(1, \cdot)\big)$ via a coordinate transformation:

$$
\begin{bmatrix} f_t(0, e_t^{(j)}) \\ f_t(1, e_t^{(j)}) \end{bmatrix} = \frac{\sqrt{2}}{2} \begin{bmatrix} \cos\frac{\pi}{4} & \sin\frac{\pi}{4} \\ -\sin\frac{\pi}{4} & \cos\frac{\pi}{4} \end{bmatrix} \begin{bmatrix} \varphi_t(e_t^{(j)}) \\ \eta_t(e_t^{(j)}) \end{bmatrix}.
\quad \text{(B.4)}
$$

Let us consider the connection between $\varphi_{t+1}(e_{t+1}^{(j)}) = f_{t+1}(0, e_{t+1}^{(j)}) - f_{t+1}(1, e_{t+1}^{(j)})$ in the time step $T = t + 1$ and $\varphi_t(e_t^{(j)}) = f_t(0, e_t^{(j)}) - f_t(1, e_t^{(j)})$ in the time step $T = t$. We use different time step subscripts for the exogenous terms, e.g., $e_{t+1}^{(j)}$ in $\varphi_{t+1}(e_{t+1}^{(j)})$ and $e_t^{(j)}$ in $\varphi_t(e_t^{(j)})$, since it is not necessarily the case that $e_{t+1}^{(j)} = e_t^{(j)}$, even if we are focusing on the same individual from $T = t$ to $T = t + 1$. Nevertheless, for the given functional forms of $f_t, g_t^D, g_t^{Y^{(\text{ori})}}$, the combination of $(d^{(j)}, d'^{(j)}, y^{(j)}, y'^{(j)})$, the value of exogenous term $e_t^{(j)}$ in the initial situation of the current one-step analysis (when $T = t$), and the hyperparameters $(\alpha_D, \alpha_Y)$, we can uniquely derive the value of

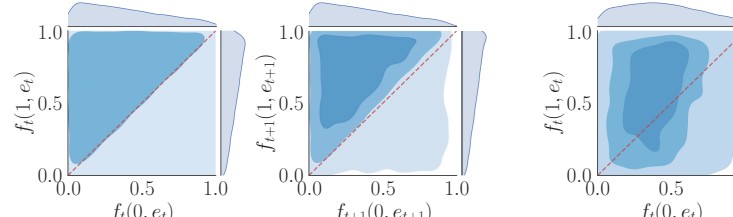

(a) the example of distribution change when the initial condition satisfies the quantitative assumption

(b) the example of distribution change when the initial condition satisfies the qualitative assumption

Figure 5: An illustration of the connection between qualitative and quantitative assumptions in terms of the one-step update of the conditional joint distribution $q_T\big(f_T(0, e_T), f_T(1, e_T)\ |\ d, d', y, y'\big)$ (when $y < y'$, and from $T = t$ to $T = t + 1$).

$\varphi_{t+1}(e_{t+1}^{(j)}) = f_{t+1}(0, e_{t+1}^{(j)}) - f_{t+1}(1, e_{t+1}^{(j)})$, and list all possible instantiations of $\varphi_{t+1}(e_{t+1}^{(j)})$ in Table 4 (if $\alpha_D > \alpha_Y$), Table 5 (if $\alpha_D < \alpha_Y$), and Table 6 (if $\alpha_D = \alpha_Y$).

Let us denote such mapping from $\varphi_t(e_t^{(j)})$ to $\varphi_{t+1}(e_{t+1}^{(j)})$ with the function $G_t$. For the purpose of simplifying notations, we can omit the superscript $(j)$ if without ambiguity, since the value of exogenous terms $e_T$ signify the unique characteristics of an individual:

$$\varphi_{t+1}(e_{t+1}) := f_{t+1}(0, e_{t+1}) - f_{t+1}(1, e_{t+1}) = G_t(f_t, g_t^D, g_t^{Y^{(\mathrm{ori})}}; d, d', y, y', e_t, \alpha_D, \alpha_Y). \quad \text{(B.5)}$$

Notice that the value of the function $G_t$ *only* relies on the information available at time step $T = t$.

### B.4.3 Aggregating Individual-Level Situation Changes

We can calculate *Single-step Tier Imbalance Reduction*, i.e., the term $\Delta_{\mathrm{STIR}}|_t^{t+1}$, as following:

$$\Delta_{\mathrm{STIR}}|_t^{t+1} := \mathbb{E}\big[\,|f_{t+1}(0, E_{t+1}) - f_{t+1}(1, E_{t+1})|\,\big] - \mathbb{E}\big[\,|f_t(0, E_t) - f_t(1, E_t)|\,\big]$$

$$\overset{(i)}{=} \mathbb{E}\big[\,|\varphi_{t+1}(E_{t+1})| - |\varphi_t(E_t)|\,\big]$$

$$\overset{(ii)}{=} \mathbb{E}\Big\{\mathbb{E}\Big[\,|\varphi_{t+1}(\xi)| - |\varphi_t(\epsilon)|\ \Big|\ \underbrace{E_{t+1} = \xi,}_{\text{The value of exogenous terms of an individual take value } \xi \text{ at } T = t+1.}$$

$$\underbrace{E_t = \epsilon,}_{\text{The value of exogenous terms of an individual take value } \epsilon \text{ at } T = t.}$$

$$\underbrace{\varphi_{t+1}(\xi) = G_t(f_t, g_t^D, g_t^{Y^{(\mathrm{ori})}}; d, d', y, y', \epsilon, \alpha_D, \alpha_Y)}_{\substack{\text{This is to make sure that we are keeping track of the same individual in the sense that,} \\ \varphi_{t+1}(\xi) \text{ when } T = t+1 \text{ is indeed a valid instantiation from } \varphi_t(\epsilon) \text{ when } T = t. \\ \text{If } \varphi_{t+1}(\cdot) \text{ is not a valid instantiation from } \varphi_t(\cdot), \text{ the contribution to the expectation is 0.}}\Big]\Big\}$$

$$\overset{(iii)}{=} \sum_{d, d', y, y' \in \{0, 1\}} P_t(d, d', y, y') \cdot \int_{\epsilon \in \mathcal{E}} \int_{\xi \in \mathcal{E}} q_t\big(f_t(0, \epsilon), f_t(1, \epsilon)\ |\ d, d', y, y'\big)$$

$$\cdot \big(|\varphi_{t+1}(\xi)| - |\varphi_t(\epsilon)|\big) \cdot \mathbb{1}\{\varphi_{t+1}(\xi) = G_t(f_t, g_t^D, g_t^{Y^{(\mathrm{ori})}}; d, d', y, y', \epsilon, \alpha_D, \alpha_Y)\}d\xi d\epsilon, \tag{B.6}$$

where the equality (i) is based on the definition of $\varphi_t(\cdot)$ and $\varphi_{t+1}(\cdot)$; the equality (ii) is derived from the Law of Iterated Expectation, keeping track of individual-level situation changes in the inner conditional expectation; the equality (iii) is the aggregation of individual-level situation changes by plugging in the conditional joint density $q_t$ calculated in Appendix B.4.1, joint probability $P_t$, and the individual-level situation changes discussed in Appendix B.4.2.

The indicator function $\mathbb{1}\{\varphi_{t+1}(\xi) = G_t(f_t, g_t^D, g_t^{Y^{(\mathrm{ori})}}; d, d', y, y', \epsilon, \alpha_D, \alpha_Y)\}$ makes sure that we are keeping track of the same individual (whose exogenous noise term equals to $\epsilon$ at time $t$) before and after the one-step dynamic, even if his/her exogenous noise term equals to $\xi$ at time $t + 1$, and that $\epsilon$ might not be equal to $\xi$.

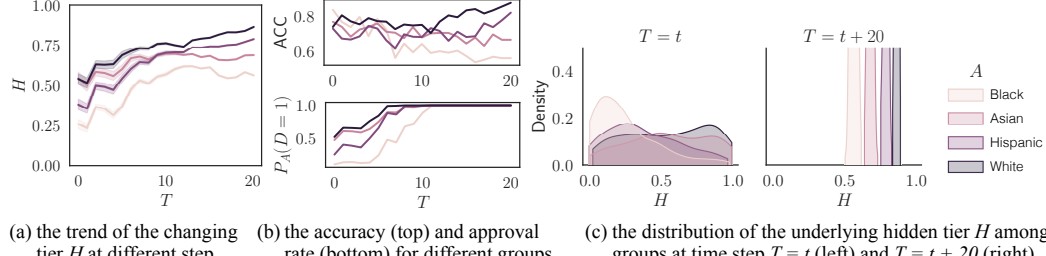

(a) the trend of the changing tier $H$ at different step

(b) the accuracy (top) and approval rate (bottom) for different groups

(c) the distribution of the underlying hidden tier $H$ among groups at time step $T = t$ (left) and $T = t + 20$ (right)

Figure 6: Illustration of the interplay between decision with accuracy-oriented predictors and the data dynamics (20 steps) on the credit score data set. Panel (a) and (b) present the step-by-step tracks of update in tier, accuracy, and approval rates for different groups; panel (c) presents group-conditioned distributions of tier before (left) and after (right) 20 steps of interventions. The legend is shared across panel (a), (b), and (c).

The fact that we are keeping track of the same individual also justifies the practice of only integrating over (conditional) densities with subscript $t$, e.g., $q_t(\cdot)$ and $P_t(\cdot)$, instead of both $t$ and $t + 1$. To see this from a different angle, keeping track of situation changes of each individual (when comparing $\varphi_{t+1}(e_{t+1})$ with $\varphi_t(e_t)$) also alleviates us from the trouble of estimating (conditional) densities that involve future information. At the time step $t$, we do not know the densities $q_{t+1}\big(f_{t+1}(0, E_{t+1}), f_{t+1}(1, E_{t+1}) \mid d, d', y, y'\big)$ and $P_{t+1}(d, d', y, y')$ since they involve future information $D_{t+1}$ and $Y_{t+1}$ at the standpoint of time step $T = t$.

## B.5 Further Illustration on Assumption 3.5 and Assumption 3.6

In this subsection, we provide further illustrations of the connection between Assumption 3.5 and Assumption 3.6. In Figure 5 we present the one-step update of the conditional joint distribution $q_T\big(f_T(0, e_T), f_T(1, e_T) \mid d, d', y, y'\big)$ from $T = t$ to $T = t + 1$ (we present the case when $y < y'$ as an example). For panel (a) and (b), the joint distribution of $(f_T(0, E_T), f_T(1, E_T))$ is plotted before and after one-step dynamics, with quantitative and qualitative assumptions respectively. The distributions are color-coded, the deeper the color, the larger the value of the joint density.

Compared to the qualitative assumption (Assumption 3.5, illustrated in Figure 5b), the quantitative assumption (Assumption 3.6, illustrated in Figure 5a) is just a special case, with quantitative characteristics built-in for technical purposes (we will make use of Assumption 3.6 in the proofs for Theorem 4.2 and Theorem 4.3 in Appendix C). From the illustrations in Figure 5, we can also see that the behaviors of the one-step update of conditional joint density under qualitative and quantitative assumptions are similar, with deeper color patterns occurring on the upper-left corner, indicating similar changes in the corresponding conditional joint densities $q_T\big(f_T(0, e_T), f_T(1, e_T) \mid d, d', y, y'\big)$ (when $y < y'$, and from $T = t$ to $T = t + 1$).

## B.6 Additional Experimental Results

In this section, we present additional experimental results on the preprocessed FICO credit score data set (Board of Governors of the Federal Reserve System (US), 2007; Hardt et al., 2016). Similar to the experiment summarized in Figure 3, we convert the cumulative distribution function (CDF) of group-wise TransRisk scores into group-wise density distributions of the credit score, and use them as the initial tier distributions for different groups.

We consider utility-maximizing decision-making strategies, i.e., the decision-making policy is accuracy oriented and there is no explicit fairness consideration. In Figure 6 we present the summary of a 20-step interplay between decision with accuracy-oriented predictors and the underlying data generating process on the credit score data set. The accuracy-oriented decision-making strategy is retrained after each one-step data dynamics. From Figure 6(a), there is no obvious evidence that the gap between step-by-step tracks of tiers for different groups is decreasing over time. This observation aligns with our theoretical analysis (Theorem 4.2) and simulation results (Figure 2) for perfect predictors.

### B.7 POTENTIAL LIMITATIONS OF OUR WORK

In this subsection, we discuss potential limitations of our work.

#### B.7.1 SPECIFIED DYNAMICS VS. CAUSAL DISCOVERY

In this paper, we present a detailed causal modeling of decision-distribution interplay on DAG (Section 2.1) and formulate the dynamic fairness notion, *Tier Balancing*, that captures the long-term fairness goal over the underlying causal factor.

The research of causal discovery, where the goal is to discover the causal relations among variables (Spirtes et al., 1993; Shimizu et al., 2006; Zhang & Hyvärinen, 2009; Zhang et al., 2011), is a highly relevant area but is out of the scope of our paper. Our *Tier Balancing* notion of dynamic fairness, as well as our analyzing framework, does not rely on a causal model derived from causal discovery. As we discussed in the comparison with previous literature in dynamic fairness studies (Appendix A), our causal model is richer and more complex, which provides the potential of a more principled reasoning of the essential decision-distribution interplay in the pursuit of long-term fairness. We acknowledge the fact that it is nice to have the ability to discover the underlying causal model of the involved dynamics, which would provide further refinements of our dynamic modeling based on the specific practical scenario of interest. Causal discovery can act as the icing on the cake, but not a necessary component, of our analysis.

#### B.7.2 THE NUMBER AND DIMENSION OF LATENT CAUSAL FACTORS

In the causal modeling of decision-distribution interplay we present in the paper, we consider one latent causal factor that carries on the influence of current decision to future distributions. Recent developments in the identification of causal structures that involve (more than one) latent factors (Xie et al., 2020; Adams et al., 2021; Kivva et al., 2021; Xie et al., 2022) provide not only a theoretical justification, but also an indication of the potential, for our effort in exploring long-term fairness inquires over latent causal factors. We believe that our detailed causal modeling of decision-distribution interplay (on both observed variables and latent causal factors) and our formulation of *Tier Balancing* notion of long-term fairness act as an important first step.

## C PROOF OF RESULTS

In this section, we provide proofs for results presented in the paper. For better readability, we provide an additional *Proof (sketch)* before proving Theorem 4.1 (proof in Appendix C.2), Theorem 4.2 (proof in Appendix C.3), and Theorem 4.3 (proof in Appendix C.4), respectively.

### C.1 PROOF FOR PROPOSITION 3.1

**Proposition.** *At time step $T = t$, for any $H_t = h_t \in (0, 1]$, under the specified dynamics, among the population where ground truth is actually observable, i.e., $Y_t^{(obs)}$ is not undefined, we have:*

$$Y_t^{(obs)} \sim \text{Bernoulli}(h_t).$$

*Proof.* To begin with, according the d-separation relation among $D_{t-1}$, $H_t$, and $Y_t^{(ori)}$ on Figure 1, we notice that $Y_t^{(ori)} \perp\!\!\!\perp D_{t-1} \mid H_t$. Therefore we have:

$$Y_t^{(ori)} \sim \text{Bernoulli}(h_t),$$
$$P(Y_t^{(ori)} = 1 \mid H_t = h_t) = h_t,$$
$$P(D_{t-1} = 1 \mid H_t = h_t) = d(h_t),$$

where $d(\cdot)$ is a function $d : (0, 1] \to [0, 1]$.

Notice that there is no claim that $D_{t-1}$ can be uniquely determined by a function of only $h_t$. We only represent the conditional probability mass $P(D_{t-1} = 1 \mid H_t = h_t)$ with a function of $h_t$ without specifying the exact functional form. In fact, as we shall see in the later part of this proof, the exact functional form of $d(\cdot)$ does not affect the validity of the result.

Since $Y_t^{(\text{obs})}$ is in fact $Y_t^{(\text{ori})}$ masked by $D_{t-1}$, i.e., $Y_t^{(\text{obs})}$ is observable only when $D_{t-1} = 1$ and is undefined when $D_{t-1} = 0$, we have:

$$P(D_{t-1} = 0, Y_t^{(\text{obs})} = 0 \mid H_t = h_t) = P(D_{t-1} = 0, Y_t^{(\text{obs})} = 1 \mid H_t = h_t) = 0.$$

This indicates that among the population where $Y_t^{(\text{obs})}$ is not undefined (the population itself may change at different time step), $\forall y \in \{0, 1\}$:

$$
\begin{aligned}
&P(Y_t^{(\text{obs})} = y \mid H_t = h_t) \\
&= P(D_{t-1} = 0, Y_t^{(\text{obs})} = y \mid H_t = h_t) + P(D_{t-1} = 1, Y_t^{(\text{obs})} = y \mid H_t = h_t) \\
&= P(D_{t-1} = 1, Y_t^{(\text{obs})} = y \mid H_t = h_t) \\
&= P(D_{t-1} = 1, Y_t^{(\text{ori})} = y \mid H_t = h_t).
\end{aligned}
$$

Then, when $d(h_t) \in (0, 1)$, we can calculate the following probability:

$$
\begin{aligned}
&P(Y_t^{(\text{obs})} = 1 \mid H_t = h_t) \\
&= \frac{P(Y_t^{(\text{obs})} = 1 \mid H_t = h_t)}{P(Y_t^{(\text{obs})} = 1 \mid H_t = h_t) + P(Y_t^{(\text{obs})} = 0 \mid H_t = h_t)} \\
&= \frac{P(D_{t-1} = 1, Y_t^{(\text{ori})} = 1 \mid H_t = h_t)}{P(D_{t-1} = 1, Y_t^{(\text{ori})} = 1 \mid H_t = h_t) + P(D_{t-1} = 1, Y_t^{(\text{ori})} = 0 \mid H_t = h_t)} \\
&= \frac{d(h_t) h_t}{d(h_t) h_t + d(h_t)(1 - h_t)} \\
&= h_t \\
&= P(Y_t^{(\text{ori})} = 1 \mid H_t = h_t);
\end{aligned}
$$

when $d(h_t) = 1$, this indicates that if $H_t = h_t$, we know for sure that this individual received a positive decision in the previous time step (when $T = t - 1$), and we have $Y_t^{(\text{ori})} = Y_t^{(\text{obs})}$ by definition; when $d(h_t) = 0$, this indicates that if $H_t = h_t$, we know for sure that this individual did not receive a positive decision in the previous time step (when $T = t - 1$), and in this case $Y_t^{(\text{obs})}$ is undefined.

Therefore, among the population where ground truth is actually observable, i.e., $Y_t^{(\text{obs})}$ is not undefined, we have:

$$Y_t^{(\text{obs})} \sim \text{Bernoulli}(h_t).$$

$\square$

## C.2 PROOF FOR THEOREM 4.1

**Theorem.** *Let us consider the general situation where both $D_t$ and $Y_t^{(\text{ori})}$ are dependent with $A_t$, i.e., $D_t \not\perp\!\!\!\perp A_t, Y_t^{(\text{ori})} \not\perp\!\!\!\perp A_t$. Then under Fact 3.2, Assumption 3.3, and Assumption 3.4, as well as the specified dynamics, when $H_t \not\perp\!\!\!\perp A_t$, only if at least one of the following conditions holds true for all $e_t \in \mathcal{E}$ can we possibly attain $H_{t+1} \perp\!\!\!\perp A_{t+1}$:*

*(1) The ratio $\frac{f_t(0, e_t)}{f_t(1, e_t)}$ has a specific domain of value:*

$$\frac{f_t(0, e_t)}{f_t(1, e_t)} = \frac{1 \pm \alpha_D \pm \alpha_Y}{1 \pm \alpha_D \pm \alpha_Y};$$

*(2) Positive (negative) labels only appear in the advantaged (disadvantaged) group, and the decision for everyone is positive (if $\alpha_D > \alpha_Y$):*

$$
\begin{cases}
f_t(0, e_t) \in [\frac{1}{1 + \alpha_D - \alpha_Y}, 1], \\
f_t(1, e_t) \in [\frac{1}{1 + \alpha_D + \alpha_Y}, 1], \\
g_t^{Y^{(\text{ori})}}(0, e_t) = 0, g_t^{Y^{(\text{ori})}}(1, e_t) = 1, \\
g_t^D(0, e_t) = g_t^D(1, e_t) = 1;
\end{cases}
$$

*(3) Negative (positive) labels only appear in the advantaged (disadvantaged) group, and the decision for everyone is positive (if $\alpha_D > \alpha_Y$):*

$$
\begin{cases}
f_t(0, e_t) \in [\frac{1}{1+\alpha_D+\alpha_Y}, 1], \\
f_t(1, e_t) \in [\frac{1}{1+\alpha_D-\alpha_Y}, 1], \\
g_t^{Y^{(ori)}}(0, e_t) = 1, g_t^{Y^{(ori)}}(1, e_t) = 0, \\
g_t^D(0, e_t) = g_t^D(1, e_t) = 1;
\end{cases}
$$

*(4) Everyone has a positive label, but the positive decision is exclusive to the advantaged group (if $\alpha_D < \alpha_Y$):*

$$
\begin{cases}
f_t(0, e_t) \in [\frac{1}{1-\alpha_D+\alpha_Y}, 1], \\
f_t(1, e_t) \in [\frac{1}{1+\alpha_D+\alpha_Y}, 1], \\
g_t^{Y^{(ori)}}(0, e_t) = g_t^{Y^{(ori)}}(1, e_t) = 1, \\
g_t^D(0, e_t) = 0, g_t^D(1, e_t) = 1;
\end{cases}
$$

*(5) Everyone has a positive label, but the positive decision is exclusive to the disadvantaged group (if $\alpha_D < \alpha_Y$):*

$$
\begin{cases}
f_t(0, e_t) \in [\frac{1}{1+\alpha_D+\alpha_Y}, 1], \\
f_t(1, e_t) \in [\frac{1}{1-\alpha_D+\alpha_Y}, 1], \\
g_t^{Y^{(ori)}}(0, e_t) = g_t^{Y^{(ori)}}(1, e_t) = 1, \\
g_t^D(0, e_t) = 1, g_t^D(1, e_t) = 0.
\end{cases}
$$

*Proof (sketch).* In order to see the exact condition under which it is possible to achieve $H_{t+1} \perp\!\!\!\perp A_{t+1}$, we consider the necessary and sufficient condition such that $H_{t+1} = f_{t+1}(A_{t+1}, E_{t+1})$ is not a function of $A_{t+1}$. This, together with Fact 3.2, Assumption 3.3, and Assumption 3.4, indicates that we need to consider the condition under which $H_{t+1} = \min\left\{1, f_t(A_t, E_t)\left[1 + \alpha_D(2D_t - 1) + \alpha_Y(2Y_t^{(ori)} - 1)\right]\right\}$ is not a function of $A_t$.

Since both $D_t$ and $Y_t^{(ori)}$ are binary, we can exhaustively consider all value combinations of $D_t$ and $Y_t^{(ori)}$, and list every possible value $H_{t+1}$ can take in each case in Table 1 (if $\alpha_D > \alpha_Y$), Table 2 (if $\alpha_D < \alpha_Y$), or Table 3 (if $\alpha_D = \alpha_Y$). By exhaustively going through possible cases, we can have a full picture of the update of $H_{t+1}$ based on $(H_t, Y_t^{(ori)}, D_t)$, and then derive conditions under which $H_{t+1}$ is *not* a function of $A_t$, i.e., we have the conditions under which it is possible to attain $H_{t+1} \perp\!\!\!\perp A_{t+1}$. □

*Proof (full).* In order to see the exact condition under which it is possible to achieve $H_{t+1} \perp\!\!\!\perp A_{t+1}$, we consider the necessary and sufficient condition such that $H_{t+1} = f_{t+1}(A_{t+1}, E_{t+1})$ is not a function of $A_{t+1}$. By Fact 3.2, Assumption 3.3, and Assumption 3.4, it is necessary and sufficient to consider the condition under which $H_{t+1} = \min\left\{1, f_t(A_t, E_t)\left[1 + \alpha_D(2D_t - 1) + \alpha_Y(2Y_t^{(ori)} - 1)\right]\right\}$ is not a function of $A_t$.

Considering the fact that both $D_t$ and $Y_t^{(ori)}$ are binary, we can compare the values of $H_{t+1}$ when $A_t = 0$ and $A_t = 1$ for all possible combinations of $D_t$ and $Y_t^{(ori)}$. For any fixed $e_t \in \mathcal{E}$, we can list all the cases in Table 1 (if $\alpha_D > \alpha_Y$), Table 2 (if $\alpha_D < \alpha_Y$), or Table 3 (if $\alpha_D = \alpha_Y$), and see if for all $e_t \in \mathcal{E}$, there is no difference in the value of $H_{t+1}$ between the cases when $A_t = 0$ and $A_t = 1$.

From Table 1, Table 2, and Table 3, we can see that if and only the following hold true can we achieve $H_{t+1} \perp\!\!\!\perp A_{t+1}$: for every $e_t \in \mathcal{E}$, whenever the joint probability $P\big(g_t^D(0, e_t) = d, g_t^{Y^{(ori)}}(0, e_t) = y, g_t^D(1, e_t) = d', g_t^{Y^{(ori)}}(1, e_t) = y'\big)$ is nonzero, the last two columns of the corresponding row(s) in the table, i.e., the exact values of $H_{t+1}$, need to match. For example, when $\alpha_D > \alpha_Y$, if we know $P\big(g_t^D(0, e_t) = 0, g_t^{Y^{(ori)}}(0, e_t) = 0, g_t^D(1, e_t) = 0, g_t^{Y^{(ori)}}(1, e_t) = 0\big) \neq 0$, we need the last two columns of Case (i) of Table 1 to equal to each other, i.e., we need $f_t(0, e_t) = f_t(1, e_t)$ to hold true.

Without further assumptions on the joint distribution of the data, we do not know which combination of $(d, y, d', y')$ will result in a nonzero joint probability:

$$P\big(g_t^D(0, e_t) = d, g_t^{Y^{(\text{ori})}}(0, e_t) = y, g_t^D(1, e_t) = d', g_t^{Y^{(\text{ori})}}(1, e_t) = y'\big) \neq 0.$$

However, considering the fact that $\sum_{d,d'\in\mathcal{D}, y,y'\in\mathcal{Y}} P\big(g_t^D(0, e_t) = d, g_t^{Y^{(\text{ori})}}(0, e_t) = y, g_t^D(1, e_t) = d', g_t^{Y^{(\text{ori})}}(1, e_t) = y'\big) = 1$ holds for all $e_t \in \mathcal{E}$, we do know that for any fixed $e_t \in \mathcal{E}$, there is at least one possible instantiation of $(d^*, y^*, d'^*, y'^*)$ such that:

$$P\big(g_t^D(0, e_t) = d^*, g_t^{Y^{(\text{ori})}}(0, e_t) = y^*, g_t^D(1, e_t) = d'^*, g_t^{Y^{(\text{ori})}}(1, e_t) = y'^*\big) \neq 0. \quad \text{(C.1)}$$

Let us first consider situations where $\alpha_D > \alpha_Y$ and focus on Table 1. The analysis on situations where $\alpha_D < \alpha_Y$ (i.e., Table 2) or $\alpha_D = \alpha_Y$ (i.e, Table 3), is of the same flavor and therefore we omit the detail in the proof.

To begin with, we can observe that not every entry of the last two columns explicitly keeps the $\min\{\cdot, 1\}$ operator. On the one hand, since $\alpha_D > \alpha_Y$ ($\alpha_D, \alpha_Y \in [0, \frac{1}{2})$, as of Assumption 3.3), we have $(1 - \alpha_D \pm \alpha_Y) \in (0, 1)$ and $f_t(a_t, e_t)(1 - \alpha_D \pm \alpha_Y) \in (0, 1)$ (since $H_t = f_t(a_t, e_t) \in (0, 1]$); therefore, we do not need to keep the $\min\{\cdot, 1\}$ operator explicit, for instance, in the second to last column of Case (v - viii). On the other hand, when the coefficients $(1 + \alpha_D \pm \alpha_Y) > 1$ we are not sure if $f_t(a_t, e_t)(1 + \alpha_D \pm \alpha_Y)$ exceed 1; therefore, we need to keep the $\min\{\cdot, 1\}$ operator explicit, for instance, in the last column of Case (v - viii).

Besides, if only one entry of the last two columns explicitly has the $\min\{\cdot, 1\}$ operator, it is equivalent to require that the terms themselves (before applying the operator) are equal (since the one without the $\min\{\cdot, 1\}$ operator is known to be within the $(0, 1)$ interval). For instance, Case (ix) requires that $\min\{f_t(0, e_t)(1 + \alpha_D - \alpha_Y), 1\} = f_t(1, e_t)(1 - \alpha_D - \alpha_Y)$, which is equivalent to requiring $f_t(0, e_t)(1 + \alpha_D - \alpha_Y) = f_t(1, e_t)(1 - \alpha_D - \alpha_Y)$.

Furthermore, if both entries of the last two columns explicitly has the $\min\{\cdot, 1\}$ operator, the exact condition of matching the last two columns depends on the actual value of $f_t(0, e_t)$ and $f_t(1, e_t)$. For instance, Case (xv) requires that $\min\{f_t(0, e_t)(1 + \alpha_D + \alpha_Y), 1\} = \min\{f_t(1, e_t)(1 + \alpha_D - \alpha_Y), 1\}$, which could be equivalent to one of the following conditions (recall that $1 + \alpha_D \pm \alpha_Y > 1$):

- if we have $f_t(0, e_t) \in [\frac{1}{1+\alpha_D+\alpha_Y}, 1]$ and $f_t(1, e_t) \in [\frac{1}{1+\alpha_D-\alpha_Y}, 1]$, we require $1 = 1$, which trivially holds true;

- if we have $f_t(0, e_t) \in (0, \frac{1}{1+\alpha_D+\alpha_Y})$ and $f_t(1, e_t) \in [\frac{1}{1+\alpha_D-\alpha_Y}, 1]$, we require $f_t(0, e_t)(1 + \alpha_D + \alpha_Y) = 1$, which cannot hold true;

- if we have $f_t(0, e_t) \in [\frac{1}{1+\alpha_D+\alpha_Y}, 1]$ and $f_t(1, e_t) \in (0, \frac{1}{1+\alpha_D-\alpha_Y})$, we require $1 = f_t(1, e_t)(1 + \alpha_D - \alpha_Y)$ which cannot hold true;

- if we have $f_t(0, e_t) \in (0, \frac{1}{1+\alpha_D+\alpha_Y})$ and $f_t(1, e_t) \in (0, \frac{1}{1+\alpha_D-\alpha_Y})$, we require $\frac{f_t(0,e_t)}{f_t(1,e_t)} = \frac{1+\alpha_D-\alpha_Y}{1+\alpha_D+\alpha_Y}$.

Recall that without further assumptions on the data distribution, we do not know which row(s) of the table correspond to a nonzero probability $P\big(g_t^D(0, e_t) = d, g_t^{Y^{(\text{ori})}}(0, e_t) = y, g_t^D(1, e_t) = d', g_t^{Y^{(\text{ori})}}(1, e_t) = y'\big)$. As a result, in general, we do not know which set of requirements we should enforce for each $e_t \in \mathcal{E}$. Therefore, we cannot derive a necessary and sufficient condition for attaining $H_{t+1} \perp\!\!\!\perp A_{t+1}$ in general cases. Nevertheless, we can summarize the previous analysis and derive the necessary condition of attaining $H_{t+1} \perp\!\!\!\perp A_{t+1}$, i.e., only if at least one of the following conditions holds true for all $e_t \in \mathcal{E}$ can we possibly attain $H_{t+1} \perp\!\!\!\perp A_{t+1}$:

(1) The ratio $\frac{f_t(0,e_t)}{f_t(1,e_t)}$ has a specific domain of value:

$$\frac{f_t(0, e_t)}{f_t(1, e_t)} = \frac{1 \pm \alpha_D \pm \alpha_Y}{1 \pm \alpha_D \pm \alpha_Y};$$

(2) Positive (negative) labels only appear in the advantaged (disadvantaged) group, and the decision for everyone is positive (if $\alpha_D > \alpha_Y$):

$$\begin{cases} f_t(0, e_t) \in [\frac{1}{1+\alpha_D-\alpha_Y}, 1], \\ f_t(1, e_t) \in [\frac{1}{1+\alpha_D+\alpha_Y}, 1], \\ g_t^{Y^{(ori)}}(0, e_t) = 0, g_t^{Y^{(ori)}}(1, e_t) = 1, \\ g_t^D(0, e_t) = g_t^D(1, e_t) = 1; \end{cases}$$

(3) Negative (positive) labels only appear in the advantaged (disadvantaged) group, and the decision for everyone is positive (if $\alpha_D > \alpha_Y$):

$$\begin{cases} f_t(0, e_t) \in [\frac{1}{1+\alpha_D+\alpha_Y}, 1], \\ f_t(1, e_t) \in [\frac{1}{1+\alpha_D-\alpha_Y}, 1], \\ g_t^{Y^{(ori)}}(0, e_t) = 1, g_t^{Y^{(ori)}}(1, e_t) = 0, \\ g_t^D(0, e_t) = g_t^D(1, e_t) = 1; \end{cases}$$

(4) Everyone has a positive label, but the positive decision is exclusive to the advantaged group (if $\alpha_D < \alpha_Y$):

$$\begin{cases} f_t(0, e_t) \in [\frac{1}{1-\alpha_D+\alpha_Y}, 1], \\ f_t(1, e_t) \in [\frac{1}{1+\alpha_D+\alpha_Y}, 1], \\ g_t^{Y^{(ori)}}(0, e_t) = g_t^{Y^{(ori)}}(1, e_t) = 1, \\ g_t^D(0, e_t) = 0, g_t^D(1, e_t) = 1; \end{cases}$$

(5) Everyone has a positive label, but the positive decision is exclusive to the disadvantaged group (if $\alpha_D < \alpha_Y$):

$$\begin{cases} f_t(0, e_t) \in [\frac{1}{1+\alpha_D+\alpha_Y}, 1], \\ f_t(1, e_t) \in [\frac{1}{1-\alpha_D+\alpha_Y}, 1], \\ g_t^{Y^{(ori)}}(0, e_t) = g_t^{Y^{(ori)}}(1, e_t) = 1, \\ g_t^D(0, e_t) = 1, g_t^D(1, e_t) = 0. \end{cases}$$

$\square$

## C.3 PROOF FOR THEOREM 4.2

**Theorem.** *Let us consider the general situation where both $D_t$ and $Y_t^{(ori)}$ are dependent with $A_t$, i.e., $D_t \not\perp\!\!\!\perp A_t, Y_t^{(ori)} \not\perp\!\!\!\perp A_t$. Under Fact 3.2, Assumption 3.3, Assumption 3.4, and Assumption 3.6, as well as the specified dynamics, when $H_t \not\perp\!\!\!\perp A_t$, the perfect predictor does not have the potential to get closer to the long-term fairness goal after one-step intervention, i.e.,*

$$D_t = Y_t^{(ori)} \implies \Delta_{\text{STIR}}^{(\text{Perfect Predictor})}\big|_t^{t+1} > 0.$$

*Proof (sketch).* The goal is to calculate if it is possible for *Single-step Tier Imbalance Reduction* $\Delta_{\text{STIR}}|_t^{t+1}$ to be smaller than 0 when using perfect predictors. As defined in Equation 5, $\Delta_{\text{STIR}}|_t^{t+1}$ is a weighted aggregation (integration followed by summation) of $|\varphi(e_{t+1})| - |\varphi(e_t)|$. The quantitative analysis involves three key components: instantiations of $\varphi_{t+1}(e_{t+1})$, the knowledge/assumptions on $q_t(f_t(0, \epsilon), f_t(1, \epsilon) \mid d, d', y, y')$, and characteristics of $P_t(d, d', y', y')$.

For the first component, we can list all possible instantiations of $\varphi_{t+1}(e_{t+1})$ in Table 4 (if $\alpha_D > \alpha_Y$), Table 5 (if $\alpha_D < \alpha_Y$), and Table 6 (if $\alpha_D = \alpha_Y$), respectively. For the second component, we can introduce a quantitative assumption on $q_t(f_t(0, \epsilon), f_t(1, \epsilon) \mid d, d', y, y')$ (Assumption 3.6). For the third component, we need to exploit the characteristic of the predictor of interest to gain further insight into the joint distribution $P_t(d, d', y, y')$. For perfect predictors, we have $P_t(d, d', y, y')$ satisfies Equation 8 (as we have discussed in Section 4.2.1).

For the purpose of calculating the value of $\Delta_{\text{STIR}}|_t^{t+1}$, the proof contains two steps: (1) exhaustively derive the value of $|\varphi(e_{t+1})| - |\varphi(e_t)|$ after one-step dynamics in all possible cases, and (2) aggregate the difference $|\varphi(e_{t+1})| - |\varphi(e_t)|$ with the help of the additional knowledge/assumptions on $q_t(f_t(0, \epsilon), f_t(1, \epsilon) \mid d, d', y, y')$ and $P_t(d, d', y, y')$. $\square$

*Proof (full).* For the perfect predictor $D_t = Y_t^{(\text{ori})}$, among all possible instantiations of $\varphi_{t+1}(e_{t+1})$ as listed in Table 4, Table 5, and Table 6, not every case corresponds to a nonzero $P_t(d, d', y, y')$ and therefore may not contribute to the computation of $\Delta_{\text{STIR}}|_t^{t+1}$ as detailed in Equation 5. By applying Equation 8 we only need to consider Case (i), Case (vi), Case (xi), and Case (xvi) in Table 4 (if $\alpha_D > \alpha_Y$), Table 5 (if $\alpha_D < \alpha_Y$), and Table 6 (if $\alpha_D = \alpha_Y$), respectively. We list all possible values of $|\varphi(e_{t+1})| - |\varphi(e_t)|$ for each of the aforementioned cases (the result applies to scenarios where $\alpha_D > \alpha_Y$, $\alpha_D < \alpha_Y$, or $\alpha_D = \alpha_Y$).

When $(d, d', y, y') = (0, 1, 0, 1)$, i.e., for Case (vi):

(vi.1.1) $|\varphi(e_{t+1})| - |\varphi(e_t)| = -(\alpha_D + \alpha_Y)\big(f_t(0, e_t) + f_t(1, e_t)\big) < 0$

$$\text{if we have } \begin{cases} f_t(0, e_t) \in (0, 1], f_t(1, e_t) \in (0, \frac{1}{1+\alpha_D+\alpha_Y}) \\ f_t(1, e_t) \leq \tan\big(\arctan\frac{1}{\alpha_D+\alpha_Y} - \frac{\pi}{4}\big) \end{cases} \quad ;$$

(vi.1.2) $|\varphi(e_{t+1})| - |\varphi(e_t)| = (\alpha_D + \alpha_Y - 2)f_t(0, e_t) + (\alpha_D + \alpha_Y + 2)f_t(1, e_t) < 0$

$$\text{if we have } \begin{cases} f_t(0, e_t) \in (0, 1], f_t(1, e_t) \in (0, \frac{1}{1+\alpha_D+\alpha_Y}) \\ f_t(1, e_t) < \tan\big(\arctan\frac{2}{\alpha_D+\alpha_Y} - \frac{\pi}{4}\big) \\ f_t(1, e_t) \geq \tan\big(\arctan\frac{1}{\alpha_D+\alpha_Y} - \frac{\pi}{4}\big) \end{cases} \quad ;$$

(vi.1.3) $|\varphi(e_{t+1})| - |\varphi(e_t)| = (\alpha_D + \alpha_Y - 2)f_t(0, e_t) + (\alpha_D + \alpha_Y + 2)f_t(1, e_t) > 0$

$$\text{if we have } \begin{cases} f_t(0, e_t) \in (0, 1], f_t(1, e_t) \in (0, \frac{1}{1+\alpha_D+\alpha_Y}) \\ f_t(1, e_t) < f_t(0, e_t) \\ f_t(1, e_t) \geq \tan\big(\arctan\frac{2}{\alpha_D+\alpha_Y} - \frac{\pi}{4}\big) \end{cases} \quad ;$$

(vi.1.4) $|\varphi(e_{t+1})| - |\varphi(e_t)| = (\alpha_D + \alpha_Y)\big(f_t(0, e_t) + f_t(1, e_t)\big) > 0$

$$\text{if we have } \begin{cases} f_t(0, e_t) \in (0, 1], f_t(1, e_t) \in (0, \frac{1}{1+\alpha_D+\alpha_Y}) \\ f_t(1, e_t) \geq f_t(0, e_t) \end{cases} \quad ;$$

(vi.2.1) $|\varphi(e_{t+1})| - |\varphi(e_t)| = 1 - (2 - \alpha_D - \alpha_Y)f_t(0, e_t) + f_t(1, e_t) > 0$

$$\text{if we have } \begin{cases} f_t(0, e_t) \in (0, 1], f_t(1, e_t) \in [\frac{1}{1+\alpha_D+\alpha_Y}, 1] \\ f_t(1, e_t) < f_t(0, e_t) \end{cases} \quad ;$$

(vi.2.2) $|\varphi(e_{t+1})| - |\varphi(e_t)| = 1 + (\alpha_D + \alpha_Y)f_t(0, e_t) - f_t(1, e_t) > 0$

$$\text{if we have } \begin{cases} f_t(0, e_t) \in (0, 1], f_t(1, e_t) \in [\frac{1}{1+\alpha_D+\alpha_Y}, 1] \\ f_t(1, e_t) \geq f_t(0, e_t) \end{cases} \quad .$$

When $(d, d', y, y') = (1, 0, 1, 0)$, i.e., for Case (xi):

(xi.1.1) $|\varphi(e_{t+1})| - |\varphi(e_t)| = -(\alpha_D + \alpha_Y)\big(f_t(0, e_t) + f_t(1, e_t)\big) < 0$

$$\text{if we have } \begin{cases} f_t(0, e_t) \in (0, \frac{1}{1+\alpha_D+\alpha_Y}), f_t(1, e_t) \in (0, 1] \\ f_t(1, e_t) \geq \tan\big(\frac{3\pi}{4} - \arctan\frac{1}{\alpha_D+\alpha_Y}\big) \end{cases} \quad ;$$

(xi.1.2) $|\varphi(e_{t+1})| - |\varphi(e_t)| = (\alpha_D + \alpha_Y + 2)f_t(0, e_t) + (\alpha_D + \alpha_Y - 2)f_t(1, e_t) < 0$

$$\text{if we have } \begin{cases} f_t(0, e_t) \in (0, \frac{1}{1+\alpha_D+\alpha_Y}), f_t(1, e_t) \in (0, 1] \\ f_t(1, e_t) < \tan\big(\frac{3\pi}{4} - \arctan\frac{1}{\alpha_D+\alpha_Y}\big) \\ f_t(1, e_t) \geq \tan\big(\frac{3\pi}{4} - \arctan\frac{2}{\alpha_D+\alpha_Y}\big) \end{cases} \quad ;$$

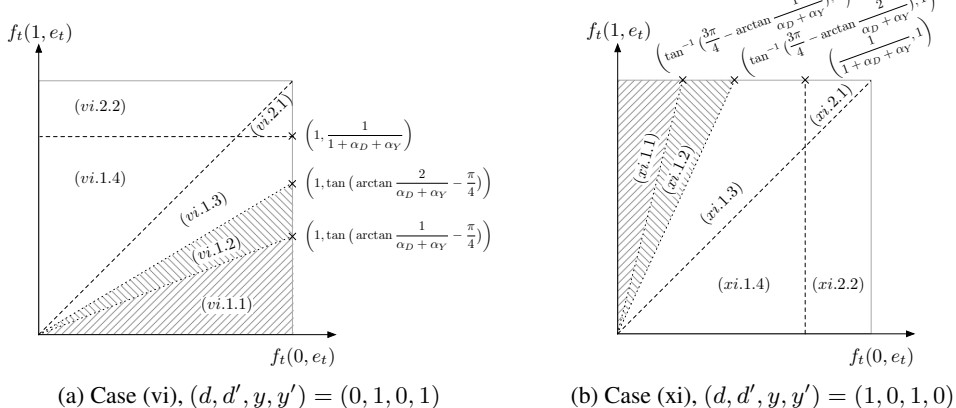

(a) Case (vi), $(d, d', y, y') = (0, 1, 0, 1)$        (b) Case (xi), $(d, d', y, y') = (1, 0, 1, 0)$

Figure 7: Illustration of the sliced squares on the $\big(f_t(0, e_t), f_t(1, e_t)\big)$ plane. Depending on the initial situation, i.e., the slice that the $\big(f_t(0, e_t), f_t(1, e_t)\big)$ pair falls upon, the term $|\varphi(e_{t+1})| - |\varphi(e_t)|$ takes different values. The shaded slices indicate that if the initial situation satisfies the corresponding condition, the calculated $|\varphi(e_{t+1})| - |\varphi(e_t)| < 0$.

(xi.1.3) $|\varphi(e_{t+1})| - |\varphi(e_t)| = (\alpha_D + \alpha_Y + 2)f_t(0, e_t) + (\alpha_D + \alpha_Y - 2)f_t(1, e_t) > 0$

$$\text{if we have } \begin{cases} f_t(0, e_t) \in (0, \frac{1}{1+\alpha_D+\alpha_Y}), f_t(1, e_t) \in (0, 1] \\ f_t(1, e_t) < \tan\left(\frac{3\pi}{4} - \arctan\frac{2}{\alpha_D+\alpha_Y}\right) \\ f_t(1, e_t) \geq f_t(0, e_t) \end{cases} ;$$

(xi.1.4) $|\varphi(e_{t+1})| - |\varphi(e_t)| = (\alpha_D + \alpha_Y)\big(f_t(0, e_t) + f_t(1, e_t)\big) > 0$

$$\text{if we have } \begin{cases} f_t(0, e_t) \in (0, \frac{1}{1+\alpha_D+\alpha_Y}), f_t(1, e_t) \in (0, 1] \\ f_t(1, e_t) < f_t(0, e_t) \end{cases} ;$$

(xi.2.1) $|\varphi(e_{t+1})| - |\varphi(e_t)| = 1 + f_t(0, e_t) - (2 - \alpha_D - \alpha_Y)f_t(1, e_t) > 0$

$$\text{if we have } \begin{cases} f_t(0, e_t) \in [\frac{1}{1+\alpha_D+\alpha_Y}, 1], f_t(1, e_t) \in (0, 1] \\ f_t(1, e_t) \geq f_t(0, e_t) \end{cases} ;$$

(xi.2.2) $|\varphi(e_{t+1})| - |\varphi(e_t)| = 1 - f_t(0, e_t) + (\alpha_D + \alpha_Y)f_t(1, e_t) > 0$

$$\text{if we have } \begin{cases} f_t(0, e_t) \in [\frac{1}{1+\alpha_D+\alpha_Y}, 1], f_t(1, e_t) \in (0, 1] \\ f_t(1, e_t) < f_t(0, e_t) \end{cases} .$$

When $(d, d', y, y') = (0, 0, 0, 0)$, i.e., for Case (i), or $(d, d', y, y') = (1, 1, 1, 1)$, i.e., for Case (xvi), $|\varphi(e_{t+1})| - |\varphi(e_t)| = 0$.

Now we proceed to the second step and aggregate $|\varphi(e_{t+1})| - |\varphi(e_t)|$ terms. According to Equation 5 and Equation B.5, for the perfect predictor we have:

$$\begin{aligned}
\Delta_{\text{STIR}}^{\text{(Perfect Predictor)}}\Big|_t^{t+1} = {} & P_t(0, 1, 0, 1) \cdot \int_{\epsilon \in \mathcal{E}} \int_{\xi \in \mathcal{E}} \big(|\varphi_{t+1}(\xi)| - |\varphi_t(\epsilon)|\big) \\
& \cdot \mathbb{1}\{\varphi_{t+1}(\xi) = G(f_t, g_t^D, g_t^{Y^{(\text{ori})}}; 0, 1, 0, 1, \epsilon, \alpha_D, \alpha_Y)\} \\
& \cdot q_t\big(f_t(0, \epsilon), f_t(1, \epsilon) \mid 0, 1, 0, 1\big) d\xi d\epsilon \\
& + P_t(1, 0, 1, 0) \cdot \int_{\epsilon \in \mathcal{E}} \int_{\xi \in \mathcal{E}} \big(|\varphi_{t+1}(\xi)| - |\varphi_t(\epsilon)|\big) \\
& \cdot \mathbb{1}\{\varphi_{t+1}(\xi) = G(f_t, g_t^D, g_t^{Y^{(\text{ori})}}; 1, 0, 1, 0, \epsilon, \alpha_D, \alpha_Y)\} \\
& \cdot q_t\big(f_t(0, \epsilon), f_t(1, \epsilon) \mid 1, 0, 1, 0\big) d\xi d\epsilon.
\end{aligned} \tag{C.2}$$

As we can see from Equation C.2, we need to perform two-dimensional integrations on the $\big(f_t(0, e_t), f_t(1, e_t)\big)$ plane, calculating the expectation of the term $|\varphi(e_{t+1})| - |\varphi(e_t)|$ over the conditional densities $q_t\big(f_t(0, \epsilon), f_t(1, \epsilon) \mid 0, 1, 0, 1\big)$ and $q_t\big(f_t(0, \epsilon), f_t(1, \epsilon) \mid 1, 0, 1, 0\big)$. Since these conditional joint densities could be convoluted in general cases, the calculation of conditional expectations in Equation C.2 could be rather complicated. Therefore, we propose to take advantage of Assumption 3.6 to quantitatively simplify the calculation yet remain consistent with the rather mild qualitative assumption (Assumption 3.5), and derive a result that is numerically clear and informative. For the purpose of better illustrating the connection between (qualitative and quantitative) assumptions on $q_t\big(f_t(0, \epsilon), f_t(1, \epsilon) \mid d, d', y, y'\big)$ and the computation of $\Delta_{\mathrm{STIR}}|_t^{t+1}$, we also provide illustrative figures as shown in Figure 7.

With the help of Assumption 3.6, we convert the conditional expectations in Equation C.2 into calculations of multiple integrals on slices within a $1 \times 1$ square on the 2-D plane, where $\phi_0$ and $\phi_1$ axes correspond to the value of $f_t(0, E_t)$ and $f_t(1, E_t)$ respectively:

$$
\begin{aligned}
&\int_{\epsilon \in \mathcal{E}} \int_{\xi \in \mathcal{E}} \big(|\varphi_{t+1}(\xi)| - |\varphi_t(\epsilon)|\big) \cdot \mathbb{1}\{\varphi_{t+1}(\xi) = G(f_t, g_t^D, g_t^{Y^{(\mathrm{ori})}}; 0, 1, 0, 1, \epsilon, \alpha_D, \alpha_Y)\} \\
&\qquad \cdot q_t\big(f_t(0, \epsilon), f_t(1, \epsilon) \mid 0, 1, 0, 1\big) d\xi d\epsilon \\
&= \gamma_{0101}^{(\mathrm{low})} \cdot \Bigg\{ \int_0^1 \int_0^{\tan\left(\arctan\frac{1}{\alpha_D+\alpha_Y} - \frac{\pi}{4}\right)\phi_0} -(\alpha_D + \alpha_Y)(\phi_0 + \phi_1)\, d\phi_1 d\phi_0 \\
&\qquad + \int_0^{\frac{1}{1+\alpha_D+\alpha_Y}} \int_{\tan\left(\arctan\frac{1}{\alpha_D+\alpha_Y} - \frac{\pi}{4}\right)\phi_0}^{\phi_0} (\alpha_D + \alpha_Y - 2)\phi_0 + (\alpha_D + \alpha_Y + 2)\phi_1\, d\phi_1 d\phi_0 \\
&\qquad + \int_{\frac{1}{1+\alpha_D+\alpha_Y}}^1 \int_{\tan\left(\arctan\frac{1}{\alpha_D+\alpha_Y} - \frac{\pi}{4}\right)\phi_0}^{\frac{1}{1+\alpha_D+\alpha_Y}} (\alpha_D + \alpha_Y - 2)\phi_0 + (\alpha_D + \alpha_Y + 2)\phi_1\, d\phi_1 d\phi_0 \\
&\qquad + \int_{\frac{1}{1+\alpha_D+\alpha_Y}}^1 \int_{\frac{1}{1+\alpha_D+\alpha_Y}}^{\phi_0} 1 - (2 - \alpha_D - \alpha_Y)\phi_0 + \phi_1\, d\phi_1 d\phi_0 \Bigg\} \\
&+ \gamma_{0101}^{(\mathrm{up})} \cdot \Bigg\{ \int_0^{\frac{1}{1+\alpha_D+\alpha_Y}} \int_{\phi_0}^{\frac{1}{1+\alpha_D+\alpha_Y}} (\alpha_D + \alpha_Y)(\phi_0 + \phi_1)\, d\phi_1 d\phi_0 \\
&\qquad + \int_0^{\frac{1}{1+\alpha_D+\alpha_Y}} \int_{\frac{1}{1+\alpha_D+\alpha_Y}}^1 1 + (\alpha_D + \alpha_Y)\phi_0 - \phi_1\, d\phi_1 d\phi_0 \\
&\qquad + \int_{\frac{1}{1+\alpha_D+\alpha_Y}}^1 \int_{\phi_0}^1 1 + (\alpha_D + \alpha_Y)\phi_0 - \phi_1\, d\phi_1 d\phi_0 \Bigg\},
\end{aligned}
$$

$$\int_{\epsilon \in \mathcal{E}} \int_{\xi \in \mathcal{E}} \left( |\varphi_{t+1}(\xi)| - |\varphi_t(\epsilon)| \right) \cdot \mathbb{1}\{\varphi_{t+1}(\xi) = G(f_t, g_t^D, g_t^{Y^{(\mathrm{ori})}}; 1, 0, 1, 0, \epsilon, \alpha_D, \alpha_Y)\}$$
$$\cdot\, q_t\big(f_t(0, \epsilon), f_t(1, \epsilon) \mid 1, 0, 1, 0\big) d\xi d\epsilon$$

$$= \gamma_{1010}^{(\mathrm{up})} \cdot \Bigg\{ \int_0^1 \int_0^{\tan^{-1}\left(\frac{3\pi}{4} - \arctan \frac{1}{\alpha_D + \alpha_Y}\right)\phi_1} -(\alpha_D + \alpha_Y)(\phi_0 + \phi_1) \, d\phi_0 d\phi_1$$

$$+ \int_0^{\frac{1}{1+\alpha_D+\alpha_Y}} \int_{\tan^{-1}\left(\frac{3\pi}{4} - \arctan \frac{1}{\alpha_D + \alpha_Y}\right)\phi_1}^{\phi_1} (\alpha_D + \alpha_Y + 2)\phi_0 + (\alpha_D + \alpha_Y - 2)\phi_1 \, d\phi_0 d\phi_1$$

$$+ \int_{\frac{1}{1+\alpha_D+\alpha_Y}}^1 \int_{\tan^{-1}\left(\frac{3\pi}{4} - \arctan \frac{1}{\alpha_D + \alpha_Y}\right)\phi_1}^{\frac{1}{1+\alpha_D+\alpha_Y}} (\alpha_D + \alpha_Y + 2)\phi_0 + (\alpha_D + \alpha_Y - 2)\phi_1 \, d\phi_0 d\phi_1$$

$$+ \int_{\frac{1}{1+\alpha_D+\alpha_Y}}^1 \int_{\frac{1}{1+\alpha_D+\alpha_Y}}^{\phi_1} 1 + \phi_0 - (2 - \alpha_D - \alpha_Y)\phi_1 \, d\phi_0 d\phi_1 \Bigg\}$$

$$+ \gamma_{1010}^{(\mathrm{low})} \cdot \Bigg\{ \int_0^{\frac{1}{1+\alpha_D+\alpha_Y}} \int_{\phi_1}^{\frac{1}{1+\alpha_D+\alpha_Y}} (\alpha_D + \alpha_Y)(\phi_0 + \phi_1) \, d\phi_0 d\phi_1$$

$$+ \int_0^{\frac{1}{1+\alpha_D+\alpha_Y}} \int_{\frac{1}{1+\alpha_D+\alpha_Y}}^1 1 - \phi_0 + (\alpha_D + \alpha_Y)\phi_1 \, d\phi_0 d\phi_1$$

$$+ \int_{\frac{1}{1+\alpha_D+\alpha_Y}}^1 \int_{\phi_1}^1 1 - \phi_0 + (\alpha_D + \alpha_Y)\phi_1 \, d\phi_0 d\phi_1 \Bigg\}.$$

Since $\gamma_{0101}^{(\mathrm{low})} + \gamma_{0101}^{(\mathrm{up})} = 2$ and $\gamma_{1010}^{(\mathrm{low})} + \gamma_{1010}^{(\mathrm{up})} = 2$, we can derive the form of $\Delta_{\mathrm{STIR}}^{(\text{Perfect Predictor})}\big|_t^{t+1}$:

$$\Delta_{\mathrm{STIR}}^{(\text{Perfect Predictor})}\big|_t^{t+1} = \big( P_t(0, 1, 0, 1) \cdot \gamma_{0101}^{(\mathrm{low})} + P_t(1, 0, 1, 0) \cdot \gamma_{1010}^{(\mathrm{up})} \big) \cdot \Bigg\{$$

$$-\frac{1 + \alpha_D + \alpha_Y}{3} \cdot \tan^2 \big( \arctan \frac{1}{\alpha_D + \alpha_Y} - \frac{\pi}{4} \big)$$

$$+ \frac{2(1 - \alpha_D - \alpha_Y)}{3} \cdot \tan \big( \arctan \frac{1}{\alpha_D + \alpha_Y} - \frac{\pi}{4} \big)$$

$$- \frac{1 - \alpha_D - \alpha_Y}{6} + \frac{3 - \alpha_D - \alpha_Y}{2(1 + \alpha_D + \alpha_Y)}$$

$$+ \frac{3(\alpha_D + \alpha_Y)^3 - 6(\alpha_D + \alpha_Y)^2 - 19(\alpha_D + \alpha_Y) - 10}{6(1 + \alpha_D + \alpha_Y)^3} \Bigg\}$$

$$+ \big( P_t(0, 1, 0, 1) + P_t(1, 0, 1, 0) \big) \cdot \Bigg[$$

$$\frac{\alpha_D + \alpha_Y - 2}{3} + \frac{\alpha_D + \alpha_Y}{1 + \alpha_D + \alpha_Y} + \frac{3(\alpha_D + \alpha_Y)^2 + 5(\alpha_D + \alpha_Y) + 2}{3(1 + \alpha_D + \alpha_Y)^3} \Bigg],$$

where $\gamma_{0101}^{(\mathrm{low})}, \gamma_{1010}^{(\mathrm{up})} \in (0, 1)$ (according to Assumption 3.6), and $\alpha_D, \alpha_Y \in [0, \frac{1}{2})$ (according to Assumption 3.3).

Let us denote $\beta(\alpha_D, \alpha_Y) := \tan \big( \arctan \frac{1}{\alpha_D + \alpha_Y} - \frac{\pi}{4} \big)$ to simplify the notation. Without loss of generality let us assume that $P_t(0, 1, 0, 1) \cdot \gamma_{0101}^{(\mathrm{low})} + P_t(1, 0, 1, 0) \cdot \gamma_{1010}^{(\mathrm{up})} > 0$.

We can further compute the partial derivatives and find out that:

$$
\frac{\partial \Delta_{\text{STIR}}^{\text{(Perfect Predictor)}} \big|_t^{t+1}}{\partial \big( P_t(0,1,0,1) \cdot \gamma_{0101}^{\text{(low)}} + P_t(1,0,1,0) \cdot \gamma_{1010}^{\text{(up)}} \big)}
$$

$$
= -\frac{1 + \alpha_D + \alpha_Y}{3} \cdot \beta^2(\alpha_D, \alpha_Y)
$$

$$
+ \frac{2(1 - \alpha_D - \alpha_Y)}{3} \cdot \beta(\alpha_D, \alpha_Y)
$$

$$
- \frac{1 - \alpha_D - \alpha_Y}{6} + \frac{3 - \alpha_D - \alpha_Y}{2(1 + \alpha_D + \alpha_Y)}
$$

$$
+ \frac{3(\alpha_D + \alpha_Y)^3 - 6(\alpha_D + \alpha_Y)^2 - 19(\alpha_D + \alpha_Y) - 10}{6(1 + \alpha_D + \alpha_Y)^3}
$$

$$
< 0, \ \forall \, \alpha_D, \alpha_Y \in [0, \tfrac{1}{2}),
$$

and that:

$$
\frac{\partial \Delta_{\text{STIR}}^{\text{(Perfect Predictor)}} \big|_t^{t+1}}{\partial (\alpha_D + \alpha_Y)} = \big( P_t(0,1,0,1) + P_t(1,0,1,0) \big) \cdot \left[ \frac{1}{3} + \frac{2}{3(1 + \alpha_D + \alpha_Y)^3} \right]
$$

$$
+ \big( P_t(0,1,0,1) \cdot \gamma_{0101}^{\text{(low)}} + P_t(1,0,1,0) \cdot \gamma_{1010}^{\text{(up)}} \big) \cdot \Bigg[
$$

$$
- \frac{2(1 + \alpha_D + \alpha_Y)}{3} \cdot \beta(\alpha_D, \alpha_Y) \cdot \frac{\partial \beta(\alpha_D, \alpha_Y)}{\partial(\alpha_D + \alpha_Y)}
$$

$$
+ \frac{2(1 - \alpha_D - \alpha_Y)}{3} \cdot \frac{\partial \beta(\alpha_D, \alpha_Y)}{\partial(\alpha_D + \alpha_Y)}
$$

$$
+ \frac{1}{6} - \frac{2}{(1 + \alpha_D + \alpha_Y)^2} + \frac{15(\alpha_D + \alpha_Y) + 11}{6(1 + \alpha_D + \alpha_Y)^3} \Bigg]
$$

$$
= \big( P_t(0,1,0,1) + P_t(1,0,1,0) \big) \cdot \left[ \frac{1}{3} + \frac{2}{3(1 + \alpha_D + \alpha_Y)^3} \right]
$$

$$
+ \big( P_t(0,1,0,1) \cdot \gamma_{0101}^{\text{(low)}} + P_t(1,0,1,0) \cdot \gamma_{1010}^{\text{(up)}} \big) \cdot \Bigg[
$$

$$
\frac{2\big(1 + \beta^2(\alpha_D, \alpha_Y)\big) \cdot \big[ (1 + \alpha_D + \alpha_Y)\beta(\alpha_D, \alpha_Y) + \alpha_D + \alpha_Y - 1 \big]}{3(1 + \alpha_D + \alpha_Y)^3}
$$

$$
+ \frac{(\alpha_D + \alpha_Y)^3 + 3(\alpha_D + \alpha_Y)^2 + 6(\alpha_D + \alpha_Y)}{6(1 + \alpha_D + \alpha_Y)^3} \Bigg]
$$

$$
> 0, \ \forall \, \gamma_{0101}^{\text{(low)}}, \gamma_{1010}^{\text{(up)}} \in (0,1), \alpha_D, \alpha_Y \in [0, \tfrac{1}{2}),
$$

where we utilize the fact that $\frac{\partial \beta(\alpha_D, \alpha_Y)}{\partial(\alpha_D + \alpha_Y)} = \big(1 + \beta^2(\alpha_D, \alpha_Y)\big) \cdot \frac{1}{1 + (\alpha_D + \alpha_Y)^2}$.

Therefore, we can conclude that

$$
\Delta_{\text{STIR}}^{(\text{Perfect Predictor})}\Big|_t^{t+1} > \lim_{\substack{\gamma_{0101}^{(\text{low})} \to 1 \\ \gamma_{1010}^{(\text{up})} \to 1 \\ \alpha_D + \alpha_Y \to 0}} \Big( P_t(0,1,0,1) \cdot \gamma_{0101}^{(\text{low})} + P_t(1,0,1,0) \cdot \gamma_{1010}^{(\text{up})} \Big) \cdot \Bigg\{
$$

$$
- \frac{1 + \alpha_D + \alpha_Y}{3} \cdot \tan^2 \Big( \arctan \frac{1}{\alpha_D + \alpha_Y} - \frac{\pi}{4} \Big)
$$

$$
+ \frac{2(1 - \alpha_D - \alpha_Y)}{3} \cdot \tan \Big( \arctan \frac{1}{\alpha_D + \alpha_Y} - \frac{\pi}{4} \Big)
$$

$$
- \frac{1 - \alpha_D - \alpha_Y}{6} + \frac{3 - \alpha_D - \alpha_Y}{2(1 + \alpha_D + \alpha_Y)}
$$

$$
+ \frac{3(\alpha_D + \alpha_Y)^3 - 6(\alpha_D + \alpha_Y)^2 - 19(\alpha_D + \alpha_Y) - 10}{6(1 + \alpha_D + \alpha_Y)^3} \Bigg\}
$$

$$
+ \big( P_t(0,1,0,1) + P_t(1,0,1,0) \big) \cdot \Bigg[
$$

$$
\frac{\alpha_D + \alpha_Y - 2}{3} + \frac{\alpha_D + \alpha_Y}{1 + \alpha_D + \alpha_Y} + \frac{3(\alpha_D + \alpha_Y)^2 + 5(\alpha_D + \alpha_Y) + 2}{3(1 + \alpha_D + \alpha_Y)^3} \Bigg]
$$

$$
= 0,
$$

i.e., under the specified assumptions and dynamics, we have

$$
\forall \gamma_{0101}^{(\text{low})}, \gamma_{1010}^{(\text{up})} \in (0,1), \alpha_D, \alpha_Y \in [0, \frac{1}{2}) : \Delta_{\text{STIR}}^{(\text{Perfect Predictor})}\Big|_t^{t+1} > 0. \tag{C.3}
$$

$\square$

### C.4 PROOF FOR THEOREM 4.3

**Theorem.** *Let us consider the general situation where both $D_t$ and $Y_t^{(ori)}$ are dependent with $A_t$, i.e., $D_t \not\perp\!\!\!\perp A_t, Y_t^{(ori)} \not\perp\!\!\!\perp A_t$. Let us further assume that the data dynamics satisfies $\alpha_D \in (0, \frac{1}{2}), \alpha_Y = 0$. Then under Fact 3.2, Assumption 3.3, and Assumption 3.4, and Assumption 3.6, as well as the specified dynamics, when $H_t \not\perp\!\!\!\perp A_t$, it is possible for the Counterfactual Fair predictor to get closer to the long-term fairness goal after one-step intervention, if certain properties of the data dynamics and the predictor behavior are satisfied simultaneously, i.e.,*

$$
\begin{cases}
g_t^D(0, E_t) = g_t^D(1, E_t) \\
\frac{P_t(1,1,0,1) + P_t(1,1,1,0)}{P_t(0,0,0,1) + P_t(0,0,1,0)} < \frac{27}{8} \\
\alpha_D \in \left( \left( \frac{P_t(1,1,0,1) + P_t(1,1,1,0)}{P_t(0,0,0,1) + P_t(0,0,1,0)} \right)^{\frac{1}{3}} - 1, \frac{1}{2} \right) \\
\alpha_Y = 0
\end{cases}
$$

$$
\implies \Delta_{\text{STIR}}^{(\text{Counterfactual Fair})}\Big|_t^{t+1} < 0.
$$

*Proof (sketch).* Similar to proving Theorem 4.2 (proof in Appendix C.3), the goal is to calculate if it is possible for the *Single-step Tier Imbalance Reduction* $\Delta_{\text{STIR}}|_t^{t+1}$ to be smaller than 0 when using Counterfactual Fair predictors.

Since $\Delta_{\text{STIR}}|_t^{t+1}$ is a weighted aggregation of $|\varphi(e_{t+1})| - |\varphi(e_t)|$ (as defined in Equation 5), the quantitative analysis involves three key components: instantiations of $\varphi_{t+1}(e_{t+1})$, the knowledge/assumptions on $q_t\big(f_t(0, \epsilon), f_t(1, \epsilon) \mid d, d', y, y'\big)$, and characteristics of $P_t(d, d', y', y')$.

For the first component, since $\alpha_Y = 0$ is a special case of scenarios where $\alpha_D > \alpha_Y$, we can list all possible instantiations of $\varphi_{t+1}(e_{t+1})$ in Table 4 (when $\alpha_D > \alpha_Y$). For the second component, we can introduce a quantitative assumption on $q_t\big(f_t(0, \epsilon), f_t(1, \epsilon) \mid d, d', y, y'\big)$ (Assumption 3.6). For the third component, we need to exploit the characteristic of the predictor of interest to gain further insight into the joint distribution $P_t(d, d', y, y')$. For Counterfactual Fair predictors, we have $P_t(d, d', y, y')$ satisfies Equation 10 (as we have discussed in Section 4.2.2).

For the purpose of calculating the value of $\Delta_{\mathrm{STIR}}|_t^{t+1}$, the proof contains two steps: (1) exhaustively derive the value of $|\varphi(e_{t+1})| - |\varphi(e_t)|$ after one-step dynamics (finished in Appendix C.3 when proving Theorem 4.2), and (2) aggregate the difference $|\varphi(e_{t+1})| - |\varphi(e_t)|$ with the help of the additional knowledge/assumptions on $q_t\big(f_t(0, \epsilon), f_t(1, \epsilon) \mid d, d', y, y'\big)$ and $P_t(d, d', y, y')$. $\qquad\square$

*Proof (full).* Based on the definition of $\Delta_{\mathrm{STIR}}|_t^{t+1}$, the proof calculates the aggregation (integration followed by summation) of the difference $|\varphi(e_{t+1})| - |\varphi(e_t)|$ with the help of the additional knowledge/assumptions on $q_t\big(f_t(0, \epsilon), f_t(1, \epsilon) \mid d, d', y, y'\big)$ and $P_t(d, d', y, y')$.

Since we assume $\alpha_D \in (0, \frac{1}{2}), \alpha_Y = 0$, we focus on possible instantiations of $\varphi_{t+1}(e_{t+1})$ as listed in Table 4 ($\alpha_D > \alpha_Y$). For the Counterfactual Fair predictor that satisfies $g_t^D(0, E_t) = g_t^D(1, E_t)$, not every case in Table 4 corresponds to a nonzero $P_t(d, d', y, y')$ and therefore may not contribute to the computation of $\Delta_{\mathrm{STIR}}|_t^{t+1}$ as detailed in Equation 5. By applying Equation 10 we need to consider Case (i), Case (ii), Case (iii), Case (iv), Case (xiii), Case (xiv), Case (xv), and Case (xvi) in Table 4.

When $(d, d', y, y')$ satisfies $y = y'$, i.e, for Case (i), Case (iv), Case (xiii), and Case (xvi), we have $|\varphi(e_{t+1})| - |\varphi(e_t)| = 0$. Therefore we only need to calculate $\Delta_{\mathrm{STIR}}^{(\text{Counterfactual Fair})}|_t^{t+1}$ for Case (ii), Case (iii), Case (xiv), and Case (xv) (although $\alpha_Y = 0$, we explicitly keep the hyperparameter $\alpha_Y$ in the proof for the purpose of notation consistency).

According to Equation 5 and Equation B.5, for the Counterfactual Fair predictor we have:

$$
\begin{aligned}
\Delta_{\mathrm{STIR}}^{(\text{Counterfactual Fair})}\Big|_t^{t+1} = {} & P_t(0, 0, 0, 1) \cdot \int_{\epsilon \in \mathcal{E}} \int_{\xi \in \mathcal{E}} \big(|\varphi_{t+1}(\xi)| - |\varphi_t(\epsilon)|\big) \\
& \cdot \mathbb{1}\{\varphi_{t+1}(\xi) = G(f_t, g_t^D, g_t^{Y^{(\mathrm{ori})}}; 0, 0, 0, 1, \epsilon, \alpha_D, \alpha_Y)\} \\
& \cdot q_t\big(f_t(0, \epsilon), f_t(1, \epsilon) \mid 0, 0, 0, 1\big) d\xi d\epsilon \\
& + P_t(0, 0, 1, 0) \cdot \int_{\epsilon \in \mathcal{E}} \int_{\xi \in \mathcal{E}} \big(|\varphi_{t+1}(\xi)| - |\varphi_t(\epsilon)|\big) \\
& \cdot \mathbb{1}\{\varphi_{t+1}(\xi) = G(f_t, g_t^D, g_t^{Y^{(\mathrm{ori})}}; 0, 0, 1, 0, \epsilon, \alpha_D, \alpha_Y)\} \\
& \cdot q_t\big(f_t(0, \epsilon), f_t(1, \epsilon) \mid 0, 0, 1, 0\big) d\xi d\epsilon \\
& + P_t(1, 1, 0, 1) \cdot \int_{\epsilon \in \mathcal{E}} \int_{\xi \in \mathcal{E}} \big(|\varphi_{t+1}(\xi)| - |\varphi_t(\epsilon)|\big) \\
& \cdot \mathbb{1}\{\varphi_{t+1}(\xi) = G(f_t, g_t^D, g_t^{Y^{(\mathrm{ori})}}; 1, 1, 0, 1, \epsilon, \alpha_D, \alpha_Y)\} \\
& \cdot q_t\big(f_t(0, \epsilon), f_t(1, \epsilon) \mid 1, 1, 0, 1\big) d\xi d\epsilon \\
& + P_t(1, 1, 1, 0) \cdot \int_{\epsilon \in \mathcal{E}} \int_{\xi \in \mathcal{E}} \big(|\varphi_{t+1}(\xi)| - |\varphi_t(\epsilon)|\big) \\
& \cdot \mathbb{1}\{\varphi_{t+1}(\xi) = G(f_t, g_t^D, g_t^{Y^{(\mathrm{ori})}}; 1, 1, 1, 0, \epsilon, \alpha_D, \alpha_Y)\} \\
& \cdot q_t\big(f_t(0, \epsilon), f_t(1, \epsilon) \mid 1, 1, 1, 0\big) d\xi d\epsilon.
\end{aligned}
\tag{C.4}
$$

Similar to the proof of the result for perfect predictors presented in Appendix C.3, with the help of Assumption 3.6, we convert the conditional expectations in Equation C.4 into calculations of multiple integrals on slices within a $1 \times 1$ square on the 2-D plane, where $\phi_0$ and $\phi_1$ axes correspond to the value of $f_t(0, E_t)$ and $f_t(1, E_t)$ respectively:

$$\int_{\epsilon \in \mathcal{E}} \int_{\xi \in \mathcal{E}} \left(|\varphi_{t+1}(\xi)| - |\varphi_t(\epsilon)|\right) \cdot \mathbb{1}\{\varphi_{t+1}(\xi) = G(f_t, g_t^D, g_t^{Y^{(\text{ori})}}; 0, 0, 0, 1, \epsilon, \alpha_D, \alpha_Y)\}$$
$$\cdot q_t\big(f_t(0, \epsilon), f_t(1, \epsilon) \mid 0, 0, 0, 1\big) d\xi d\epsilon$$
$$= \gamma_{0001}^{(\text{low})} \cdot \left\{ \int_0^1 \int_0^{\frac{1-\alpha_D-\alpha_Y}{1-\alpha_D+\alpha_Y}\phi_0} -(\alpha_D + \alpha_Y)\phi_0 + (\alpha_D - \alpha_Y)\phi_1 \ d\phi_1 d\phi_0 \right.$$
$$\left. + \int_0^1 \int_{\frac{1-\alpha_D-\alpha_Y}{1-\alpha_D+\alpha_Y}\phi_0}^{\phi_0} -(2 - \alpha_D - \alpha_Y)\phi_0 + (2 - \alpha_D + \alpha_Y)\phi_1 \ d\phi_1 d\phi_0 \right\}$$
$$+ \gamma_{0001}^{(\text{up})} \cdot \int_0^1 \int_0^{\phi_1} (\alpha_D + \alpha_Y)\phi_0 - (\alpha_D - \alpha_Y)\phi_1 \ d\phi_0 d\phi_1,$$

$$\int_{\epsilon \in \mathcal{E}} \int_{\xi \in \mathcal{E}} \left(|\varphi_{t+1}(\xi)| - |\varphi_t(\epsilon)|\right) \cdot \mathbb{1}\{\varphi_{t+1}(\xi) = G(f_t, g_t^D, g_t^{Y^{(\text{ori})}}; 0, 0, 1, 0, \epsilon, \alpha_D, \alpha_Y)\}$$
$$\cdot q_t\big(f_t(0, \epsilon), f_t(1, \epsilon) \mid 0, 0, 1, 0\big) d\xi d\epsilon$$
$$= \gamma_{0010}^{(\text{up})} \cdot \left\{ \int_0^1 \int_0^{\frac{1-\alpha_D-\alpha_Y}{1-\alpha_D+\alpha_Y}\phi_1} (\alpha_D - \alpha_Y)\phi_0 - (\alpha_D + \alpha_Y)\phi_1 \ d\phi_0 d\phi_1 \right.$$
$$\left. + \int_0^1 \int_{\frac{1-\alpha_D-\alpha_Y}{1-\alpha_D+\alpha_Y}\phi_1}^{\phi_1} (2 - \alpha_D + \alpha_Y)\phi_0 - (2 - \alpha_D - \alpha_Y)\phi_1 \ d\phi_0 d\phi_1 \right\}$$
$$+ \gamma_{0010}^{(\text{low})} \cdot \int_0^1 \int_0^{\phi_0} -(\alpha_D - \alpha_Y)\phi_0 + (\alpha_D + \alpha_Y)\phi_1 \ d\phi_1 d\phi_0,$$

$$\int_{\epsilon \in \mathcal{E}} \int_{\xi \in \mathcal{E}} \left( |\varphi_{t+1}(\xi)| - |\varphi_t(\epsilon)| \right) \cdot \mathbb{1}\{\varphi_{t+1}(\xi) = G(f_t, g_t^D, g_t^{Y^{(\text{ori})}}; 1, 1, 0, 1, \epsilon, \alpha_D, \alpha_Y)\}$$

$$\cdot q_t\big(f_t(0, \epsilon), f_t(1, \epsilon) \mid 1, 1, 0, 1\big) d\xi d\epsilon$$

$$= \gamma_{1101}^{(\text{low})} \cdot \left\{ \int_0^{\frac{1}{1+\alpha_D+\alpha_Y}} \int_{\phi_1}^{\frac{1+\alpha_D+\alpha_Y}{1+\alpha_D-\alpha_Y}\phi_1} -(2 + \alpha_D - \alpha_Y)\phi_0 + (2 + \alpha_D + \alpha_Y)\phi_1 \, d\phi_0 d\phi_1 \right.$$

$$+ \int_0^{\frac{1}{1+\alpha_D-\alpha_Y}} \int_0^{\frac{1+\alpha_D-\alpha_Y}{1+\alpha_D+\alpha_Y}\phi_0} (\alpha_D - \alpha_Y)\phi_0 - (\alpha_D + \alpha_Y)\phi_1 \, d\phi_1 d\phi_0$$

$$+ \left. \int_{\frac{1}{1+\alpha_D+\alpha_Y}}^{\frac{1}{1+\alpha_D-\alpha_Y}} \int_{\frac{1}{1+\alpha_D+\alpha_Y}}^{\phi_0} 1 - (2 + \alpha_D - \alpha_Y)\phi_0 + \phi_1 \, d\phi_1 d\phi_0 \right\}$$

$$+ \gamma_{1101}^{(\text{up})} \cdot \left\{ \int_0^{\frac{1}{1+\alpha_D+\alpha_Y}} \int_0^{\phi_1} -(\alpha_D - \alpha_Y)\phi_0 + (\alpha_D + \alpha_Y)\phi_1 \, d\phi_0 d\phi_1 \right.$$

$$+ \int_0^{\frac{1}{1+\alpha_D+\alpha_Y}} \int_{\frac{1}{1+\alpha_D+\alpha_Y}}^1 1 - (\alpha_D - \alpha_Y)\phi_0 - \phi_1 \, d\phi_1 d\phi_0$$

$$+ \left. \int_{\frac{1}{1+\alpha_D+\alpha_Y}}^{\frac{1}{1+\alpha_D-\alpha_Y}} \int_{\phi_0}^1 1 - (\alpha_D - \alpha_Y)\phi_0 - \phi_1 \, d\phi_1 d\phi_0 \right\},$$

$$\int_{\epsilon \in \mathcal{E}} \int_{\xi \in \mathcal{E}} \left( |\varphi_{t+1}(\xi)| - |\varphi_t(\epsilon)| \right) \cdot \mathbb{1}\{\varphi_{t+1}(\xi) = G(f_t, g_t^D, g_t^{Y^{(\text{ori})}}; 1, 1, 1, 0, \epsilon, \alpha_D, \alpha_Y)\}$$

$$\cdot q_t\big(f_t(0, \epsilon), f_t(1, \epsilon) \mid 1, 1, 1, 0\big) d\xi d\epsilon$$

$$= \gamma_{1110}^{(\text{up})} \cdot \left\{ \int_0^{\frac{1}{1+\alpha_D+\alpha_Y}} \int_{\phi_0}^{\frac{1+\alpha_D+\alpha_Y}{1+\alpha_D-\alpha_Y}\phi_0} (2 + \alpha_D + \alpha_Y)\phi_0 - (2 + \alpha_D - \alpha_Y)\phi_1 \, d\phi_1 d\phi_0 \right.$$

$$+ \int_0^{\frac{1}{1+\alpha_D-\alpha_Y}} \int_0^{\frac{1+\alpha_D-\alpha_Y}{1+\alpha_D+\alpha_Y}\phi_1} -(\alpha_D + \alpha_Y)\phi_0 + (\alpha_D - \alpha_Y)\phi_1 \, d\phi_0 d\phi_1$$

$$+ \left. \int_{\frac{1}{1+\alpha_D-\alpha_Y}}^1 \int_0^{\frac{1}{1+\alpha_D+\alpha_Y}} 1 - \phi_0 - (\alpha_D + \alpha_Y)\phi_1 \, d\phi_1 d\phi_0 \right\}$$

$$+ \gamma_{1110}^{(\text{low})} \cdot \int_0^{\frac{1}{1+\alpha_D+\alpha_Y}} \int_0^{\phi_0} (\alpha_D + \alpha_Y)\phi_0 - (\alpha_D - \alpha_Y)\phi_1 \, d\phi_1 d\phi_0.$$

Since $\gamma_{0001}^{(\text{low})} + \gamma_{0001}^{(\text{up})} = 2$, $\gamma_{0010}^{(\text{low})} + \gamma_{0010}^{(\text{up})} = 2$, $\gamma_{1101}^{(\text{low})} + \gamma_{1101}^{(\text{up})} = 2$, and $\gamma_{1110}^{(\text{low})} + \gamma_{1110}^{(\text{up})} = 2$, we can derive the form of the term $\Delta_{\text{STIR}}^{(\text{Counterfactual Fair})}\big|_t^{t+1}$:

$$
\Delta_{\text{STIR}}^{(\text{Counterfactual Fair})}\Big|_t^{t+1} = \left(P_t(0,0,0,1) \cdot \gamma_{0001}^{(\text{low})} + P_t(0,0,1,0) \cdot \gamma_{0010}^{(\text{up})}\right) \cdot \Bigg\{
$$

$$
\frac{(\alpha_D - \alpha_Y)(1 - \alpha_D - \alpha_Y)^2}{6(1 - \alpha_D + \alpha_Y)^2} - \frac{(\alpha_D + \alpha_Y)(1 - \alpha_D - \alpha_Y)}{3(1 - \alpha_D + \alpha_Y)}
$$

$$
+ \frac{2\alpha_Y}{3(1 - \alpha_D + \alpha_Y)}\left[-(2 - \alpha_D - \alpha_Y) + \frac{(2 - \alpha_D + \alpha_Y)(1 - \alpha_D)}{1 - \alpha_D + \alpha_Y}\right] + \frac{\alpha_D}{6} - \frac{\alpha_Y}{2}\Bigg\}
$$

$$
+ P_t(1,1,0,1) \cdot \gamma_{1101}^{(\text{low})} \cdot \Bigg\{
$$

$$
\frac{2\alpha_Y}{3(1 + \alpha_D - \alpha_Y)(1 + \alpha_D + \alpha_Y)^3}\left[(2 + \alpha_D + \alpha_Y) - \frac{(2 + \alpha_D - \alpha_Y)(1 + \alpha_D)}{1 + \alpha_D - \alpha_Y}\right]
$$

$$
+ \frac{1}{3(1 + \alpha_D + \alpha_Y)(1 + \alpha_D - \alpha_Y)^3}\Big[
$$

$$
(\alpha_D - \alpha_Y)(1 + \alpha_D - \alpha_Y) - \frac{(\alpha_D + \alpha_Y)(1 + \alpha_D - \alpha_Y)^2}{2(1 + \alpha_D + \alpha_Y)}\Big]
$$

$$
- \frac{1}{(1 + \alpha_D + \alpha_Y)^3}\left(\frac{\alpha_D}{6} + \frac{\alpha_Y}{2}\right)
$$

$$
- \frac{2}{3}(1 + \alpha_D - \alpha_Y) \cdot \left[\frac{1}{(1 + \alpha_D - \alpha_Y)^3} - \frac{1}{(1 + \alpha_D + \alpha_Y)^3}\right]
$$

$$
+ \left[\frac{3 + 2\alpha_D}{2(1 + \alpha_D + \alpha_Y)} + \frac{1 + \alpha_D - \alpha_Y}{2}\right] \cdot \left[\frac{1}{(1 + \alpha_D - \alpha_Y)^2} - \frac{1}{(1 + \alpha_D + \alpha_Y)^2}\right]
$$

$$
- \left[\frac{3 + 2\alpha_D + 2\alpha_Y}{2(1 + \alpha_D + \alpha_Y)^2} + \frac{1}{2}\right] \cdot \left[\frac{1}{1 + \alpha_D - \alpha_Y} - \frac{1}{1 + \alpha_D + \alpha_Y}\right] - \frac{(\alpha_D + \alpha_Y)\alpha_Y}{(1 + \alpha_D + \alpha_Y)^3}\Bigg\}
$$

$$
+ P_t(1,1,1,0) \cdot \gamma_{1110}^{(\text{up})} \cdot \Bigg\{
$$

$$
\frac{2\alpha_Y}{3(1 + \alpha_D - \alpha_Y)(1 + \alpha_D + \alpha_Y)^3}\left[(2 + \alpha_D + \alpha_Y) - \frac{(2 + \alpha_D - \alpha_Y)(1 + \alpha_D)}{1 + \alpha_D - \alpha_Y}\right]
$$

$$
+ \frac{1}{3(1 + \alpha_D + \alpha_Y)(1 + \alpha_D - \alpha_Y)^3}\Big[
$$

$$
(\alpha_D - \alpha_Y)(1 + \alpha_D - \alpha_Y) - \frac{(\alpha_D + \alpha_Y)(1 + \alpha_D - \alpha_Y)^2}{2(1 + \alpha_D + \alpha_Y)}\Big]
$$

$$
+ \frac{\alpha_D - \alpha_Y}{(1 + \alpha_D + \alpha_Y)(1 + \alpha_D - \alpha_Y)} - \frac{(\alpha_D + \alpha_Y)(\alpha_D - \alpha_Y)}{2(1 + \alpha_D + \alpha_Y)^2(1 + \alpha_D - \alpha_Y)}
$$

$$
- \frac{1}{2(1 + \alpha_D + \alpha_Y)}\left[1 - \frac{1}{(1 + \alpha_D - \alpha_Y)^2}\right] - \left(\frac{\alpha_D}{6} + \frac{\alpha_Y}{2}\right)\frac{1}{(1 + \alpha_D + \alpha_Y)^3}\Bigg\}
$$

$$
+ \left(P_t(0,0,0,1) + P_t(0,0,1,0)\right) \cdot \left(-\frac{\alpha_D}{3} + \alpha_Y\right)
$$

$$
+ P_t(1,1,0,1) \cdot \Bigg\{\frac{1}{(1 + \alpha_D + \alpha_Y)^3}\left(\frac{\alpha_D}{3} + \alpha_Y\right)
$$

$$
+ \frac{2(\alpha_D + \alpha_Y)\alpha_Y}{(1 + \alpha_D + \alpha_Y)^3} + \frac{1}{1 + \alpha_D - \alpha_Y} - \frac{1}{1 + \alpha_D + \alpha_Y}
$$

$$
+ \left(\frac{1}{3} + \frac{2\alpha_D}{3} - \frac{2\alpha_Y}{3}\right) \cdot \left[\frac{1}{(1 + \alpha_D - \alpha_Y)^3} - \frac{1}{(1 + \alpha_D + \alpha_Y)^3}\right]
$$

$$
- (1 + \alpha_D - \alpha_Y) \cdot \left[\frac{1}{(1 + \alpha_D - \alpha_Y)^2} - \frac{1}{(1 + \alpha_D + \alpha_Y)^2}\right]\Bigg\}
$$

$$
+ P_t(1,1,1,0) \cdot \left(\frac{\alpha_D}{3} + \alpha_Y\right)\frac{1}{(1 + \alpha_D + \alpha_Y)^3},
$$

where $\gamma_{0101}^{(\text{low})}, \gamma_{1010}^{(\text{up})} \in (0,1)$ (according to Assumption 3.6), and $\alpha_D, \alpha_Y \in [0, \frac{1}{2})$ (according to Assumption 3.3).

Now let us consider the data dynamics where $\alpha_Y = 0$ and simplify the form of $\Delta_{\text{STIR}}^{(\text{Counterfactual Fair})}\big|_t^{t+1}$:

$$
\begin{aligned}
&\Delta_{\text{STIR}}^{(\text{Counterfactual Fair})}\big|_t^{t+1} \\
&= -\frac{\alpha_D}{3} \cdot \big( P_t(0,0,0,1) + P_t(0,0,1,0) \big) + \frac{\alpha_D}{3(1+\alpha_D)^3} \cdot \big( P_t(1,1,0,1) + P_t(1,1,1,0) \big).
\end{aligned}
$$

As we can see, as long as we have $\frac{P_t(1,1,0,1)+P_t(1,1,1,0)}{P_t(0,0,0,1)+P_t(0,0,1,0)} < \frac{27}{8}$ and at the same time the parameter satisfies $\alpha_D \in \big( \big( \frac{P_t(1,1,0,1)+P_t(1,1,1,0)}{P_t(0,0,0,1)+P_t(0,0,1,0)} \big)^{\frac{1}{3}} - 1, \frac{1}{2} \big)$, it is possible for the counterfactual fair predictor to achieve a negative value for $\Delta_{\text{STIR}}\big|_t^{t+1}$ after a one-step intervention:

$$
\left\{
\begin{array}{l}
\frac{P_t(1,1,0,1)+P_t(1,1,1,0)}{P_t(0,0,0,1)+P_t(0,0,1,0)} < \frac{27}{8} \\
\alpha_D \in \big( \big( \frac{P_t(1,1,0,1)+P_t(1,1,1,0)}{P_t(0,0,0,1)+P_t(0,0,1,0)} \big)^{\frac{1}{3}} - 1, \frac{1}{2} \big) \\
\alpha_Y = 0
\end{array}
\right.
\implies \Delta_{\text{STIR}}^{(\text{Counterfactual Fair})}\big|_t^{t+1} < 0.
$$

$\square$

Table 1: When $\alpha_D > \alpha_Y$, compare cases of $H_{t+1}$ with different values of $A_t$.

| Case | $D_t$ | | $Y_t^{(ori)}$ | | $H_{t+1}$ | |
| --- | --- | --- | --- | --- | --- | --- |
| | if $A_t = 0$ | if $A_t = 1$ | if $A_t = 0$ | if $A_t = 1$ | if $A_t = 0$ | if $A_t = 1$ |
| (i) | 0 | 0 | 0 | 0 | $f_t(0, e_t)$ | $f_t(1, e_t)$ |
| (ii) | 0 | 0 | 0 | 1 | $f_t(0, e_t)(1 - \alpha_D - \alpha_Y)$ | $f_t(1, e_t)(1 - \alpha_D + \alpha_Y)$ |
| (iii) | 0 | 0 | 1 | 0 | $f_t(0, e_t)(1 - \alpha_D + \alpha_Y)$ | $f_t(1, e_t)(1 - \alpha_D - \alpha_Y)$ |
| (iv) | 0 | 0 | 1 | 1 | $f_t(0, e_t)$ | $f_t(1, e_t)$ |
| (v) | 0 | 1 | 0 | 0 | $f_t(0, e_t)(1 - \alpha_D - \alpha_Y)$ | $\min\{f_t(1, e_t)(1 + \alpha_D - \alpha_Y), 1\}$ |
| (vi) | 0 | 1 | 0 | 1 | $f_t(0, e_t)(1 - \alpha_D - \alpha_Y)$ | $\min\{f_t(1, e_t)(1 + \alpha_D + \alpha_Y), 1\}$ |
| (vii) | 0 | 1 | 1 | 0 | $f_t(0, e_t)(1 - \alpha_D + \alpha_Y)$ | $\min\{f_t(1, e_t)(1 + \alpha_D - \alpha_Y), 1\}$ |
| (viii) | 0 | 1 | 1 | 1 | $f_t(0, e_t)(1 - \alpha_D + \alpha_Y)$ | $\min\{f_t(1, e_t)(1 + \alpha_D + \alpha_Y), 1\}$ |
| (ix) | 1 | 0 | 0 | 0 | $\min\{f_t(0, e_t)(1 + \alpha_D - \alpha_Y), 1\}$ | $f_t(1, e_t)(1 - \alpha_D - \alpha_Y)$ |
| (x) | 1 | 0 | 0 | 1 | $\min\{f_t(0, e_t)(1 + \alpha_D - \alpha_Y), 1\}$ | $f_t(1, e_t)(1 - \alpha_D + \alpha_Y)$ |
| (xi) | 1 | 0 | 1 | 0 | $\min\{f_t(0, e_t)(1 + \alpha_D + \alpha_Y), 1\}$ | $f_t(1, e_t)(1 - \alpha_D - \alpha_Y)$ |
| (xii) | 1 | 0 | 1 | 1 | $\min\{f_t(0, e_t)(1 + \alpha_D + \alpha_Y), 1\}$ | $f_t(1, e_t)(1 - \alpha_D + \alpha_Y)$ |
| (xiii) | 1 | 1 | 0 | 0 | $f_t(0, e_t)$ | $f_t(1, e_t)$ |
| (xiv) | 1 | 1 | 0 | 1 | $\min\{f_t(0, e_t)(1 + \alpha_D - \alpha_Y), 1\}$ | $\min\{f_t(1, e_t)(1 + \alpha_D + \alpha_Y), 1\}$ |
| (xv) | 1 | 1 | 1 | 0 | $\min\{f_t(0, e_t)(1 + \alpha_D + \alpha_Y), 1\}$ | $\min\{f_t(1, e_t)(1 + \alpha_D - \alpha_Y), 1\}$ |
| (xvi) | 1 | 1 | 1 | 1 | $f_t(0, e_t)$ | $f_t(1, e_t)$ |

Table 2: When $\alpha_D < \alpha_Y$, compare cases of $H_{t+1}$ with different values of $A_t$.

| Case | $D_t$ | | $Y_t^{(\text{ori})}$ | | $H_{t+1}$ | |
|---|---|---|---|---|---|---|
| | if $A_t = 0$ | if $A_t = 1$ | if $A_t = 0$ | if $A_t = 1$ | if $A_t = 0$ | if $A_t = 1$ |
| (i) | 0 | 0 | 0 | 0 | $f_t(0, e_t)$ | $f_t(1, e_t)$ |
| (ii) | 0 | 0 | 0 | 1 | $f_t(0, e_t)(1 - \alpha_D - \alpha_Y)$ | $\min\{f_t(1, e_t)(1 - \alpha_D + \alpha_Y), 1\}$ |
| (iii) | 0 | 0 | 1 | 0 | $\min\{f_t(0, e_t)(1 - \alpha_D + \alpha_Y), 1\}$ | $f_t(1, e_t)(1 - \alpha_D - \alpha_Y)$ |
| (iv) | 0 | 0 | 1 | 1 | $f_t(0, e_t)$ | $f_t(1, e_t)$ |
| (v) | 0 | 1 | 0 | 0 | $f_t(0, e_t)(1 - \alpha_D - \alpha_Y)$ | $f_t(1, e_t)(1 + \alpha_D - \alpha_Y)$ |
| (vi) | 0 | 1 | 0 | 1 | $f_t(0, e_t)(1 - \alpha_D - \alpha_Y)$ | $\min\{f_t(1, e_t)(1 + \alpha_D + \alpha_Y), 1\}$ |
| (vii) | 0 | 1 | 1 | 0 | $\min\{f_t(0, e_t)(1 - \alpha_D + \alpha_Y), 1\}$ | $f_t(1, e_t)(1 + \alpha_D - \alpha_Y)$ |
| (viii) | 0 | 1 | 1 | 1 | $\min\{f_t(0, e_t)(1 - \alpha_D + \alpha_Y), 1\}$ | $\min\{f_t(1, e_t)(1 + \alpha_D + \alpha_Y), 1\}$ |
| (ix) | 1 | 0 | 0 | 0 | $f_t(0, e_t)(1 + \alpha_D - \alpha_Y)$ | $f_t(1, e_t)(1 - \alpha_D - \alpha_Y)$ |
| (x) | 1 | 0 | 0 | 1 | $f_t(0, e_t)(1 + \alpha_D - \alpha_Y)$ | $\min\{f_t(1, e_t)(1 - \alpha_D + \alpha_Y), 1\}$ |
| (xi) | 1 | 0 | 1 | 0 | $\min\{f_t(0, e_t)(1 + \alpha_D + \alpha_Y), 1\}$ | $f_t(1, e_t)(1 - \alpha_D - \alpha_Y)$ |
| (xii) | 1 | 0 | 1 | 1 | $\min\{f_t(0, e_t)(1 + \alpha_D + \alpha_Y), 1\}$ | $\min\{f_t(1, e_t)(1 - \alpha_D + \alpha_Y), 1\}$ |
| (xiii) | 1 | 1 | 0 | 0 | $f_t(0, e_t)$ | $f_t(1, e_t)$ |
| (xiv) | 1 | 1 | 0 | 1 | $f_t(0, e_t)(1 + \alpha_D - \alpha_Y)$ | $\min\{f_t(1, e_t)(1 + \alpha_D + \alpha_Y), 1\}$ |
| (xv) | 1 | 1 | 1 | 0 | $\min\{f_t(0, e_t)(1 + \alpha_D + \alpha_Y), 1\}$ | $f_t(1, e_t)(1 + \alpha_D - \alpha_Y)$ |
| (xvi) | 1 | 1 | 1 | 1 | $f_t(0, e_t)$ | $f_t(1, e_t)$ |

Table 3: When $\alpha_D = \alpha_Y = \alpha$, compare cases of $H_{t+1}$ with different values of $A_t$.

| Case | $D_t$ | | $Y_t^{(ori)}$ | | $H_{t+1}$ | |
|------|-------|-------|---------------|-------|-----------|-----------|
| | if $A_t = 0$ | if $A_t = 1$ | if $A_t = 0$ | if $A_t = 1$ | if $A_t = 0$ | if $A_t = 1$ |
| (i) | 0 | 0 | 0 | 0 | $f_t(0, e_t)$ | $f_t(1, e_t)$ |
| (ii) | 0 | 0 | 0 | 1 | $f_t(0, e_t)(1 - 2\alpha)$ | $f_t(1, e_t)$ |
| (iii) | 0 | 0 | 1 | 0 | $f_t(0, e_t)$ | $f_t(1, e_t)(1 - 2\alpha)$ |
| (iv) | 0 | 0 | 1 | 1 | $f_t(0, e_t)$ | $f_t(1, e_t)$ |
| (v) | 0 | 1 | 0 | 0 | $f_t(0, e_t)(1 - 2\alpha)$ | $f_t(1, e_t)$ |
| (vi) | 0 | 1 | 0 | 1 | $f_t(0, e_t)(1 - 2\alpha)$ | $\min\{f_t(1, e_t)(1 + 2\alpha), 1\}$ |
| (vii) | 0 | 1 | 1 | 0 | $f_t(0, e_t)$ | $f_t(1, e_t)$ |
| (viii) | 0 | 1 | 1 | 1 | $f_t(0, e_t)$ | $\min\{f_t(1, e_t)(1 + 2\alpha), 1\}$ |
| (ix) | 1 | 0 | 0 | 0 | $f_t(0, e_t)$ | $f_t(1, e_t)(1 - 2\alpha)$ |
| (x) | 1 | 0 | 0 | 1 | $f_t(0, e_t)$ | $f_t(1, e_t)$ |
| (xi) | 1 | 0 | 1 | 0 | $\min\{f_t(0, e_t)(1 + 2\alpha), 1\}$ | $f_t(1, e_t)(1 - 2\alpha)$ |
| (xii) | 1 | 0 | 1 | 1 | $\min\{f_t(0, e_t)(1 + 2\alpha), 1\}$ | $f_t(1, e_t)$ |
| (xiii) | 1 | 1 | 0 | 0 | $f_t(0, e_t)$ | $f_t(1, e_t)$ |
| (xiv) | 1 | 1 | 0 | 1 | $f_t(0, e_t)$ | $\min\{f_t(1, e_t)(1 + 2\alpha), 1\}$ |
| (xv) | 1 | 1 | 1 | 0 | $\min\{f_t(0, e_t)(1 + 2\alpha), 1\}$ | $f_t(1, e_t)$ |
| (xvi) | 1 | 1 | 1 | 1 | $f_t(0, e_t)$ | $f_t(1, e_t)$ |

Table 4: When $\alpha_D > \alpha_Y$, list possible instantiations of $\varphi_{t+1}(e_{t+1})$.

| Case | $D_t$ | | $Y_t^{(ori)}$ | | | $\varphi_{t+1}(e_{t+1}) = f_{t+1}(0, e_{t+1}) - f_{t+1}(1, e_{t+1})$ |
|---|---|---|---|---|---|---|
| | if $A_t = 0$ | if $A_t = 1$ | if $A_t = 0$ | if $A_t = 1$ | | |
| (i) | 0 | 0 | 0 | 0 | | $\varphi_t(e_t)$ |
| (ii) | 0 | 0 | 0 | 1 | | $\varphi_t(e_t)(1 - \alpha_D) - \alpha_Y \eta_t(e_t)$ |
| (iii) | 0 | 0 | 1 | 0 | | $\varphi_t(e_t)(1 - \alpha_D) + \alpha_Y \eta_t(e_t)$ |
| (iv) | 0 | 0 | 1 | 1 | | $\varphi_t(e_t)$ |
| (v) | 0 | 1 | 0 | 0 | (v.1) | $\varphi_t(e_t)(1 - \alpha_Y) - \alpha_D \eta_t(e_t)$, if $f_t(1, e_t) \in (0, \frac{1}{1 + \alpha_D - \alpha_Y})$ |
| | - | - | - | - | (v.2) | $f_t(0, e_t)(1 - \alpha_D - \alpha_Y) - 1$, otherwise |
| (vi) | 0 | 1 | 0 | 1 | (vi.1) | $\varphi_t(e_t) - (\alpha_D + \alpha_Y)\eta_t(e_t)$, if $f_t(1, e_t) \in (0, \frac{1}{1 + \alpha_D + \alpha_Y})$ |
| | - | - | - | - | (vi.2) | $f_t(0, e_t)(1 - \alpha_D - \alpha_Y) - 1$, otherwise |
| (vii) | 0 | 1 | 1 | 0 | (vii.1) | $\varphi_t(e_t) - (\alpha_D - \alpha_Y)\eta_t(e_t)$, if $f_t(1, e_t) \in (0, \frac{1}{1 + \alpha_D - \alpha_Y})$ |
| | - | - | - | - | (vii.2) | $f_t(0, e_t)(1 - \alpha_D + \alpha_Y) - 1$, otherwise |
| (viii) | 0 | 1 | 1 | 1 | (viii.1) | $\varphi_t(e_t)(1 + \alpha_Y) - \alpha_D \eta_t(e_t)$, if $f_t(1, e_t) \in (0, \frac{1}{1 + \alpha_D + \alpha_Y})$ |
| | - | - | - | - | (viii.2) | $f_t(0, e_t)(1 - \alpha_D + \alpha_Y) - 1$, otherwise |
| (ix) | 1 | 0 | 0 | 0 | (ix.1) | $\varphi_t(e_t)(1 - \alpha_Y) + \alpha_D \eta_t(e_t)$, if $f_t(0, e_t) \in (0, \frac{1}{1 + \alpha_D - \alpha_Y})$ |
| | - | - | - | - | (ix.2) | $1 - f_t(1, e_t)(1 - \alpha_D - \alpha_Y)$, otherwise |
| (x) | 1 | 0 | 0 | 1 | (x.1) | $\varphi_t(e_t) + (\alpha_D - \alpha_Y)\eta_t(e_t)$, if $f_t(0, e_t) \in (0, \frac{1}{1 + \alpha_D - \alpha_Y})$ |
| | - | - | - | - | (x.2) | $1 - f_t(1, e_t)(1 - \alpha_D + \alpha_Y)$, otherwise |
| (xi) | 1 | 0 | 1 | 0 | (xi.1) | $\varphi_t(e_t) + (\alpha_D + \alpha_Y)\eta_t(e_t)$, if $f_t(0, e_t) \in (0, \frac{1}{1 + \alpha_D + \alpha_Y})$ |
| | - | - | - | - | (xi.2) | $1 - f_t(1, e_t)(1 - \alpha_D - \alpha_Y)$, otherwise |
| (xii) | 1 | 0 | 1 | 1 | (xii.1) | $\varphi_t(e_t)(1 + \alpha_Y) + \alpha_D \eta_t(e_t)$, if $f_t(0, e_t) \in (0, \frac{1}{1 + \alpha_D + \alpha_Y})$ |
| | | | | | (xii.2) | $1 - f_t(1, e_t)(1 - \alpha_D + \alpha_Y)$, otherwise |

| Case | $D_t$ | | $Y_t^{(ori)}$ | | $\varphi_{t+1}(e_{t+1}) = f_{t+1}(0, e_{t+1}) - f_{t+1}(1, e_{t+1})$ |
|---|---|---|---|---|---|
| | if $A_t = 0$ | if $A_t = 1$ | if $A_t = 0$ | if $A_t = 1$ | |
| (xiii) | 1 | 1 | 0 | 0 | $\varphi_t(e_t)$ |
| (xiv) | 1 | 1 | 0 | 1 | (xiv.1)  $\varphi_t(e_t)(1 + \alpha_D) - \alpha_Y \eta_t(e_t)$, if $f_t(0, e_t) \in (0, \frac{1}{1+\alpha_D-\alpha_Y})$ and $f_t(1, e_t) \in (0, \frac{1}{1+\alpha_D+\alpha_Y})$ |
| - | - | - | - | - | (xiv.2)  $f_t(0, e_t)(1 + \alpha_D - \alpha_Y) - 1$, if $f_t(0, e_t) \in (0, \frac{1}{1+\alpha_D-\alpha_Y})$ and $f_t(1, e_t) \in [\frac{1}{1+\alpha_D+\alpha_Y}, 1]$ |
| - | - | - | - | - | (xiv.3)  $1 - f_t(1, e_t)(1 + \alpha_D + \alpha_Y)$, if $f_t(0, e_t) \in [\frac{1}{1+\alpha_D-\alpha_Y}, 1]$ and $f_t(1, e_t) \in (0, \frac{1}{1+\alpha_D+\alpha_Y})$ |
| - | - | - | - | - | (xiv.4)  $0$, otherwise |
| (xv) | 1 | 1 | 1 | 0 | (xv.1)  $\varphi_t(e_t)(1 + \alpha_D) + \alpha_Y \eta_t(e_t)$, if $f_t(0, e_t) \in (0, \frac{1}{1+\alpha_D+\alpha_Y})$ and $f_t(1, e_t) \in (0, \frac{1}{1+\alpha_D-\alpha_Y})$ |
| - | - | - | - | - | (xv.2)  $f_t(0, e_t)(1 + \alpha_D + \alpha_Y) - 1$, if $f_t(0, e_t) \in (0, \frac{1}{1+\alpha_D+\alpha_Y})$ and $f_t(1, e_t) \in [\frac{1}{1+\alpha_D-\alpha_Y}, 1]$ |
| - | - | - | - | - | (xv.3)  $1 - f_t(1, e_t)(1 + \alpha_D - \alpha_Y)$, if $f_t(0, e_t) \in [\frac{1}{1+\alpha_D+\alpha_Y}, 1]$ and $f_t(1, e_t) \in (0, \frac{1}{1+\alpha_D-\alpha_Y})$ |
| - | - | - | - | - | (xv.4)  $0$, otherwise |
| (xvi) | 1 | 1 | 1 | 1 | $\varphi_t(e_t)$ |

Table 5: When $\alpha_D < \alpha_Y$, list possible instantiations of $\varphi_{t+1}(e_{t+1})$.

| Case | $D_t$ | | $Y_t^{(ori)}$ | | $\varphi_{t+1}(e_{t+1}) = f_{t+1}(0, e_{t+1}) - f_{t+1}(1, e_{t+1})$ | |
|------|-------|------|---------------|------|---|---|
| | if $A_t = 0$ | if $A_t = 1$ | if $A_t = 0$ | if $A_t = 1$ | | |
| (i) | 0 | 0 | 0 | 0 | | $\varphi_t(e_t)$ |
| (ii) | 0 | 0 | 0 | 1 | (ii.1) | $\varphi_t(e_t)(1 - \alpha_D) - \alpha_Y \eta_t(e_t)$, if $f_t(1, e_t) \in (0, \frac{1}{1 - \alpha_D + \alpha_Y})$ |
| | - | - | - | - | (ii.2) | $f_t(0, e_t)(1 - \alpha_D - \alpha_Y) - 1$, otherwise |
| (iii) | 0 | 0 | 1 | 0 | (iii.1) | $\varphi_t(e_t)(1 - \alpha_D) + \alpha_Y \eta_t(e_t)$, if $f_t(0, e_t) \in (0, \frac{1}{1 - \alpha_D + \alpha_Y})$ |
| | - | - | - | - | (iii.2) | $1 - f_t(1, e_t)(1 - \alpha_D - \alpha_Y)$, otherwise |
| (iv) | 0 | 0 | 1 | 1 | | $\varphi_t(e_t)$ |
| (v) | 0 | 1 | 0 | 0 | | $\varphi_t(e_t)(1 - \alpha_Y) - \alpha_D \eta_t(e_t)$ |
| (vi) | 0 | 1 | 0 | 1 | (vi.1) | $\varphi_t(e_t) - (\alpha_D + \alpha_Y)\eta_t(e_t)$, if $f_t(1, e_t) \in (0, \frac{1}{1 + \alpha_D + \alpha_Y})$ |
| | - | - | - | - | (vi.2) | $f_t(0, e_t)(1 - \alpha_D - \alpha_Y) - 1$, otherwise |
| (vii) | 0 | 1 | 1 | 0 | (vii.1) | $\varphi_t(e_t) - (\alpha_D - \alpha_Y)\eta_t(e_t)$, if $f_t(0, e_t) \in (0, \frac{1}{1 - \alpha_D + \alpha_Y})$ |
| | - | - | - | - | (vii.2) | $1 - f_t(1, e_t)(1 + \alpha_D - \alpha_Y)$, otherwise |
| (viii) | 0 | 1 | 1 | 1 | (viii.1) | $\varphi_t(e_t)(1 + \alpha_Y) - \alpha_D \eta_t(e_t)$, if $f_t(0, e_t) \in (0, \frac{1}{1 - \alpha_D + \alpha_Y})$ and $f_t(1, e_t) \in (0, \frac{1}{1 + \alpha_D + \alpha_Y})$ |
| | - | - | - | - | (viii.2) | $f_t(0, e_t)(1 - \alpha_D + \alpha_Y) - 1$, if $f_t(0, e_t) \in (0, \frac{1}{1 - \alpha_D + \alpha_Y})$ and $f_t(1, e_t) \in [\frac{1}{1 + \alpha_D + \alpha_Y}, 1]$ |
| | - | - | - | - | (viii.3) | $1 - f_t(1, e_t)(1 + \alpha_D + \alpha_Y)$, if $f_t(0, e_t) \in [\frac{1}{1 - \alpha_D + \alpha_Y}, 1]$ and $f_t(1, e_t) \in (0, \frac{1}{1 + \alpha_D + \alpha_Y})$ |
| | - | - | - | - | (viii.4) | $0$, otherwise |
| (ix) | 1 | 0 | 0 | 0 | | $\varphi_t(e_t)(1 - \alpha_Y) + \alpha_D \eta_t(e_t)$ |
| (x) | 1 | 0 | 0 | 1 | (x.1) | $\varphi_t(e_t) + (\alpha_D - \alpha_Y)\eta_t(e_t)$, if $f_t(1, e_t) \in (0, \frac{1}{1 - \alpha_D + \alpha_Y})$ |
| | - | - | - | - | (x.2) | $f_t(0, e_t)(1 + \alpha_D - \alpha_Y) - 1$, otherwise |
| (xi) | 1 | 0 | 1 | 0 | (xi.1) | $\varphi_t(e_t) + (\alpha_D + \alpha_Y)\eta_t(e_t)$, if $f_t(0, e_t) \in (0, \frac{1}{1 + \alpha_D + \alpha_Y})$ |
| | - | - | - | - | (xi.2) | $1 - f_t(1, e_t)(1 - \alpha_D - \alpha_Y)$, otherwise |

| Case | $D_t$ | | $Y_t^{(ori)}$ | | $\varphi_{t+1}(e_{t+1}) = f_{t+1}(0,e_{t+1}) - f_{t+1}(1,e_{t+1})$ | |
|---|---|---|---|---|---|---|
| | if $A_t = 0$ | if $A_t = 1$ | if $A_t = 0$ | if $A_t = 1$ | | |
| (xii) | 1 | 0 | 1 | 1 | (xii.1) | $\varphi_t(e_t)(1+\alpha_Y) + \alpha_D \eta_t(e_t)$, if $f_t(0,e_t) \in (0, \frac{1}{1+\alpha_D+\alpha_Y})$ and $f_t(1,e_t) \in (0, \frac{1}{1-\alpha_D+\alpha_Y})$ |
| | - | - | - | - | (xii.2) | $f_t(0,e_t)(1+\alpha_D+\alpha_Y) - 1$, if $f_t(0,e_t) \in (0, \frac{1}{1+\alpha_D+\alpha_Y})$ and $f_t(1,e_t) \in [\frac{1}{1-\alpha_D+\alpha_Y}, 1]$ |
| | - | - | - | - | (xii.3) | $1 - f_t(1,e_t)(1-\alpha_D+\alpha_Y)$, if $f_t(0,e_t) \in [\frac{1}{1+\alpha_D+\alpha_Y}, 1]$ and $f_t(1,e_t) \in (0, \frac{1}{1-\alpha_D+\alpha_Y})$ |
| | - | - | - | - | (xii.4) | $0$, otherwise |
| (xiii) | 1 | 1 | 0 | 0 | | $\varphi_t(e_t)$ |
| (xiv) | 1 | 1 | 0 | 1 | (xiv.1) | $\varphi_t(e_t)(1+\alpha_D) - \alpha_Y \eta_t(e_t)$, if $f_t(1,e_t) \in (0, \frac{1}{1+\alpha_D+\alpha_Y})$ |
| | - | - | - | - | (xiv.2) | $f_t(0,e_t)(1+\alpha_D-\alpha_Y) - 1$, otherwise |
| (xv) | 1 | 1 | 1 | 0 | (xv.1) | $\varphi_t(e_t)(1+\alpha_D) + \alpha_Y \eta_t(e_t)$, if $f_t(0,e_t) \in (0, \frac{1}{1+\alpha_D+\alpha_Y})$ |
| | - | - | - | - | (xv.2) | $1 - f_t(1,e_t)(1+\alpha_D-\alpha_Y)$, otherwise |
| (xvi) | 1 | 1 | 1 | 1 | | $\varphi_t(e_t)$ |

Table 6: When $\alpha_D = \alpha_Y = \alpha$, list possible instantiations of $\varphi_{t+1}(e_{t+1})$.

| Case | $D_t$ | | $Y_t^{(ori)}$ | | $\varphi_{t+1}(e_{t+1}) = f_{t+1}(0, e_{t+1}) - f_{t+1}(1, e_{t+1})$ | |
|------|-------|-------|---------------|-------|---|---|
| | if $A_t = 0$ | if $A_t = 1$ | if $A_t = 0$ | if $A_t = 1$ | | |
| (i) | 0 | 0 | 0 | 0 | | $\varphi_t(e_t)$ |
| (ii) | 0 | 0 | 0 | 1 | | $\varphi_t(e_t)(1 - \alpha) - \alpha\eta_t(e_t)$ |
| (iii) | 0 | 0 | 1 | 0 | | $\varphi_t(e_t)(1 - \alpha) + \alpha\eta_t(e_t)$ |
| (iv) | 0 | 0 | 1 | 1 | | $\varphi_t(e_t)$ |
| (v) | 0 | 1 | 0 | 0 | | $\varphi_t(e_t)(1 - \alpha) - \alpha\eta_t(e_t)$ |
| (vi) | 0 | 1 | 0 | 1 | (vi.1) | $\varphi_t(e_t) - 2\alpha\eta_t(e_t)$, if $f_t(1, e_t) \in (0, \frac{1}{1+2\alpha})$ |
| - | - | - | - | - | (vi.2) | $f_t(0, e_t)(1 - 2\alpha) - 1$, otherwise |
| (vii) | 0 | 1 | 1 | 0 | | $\varphi_t(e_t)$ |
| (viii) | 0 | 1 | 1 | 1 | (viii.1) | $\varphi_t(e_t)(1 + \alpha) - \alpha\eta_t(e_t)$, if $f_t(1, e_t) \in (0, \frac{1}{1+2\alpha})$ |
| - | - | - | - | - | (viii.2) | $f_t(0, e_t) - 1$, otherwise |
| (ix) | 1 | 0 | 0 | 0 | | $\varphi_t(e_t)(1 - \alpha) + \alpha\eta_t(e_t)$ |
| (x) | 1 | 0 | 0 | 1 | | $\varphi_t(e_t)$ |
| (xi) | 1 | 0 | 1 | 0 | (xi.1) | $\varphi_t(e_t) + 2\alpha\eta_t(e_t)$, if $f_t(0, e_t) \in (0, \frac{1}{1+2\alpha})$ |
| - | - | - | - | - | (xi.2) | $1 - f_t(1, e_t)(1 - 2\alpha)$, otherwise |
| (xii) | 1 | 0 | 1 | 1 | (xii.1) | $\varphi_t(e_t)(1 + \alpha) + \alpha\eta_t(e_t)$, if $f_t(0, e_t) \in (0, \frac{1}{1+2\alpha})$ |
| - | - | - | - | - | (xii.2) | $1 - f_t(1, e_t)$, otherwise |
| (xiii) | 1 | 1 | 0 | 0 | | $\varphi_t(e_t)$ |
| (xiv) | 1 | 1 | 0 | 1 | (xiv.1) | $\varphi_t(e_t)(1 + \alpha) - \alpha\eta_t(e_t)$, if $f_t(1, e_t) \in (0, \frac{1}{1+2\alpha})$ |
| - | - | - | - | - | (xiv.2) | $f_t(0, e_t) - 1$, otherwise |
| (xv) | 1 | 1 | 1 | 0 | (xv.1) | $\varphi_t(e_t)(1 + \alpha) + \alpha\eta_t(e_t)$, if $f_t(0, e_t) \in (0, \frac{1}{1+2\alpha})$ |
| - | - | - | - | - | (xv.2) | $1 - f_t(1, e_t)$, otherwise |
| (xvi) | 1 | 1 | 1 | 1 | | $\varphi_t(e_t)$ |

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
