# OpenReview forum: "Tier Balancing: Towards Dynamic Fairness over Underlying Causal Factors"
_ICLR.cc/2023/Conference — ICLR 2023 poster_

### Official Review · Reviewer_qVYP · 2022-10-25

**Confidence:** 2
**Correctness:** 3
**Technical Novelty And Significance:** 4
**Empirical Novelty And Significance:** 3
**Recommendation:** 6

**Clarity, Quality, Novelty And Reproducibility:**

The paper is well written with comprehensive studies on their newly proposed long-term fairness.

The work is quite new to my limited knowledge of fairness.

The code and proof are provided with details.

**Strength And Weaknesses:**

*Strength*:

1. This paper considers an interesting problem of long-term fairness and provides a corresponding new definition.

2. The authors gave a systematical analysis of achieving long-term fairness through single-step intervention and dynamic interventions.

3. Numerical studies show their promising prospect.

4. The paper is well written.

*Weaknesses*:

1. The connection to existing fairness works can be further improved. For instance, how about linking the proposed TB to existing fairness metrics mathematically? It would be nice to see it under the non-stationary dynamic that boils down or is close to classical ones.

2. I would expect more motivation of proposing long-term fairness from real examples. The authors may consider expanding the FICO data and elaborating on it in the introduction, instead of simply stating 'in our running example of credit application' without any content.

3. It is not clear if the assumptions stated are reasonable. For instance, in Assumption 3.3, is number 2 in Equation 3 and $\alpha \in [0,1/2)$ necessary? Can we change it to other combos, say 3 and 1/3? More justification is also needed for Assumption 3.4 besides using the credit example. The qualitative assumptions in 3.5 and 3.6 should be justified in comparison to the literature.

4. Sensitivity analysis may be considered on assumption violation to enrich the numerical results.



**Summary Of The Paper:**

This paper proposed a new concept for long-term fairness, called Tier Balancing (TB), under decision-making dynamics. The proposed TB characterizes the decision-distribution interplay with latent causal factors. Given some specific data dynamics, the authors proved that one cannot directly achieve the long-term fairness goal through a single-step decision. This further motivates them to develop a sequence of algorithmic interventions to get closer to the long-term fairness goal. Their method has been examined by both synthetic and real data.

**Summary Of The Review:**

Overall, this paper considers an interesting problem of long-term fairness and is well-written. The authors gave a systematical analysis of achieving long-term fairness through single-step intervention v.s. dynamic interventions. Yet, I have a few concerns regarding their connection to the literature, the motivation, and the required strong assumptions.

---

> ### Author Response · Authors · 2022-11-17
> **Response to Reviewer qVYP (1 / 2)**
>
> We greatly appreciate the reviewer’s time, thoughtful and encouraging comments, and constructive suggestions!
> In our revised manuscript, we (1) **condensed and updated notations** to make sure they are informative, rigorous, and concise,
> (2) added **additional discussions related to motivation,  modeling details, and related literature** and reorganize the materials according to reviewer's suggestions thoroughly (main text and appendix),
> and (3) added **additional experimental results** and further improve the connection between our theoretical and empirical results.
> We also included **the list of side notes on page 13** to help locate the related contents.
> Below please see our responses to specific points in the review comments:
>
> ---
>
> ### **C1:** "The connection to existing fairness works can be further improved. For instance, how about linking the proposed TB to existing fairness metrics mathematically? It would be nice to see it under the non-stationary dynamic that boils down or is close to classical ones."
>
> **R1:** Thank you for your thoughtful comment and suggestion on the connection of TB to existing notions. Let us respond in two folds:
>
> (1) If we are okay with focusing on binary latent causal factors, we can treat the potential outcome $Y_t^{(\mathrm{ori})}$ itself as the latent tier. If we only use the trivial decision-making strategy and always approve the loan, i.e., $D_{t - 1} \equiv 1$, then $Y_t^{(\mathrm{obs})} = Y_t^{(\mathrm{ori})}$ and mathematically, TB boils down to Demographic Parity (DP) since requiring $Y_t^{(\mathrm{ori})} \perp\\!\\!\perp A_t$ (TB) is equivalent to requiring $Y_t^{(\mathrm{obs})} \perp\\!\\!\perp A_t$ (DP).
>
> (2) A more interesting result is: when TB is initially satisfied, under the non-stationary dynamic one can keep applying nontrivial DP decision-making strategies to main the status of satisfying TB. We believe this provides additional connection between DP and TB and include this result in Appendix B.2. For the convenience of locating the related discussion, we provide the side note `Re: C1 by Reviewer qVYP` on page 17.
>
> ---
>
> ### **C2:** "I would expect more motivation of proposing long-term fairness from real examples. The authors may consider expanding the FICO data and elaborating on it in the introduction, instead of simply stating 'in our running example of credit application' without any content."
>
> **R2:** Thank you for your insightful and constructive suggestions. Following your suggestion, we have updated the introduction and stated the high-level motivation of inducing a fair future in the long-run (by modeling and understanding the interplay between underlying data generating processes and decision-making dynamics).
>
> We also present additional discussions with respect to practical scenarios of interest in Appendix B.1.2 (due to space limit). For the convenience of locating the related discussion, we provide the side note `Re: C2 by Reviewer qVYP` on page 16.
>
> ---
>
> ### **Q3:** "Assumption 3.3, is number 2 in Equation 3 and $\alpha \in [0, 1/2)$ necessary? Can we change it to other combos, say 3 and 1/3?"
>
> **R3:** Thank you for considering our model in detail. The number $2$ and $\alpha \in [0, 1/2)$ in Assumption 3.3 are only regularization conditions. We opt for this combination of values because for binary $D_t$ and $Y_t^{\mathrm{(ori)}}$ with domains of value $\\{0, 1\\}$, $(2 D_t - 1)$ and $(2 Y_t^{\mathrm{(ori)}} - 1)$ take value in $\\{-1, +1\\}$ and can act as $-$ and $+$ symbols (corresponding to the decrease and increase update of the next tier $H_{t + 1}$). This, together with $\alpha \in [0, 1/2)$ makes sure the updated tier is not abruptly "saturated" towards $0$ or $1$.
>
> ---
>
> (continuing)

---

> > ### Author Response · Authors · 2022-11-17
> > **Response to Reviewer qVYP (2 / 2)**
> >
> > (continued)
> >
> > ---
> >
> > ### **C4:** "More justification is also needed for Assumption 3.4 besides using the credit example."
> >
> > **R4:** Thank you for your constructive suggestion. Assumption 3.4 can be justified in the repetitive resource application scenarios, where resource is allocated to agents (in a more general sense) in a sequence of decisions.
> > For instance, in credit application, the agents are clients whose protected feature (e.g., race) do not change across time.
> > In predictive policing, the agents are neighborhoods whose protected feature (e.g., the ratio of demographic groups in population) do not change, or change at a very slow rate compared to the timescale of decision-making in patrol resource allocation.
> > In the dual market pipeline (e.g., temporary labor markets followed by the permanent labor market) or the admission-followed-by-hiring pipeline, the agents are applicants whose protected feature (e.g., race) remains the same across time.
> > We believe this is also related to your comment **C2** where we discuss practical scenarios.
> > For the convenience of locating the discussion, we provide the side note `Re: C2 by Reviewer qVYP` on page 16.
> >
> > ---
> >
> > ### **C5:** "The qualitative assumptions in 3.5 and 3.6 should be justified in comparison to the literature."
> >
> > **R5:** Assumptions 3.5 and 3.6 are introduced to accompany our analysis for TB fairness notion and are purely for technical purposes.
> > The qualitative part of the assumption is rather mild, stating that discriminations against certain group often follow a similar pattern, e.g., favorable (undesirable) outcomes tend to be enjoyed (suffered) by the advantaged (disadvantaged) group.
> > This is implicitly assumed by previous algorithmic fairness literature, e.g., Hardt et al. (2016), Kusner et al. (2017), Chiappa (2019), etc.
> >
> > ---
> >
> > ### **C6:** "Sensitivity analysis may be considered on assumption violation to enrich the numerical results."
> >
> > **R6:** Thank you for your very helpful suggestion. Following your suggestion, we have added additional experimental results for accuracy-oriented decision-making strategies in Appendix B.6.
> > The purpose of this experiment is to empirically show the results on the real-world FICO score data set when certain assumptions (in theoretical results) are violated.
> >
> > Since we initialize the distribution of the latent tier $H$ based on CDF of TransRisk scores, the quantitative assumption 3.6 is violated.
> > Since accuracy-oriented predictor does not reach $100\%$ accuracy, the assumption of perfect predictor is also violated.
> > As we can see from the additional experimental results, there is no obvious evidence that accuracy-oriented predictor can help in achieving TB.
> > The observation aligns with our theoretical analysis on perfect predictors, even if aforementioned assumptions are violated in the empirical study. For the convenience of locating related results, we provide the side note `Re: C6 by Reviewer qVYP` on page 21.
> >
> > ---
> >
> > ### References
> >
> > Moritz Hardt, Eric Price, and Nati Srebro. Equality of opportunity in supervised learning. In Advances in Neural Information Processing Systems, pp. 3315–3323, 2016.
> >
> > Matt J Kusner, Joshua Loftus, Chris Russell, and Ricardo Silva. Counterfactual fairness. In Advances in Neural Information Processing Systems, pp. 4066–4076, 2017.
> >
> > Silvia Chiappa. Path-specific counterfactual fairness. In Proceedings of the AAAI Conference on Artificial Intelligence, volume 33, pp. 7801–7808, 2019.

---

### Official Review · Reviewer_WEUm · 2022-10-27

**Confidence:** 3
**Clarity, Quality, Novelty And Reproducibility:** I didn't find the paper to be very cl…
**Correctness:** 3
**Technical Novelty And Significance:** 3
**Empirical Novelty And Significance:** 3
**Recommendation:** 6

**Strength And Weaknesses:**

[Strength]

S1: The paper tries to demonstrate how to achieve long-term dynamic fairness

[Weakness]

W1: Writing could be improved; it is difficult to parse through sections 3 and 4 at first glance. I would encourage condensing the notations and adding more motivations for the said approach.

W2: It seems that the conditional independence between the latent factors and the sensitive attributes could fail, for example, in health care or public health setting [1].

W3: Adding more intuition about masking the observed outcomes based on the decisions would help. Moreover, the setup of repeated applications, although relevant in the credit setting, is not very general. When the treatment is designed for improving the outcome, as in healthcare settings, repeated treatments could, in turn, be a negative factor. Adding more settings where this framework applies would be helpful.


1. Mhasawade, Vishwali, and Rumi Chunara. "Causal multi-level fairness." Proceedings of the 2021 AAAI/ACM Conference on AI, Ethics, and Society. 2021.

**Summary Of The Paper:**

The paper proposes a technique to achieve long-term dynamic fairness by incorporating latent factors that can affect the outcomes and also related to the sensitive attributes. The paper also shows that in general it is not possible to achieve long-term fairness only through a one-step intervention, i.e., static decision-making but rather requires sequential decisions in certain situations. The paper also shows that under these conditions it is possible to get closer to the long-term fairness goal through Counterfactual Fair decisions. The empirical results support the theoretical claims.

**Summary Of The Review:**

The paper tries to solve the important issue of achieving long-term fairness. I vote for marginally accepting with the hope that clarity is improved.

---

> ### Author Response · Authors · 2022-11-17
> **Response to Reviewer WEUm**
>
> We greatly appreciate the reviewer’s time, insightful and encouraging comments, and constructive suggestions!
> In our revised manuscript, we (1) **condensed and updated notations** to make sure they are informative, rigorous, and concise,
> (2) added **additional discussions related to motivation,  modeling details, and related literature** and reorganize the materials according to reviewer's suggestions thoroughly (main text and appendix),
> and (3) added **additional experimental results** and further improve the connection between our theoretical and empirical results.
> We also included **the list of side notes on page 13** to help locate the related contents.
> Below please see our responses to specific points in the review comments:
>
> ---
>
> ### **W1:** "Writing could be improved; it is difficult to parse through sections 3 and 4 at first glance. I would encourage condensing the notations and adding more motivations for the said approach."
>
> **R1:** Thank you for your comments and suggestions. Following your suggestion, we have updated and condensed our notations and also refined the material organizations between the main text (including but not limited to Sections 3 and 4) and the appendix. For the purpose of making the motivation and takeaway messages clear and informative, we include (1) additional discussions on decision-distribution interplay (Appendix B.1), (2) a remark on Fact 3.2 (Appendix B.3), (3) detailed derivation and implication of $\Delta_{\mathrm{STIR}} \rvert_{t}^{t + 1}$ (Appendix B.4), and also provide corresponding pointers in the main text. We use the blue-colored font to indicate the updates in our latest manuscript.
>
> ---
>
> ### **W2:** "It seems that the conditional independence between the latent factors and the sensitive attributes could fail, for example, in health care or public health setting [Mhasawade and Chunara, 2021]."
>
> **R2:** Thank you for providing a very related reference. We totally agree with you that the application of fairness notions should consider the practical scenarios. In light of your comment, we have added additional discussions on modeling choices for different practical scenarios. We also consider the settings where our causal modeling and fairness notion may or may not be suitable and provide discussions accordingly (Appendix B.1.2). For the convenience of locating the related discussion, we provide the side note `Re: W2 by Reviewer WEUm` on page 17.
>
> ---
>
> ### **W3:** "Adding more intuition about masking the observed outcomes based on the decisions would help."
>
> **R3:** Thank you for your thoughtful comment on our model detail. We believe you are referring to the distinction and connection between the (previous) decision $D_{t - 1}$, the potential outcome $Y_t^{(\mathrm{ori})}$, and the observed outcome $Y_t^{(\mathrm{obs})}$.
>
> For example, we can observe $Y_t^{(\mathrm{obs})}$ _only_ if in the previous step $D_{t - 1} = 1$. If the individual was denied the loan in the previous step ($D_{t - 1} = 0$), $Y_t^{(\mathrm{obs})}$ is not observable, i.e., masked by $D_{t - 1}$. From the viewpoint of decision-making, such data point is not informative in the sense that $Y_t^{(\mathrm{obs})}$ for this particular individual is undefined. Motivated by different roles played by $Y_t^{(\mathrm{ori})}$ and $Y_t^{(\mathrm{obs})}$, we explicitly model this "masking" mechanism. For the convenience of locating the related discussion, we provide the side note `Re: W3 by Reviewer WEUm` on page 3.
>
> ---
>
> ### **C4:** "When the treatment is designed for improving the outcome, as in healthcare settings, repeated treatments could, in turn, be a negative factor. Adding more settings where this framework applies would be helpful."
>
> **R4:** Thank you for your insightful comment and constructive suggestion. Our causal modeling for repetitive resource allocation can be applied (with adaptation) in scenarios where agents apply for resources and subject to a sequence of decisions, for instance, credit applications, predictive policing, dual market pipelines, admission-followed-by-hiring pipelines. Following your suggestion, we add additional discussions on scenarios our framework can be applied in Appendix B.1.2. For the convenience of locating the related discussion, we provide the side note `Re: C4 by Reviewer WEUm` on page 17.

---

### Official Review · Reviewer_pPYU · 2022-11-02

**Confidence:** 5
**Correctness:** 3
**Technical Novelty And Significance:** 2
**Empirical Novelty And Significance:** 1
**Recommendation:** 5

**Clarity, Quality, Novelty And Reproducibility:**

Details are in strengths and weaknesses above.

While the narrative of the paper is clear, the technical exposition is disorganized and hard to follow. The concept of tier balancing is not as novel as is claimed. The results appear to be sound, but they hold under very specific modeling assumptions. The result pertaining to counterfactually fair decisions helping improve long term fairness needs further polishing. Experiments appear reproducible, but they do not support the message of the paper strongly.




**Strength And Weaknesses:**

### Strengths

* Long-term fairness is a very strongly motivated problem and various perspectives on characterizing and achieving it are welcome.
* Connecting causal and dynamic models in this framework is natural, and the paper’s model has some compelling features that could reflect many practical instances.

### Weaknesses

* The paper claims that the concept of tier balancing as the implicit long-term fairness objective is novel. However, many of the initial long-term fairness papers encoded similar social equality notions, e.g., the Mouzannar et al 2019 reference, where qualifications are equalized across protected groups.
* The dynamic model that is adopted has some key social implications which are not discussed. Namely, social mobility (tiers shifting) is influenced only by social outcomes and interventions. This is markedly false in real life, where prejudices and privileges dictated by protected groups affect mobility. Unfortunately, this is not a simple fix for this paper, since the introduction of any such dependence would severely change its analysis and conclusions.
* The dynamic model of the evolution of tiers, Eq. (3), is also too specific. In particular, I put in question the choice of a deterministic evolution, given labels and decisions. The entirety of the analysis hinges on this, as it determines exactly relationships between consecutive time steps, as enumerated case-by-case in the appendix tables. It is not clear how well the conclusions drawn from this determinism would apply to a more realistic stochastic model of social mobility. (A more specific nitpick I have is why mobility is influenced by non-instantiated labels, that is, why depend on $Y^{\textrm{(ori)}}$ and not on $Y^{\textrm{(obs)}}$? What I did not exhibit should not affect me.)
* The technical exposition is generally disorganized and may have some flaws. I had to go back and forth between the main text and appendices to piece together the mathematical formalisms. Here are some issues I encountered:
   * Most of the analysis is presented in terms of the functional causal model of the dynamic model. Here, at any time $t$, we characterize the tier $H_t$, the label $Y_t$ and the decision $D_t$ as respective functions of the attribute $A_t$ and exogenous randomness $E_t$. This is okay, except that the exogenous randomness must incorporate all the past evolution of society, namely because the current tier depends on them. As such, I believe it is not true that $E_t$ are independent for different $t$, as is mentioned in the appendix (p. 16).
   * $\Delta_\textrm{STIR}$ is not defined before it is characterized. I believe that, at time $t$, the definition is $\mathbf{E}[|f_{t+1}(0,E_{t+1})-f_{t+1}(1,E_{t+1})|]- \mathbf{E}[|f_t(0,E_t)-f_t(1,E_t)|]$. This should be stated first and foremost and the dependence on $t$ should be explicit.
   * $q_t$ is introduced as a conditional density on the pair of tiers (for each attribute). This might lead to understand it as a density with respect to the Lebesgue measure in $\mathbf{R}^2$, since tiers are real valued. However, this is not the case. This density is rather relative to the measure of the exogenous variables $E_t$, and should indeed always be written as a function of the value the latter takes ($\epsilon$), as follows: $q_t(f_t(0,\epsilon),f_t(1,\epsilon)|d,d’,y,y’                                                                                                                                                                                                               )$. It is worth clarifying this, otherwise the integration in Eq. (5) or the definition on p. 16 don’t make sense.
* Some of the main results are presented in a way that hides the intuition of why they hold:
   * Theorem 4.1’s limitations of why we cannot establish tier balance is due to the fact that the only interventions we have to control continuous random variables is a pair of discrete random variables.
   * Theorem 4.2’s perfect estimator making tier balance worse is due to the fact that the dynamics in Assumption 3.6 are polarizing based on labels only.
   * As for Theorem 4.3, there seems to be an issue in the proof. When simplifying $\Delta_\textrm{STIR}$ by assuming that $\alpha_Y=0$, it seems that the values of $\gamma$ (which determine $q_t$) are also set to something specific. If this is true, then it should also be part of the Theorem’s hypothesis.
* The experiments do not seem to strongly support the analysis. While we see the polarization caused by the perfect predictor in Fig.2, it is not clear whether the counterfactually fair one is any better, as the tiers do not appear balanced in Fig. 3(c).


### Question to Authors:
* I tried to understand Eq. (5). From what I gather, you wanted to not have to outright characterize the time $t+1$ quantities. Instead, you wanted to couple calculating $\mathbf{E}[|f_{t+1}(0,E_{t+1})-f_{t+1}(1,E_{t+1})|]$ at the same time as $\mathbf{E}[|f_t(0,E_t)-f_t(1,E_t)|]$, by using the fact that the tier dynamics allow you to know exactly what happens at time  $t+1$, conditionally on the decisions and labels. However, I’m not sure the coupling is done right. Instead of pushing the $E_t$ distribution forward to get $E_{t+1}$, you are forcing them to be the same, and have this secondary integral in $\xi$, which I am unable to make sense of. Instead, we know that conditionally on the labels, decisions, and $E_t=\epsilon$ (or equivalently the past tiers), exactly what the future tiers will be, so why not just do:
$$ \Delta_\textrm{STIR}(t) = \sum_{d,d’,y,y’} P_t(d,d’,y,y’) \int q_t(f_t(0,\epsilon),f_t(1,\epsilon)|d,d’,y,y’) \cdot (|\varphi_(t+1)(\epsilon,d,d’,y,y’)-\varphi_t(\epsilon)|) \mathrm{d} \epsilon $$
Where $\varphi_(t+1)(\epsilon,d,d’,y,y’)$ is precisely the update given in the tables in the appendix, which depend only on the past $\epsilon$ and the decisions and labels $d,d’,y,y’$.
* In Appendix B.2, you talk about tracking individuals. This is related to the previous point, but I am not sure how you are defining individuals, based on what properties or partitioning of the population? Part of it seems to suggest a kind of probability transport, through which you couple the integrations at both times, but ultimately you are using the fact that the next time step is determined by the first given the decisions and labels. Please clarify.

### Typos/Suggestions:
- (p.3) since their different roles > because of their different roles
- (p.5) inquiring > investigating
- (p.6) 3.2.2 Simplification Assumptions



**Summary Of The Paper:**

This paper is about defining long-term fairness in dynamic settings and characterizing whether it can be achieved with myopic decisions. A specific dynamic model is proposed, where tiers (representing social status) are a root cause of labels (representing social outcomes), and which in turn can be affected by decisions (representing social interventions). Another root cause influencing decisions are the protected attributes (representing historically underserved groups). Long-term fairness is defined as independence of tiers from protected attributes (representing a form of social equality), achieved over some time horizon. This is dubbed tier-balancing. Under several assumptions, the paper determines that (1) such fairness is not generally achievable over a single time step horizon, that (2) a perfect decision (mirroring the true label) worsens fairness, and that (3) counterfactually fair decisions could improve fairness.

**Summary Of The Review:**

The long term fairness consideration of this paper remains a timely topic. However, the specific notion adopted here is not novel. More importantly, the paper makes very specific modeling choices, ranging from outright violating the reality of prejudice to making social mobility not reflect the stochasticity in real life.Since the paper hinges on these modeling choices, changing them is not a simple exercise. Furthermore, experiments do not strongly support the analysis. As such, the relevance of this work to the community at large is limited. The paper is further hampered by unclear technical exposition.

[Post discussions: Thanks to the improved mathematical notation in the main text and other clarification, the score is raised. Some key concerns remain, detailed in a remark responding to authors.]

---

> ### Author Response · Authors · 2022-11-17
> **Response to Reviewer pPYU (1 / 3)**
>
> We are extremely grateful for the time devoted, and your thoughtful and detailed comments and suggestions, many of which help us further improve the manuscript!
> In our revised manuscript, we (1) **condensed and updated notations** to make sure they are informative, rigorous, and concise,
> (2) added **additional discussions related to motivation,  modeling details, and related literature** and reorganize the materials according to reviewer's suggestions thoroughly (main text and appendix),
> and (3) added **additional experimental results** and further improve the connection between our theoretical and empirical results.
> We also included **the list of side notes on page 13** to help locate the related contents.
>
> Please also kindly notice that there might be some misunderstandings.
> We hope our responses are helpful and can fully address your concern.
> We are eagerly looking forward to your feedback on our responses and updated manuscript.
> Below please see our responses to specific comments/questions in the review comment:
>
> ---
>
> ### **C1:** "The paper claims that the concept of tier balancing as the implicit long-term fairness objective is novel. However, many of the initial long-term fairness papers encoded similar social equality notions, e.g., the Mouzannar et al 2019 reference, where qualifications are equalized across protected groups."
>
> **R1:** Thank you for considering the relevant literature. Our notion is very different from the one considered by Mouzannar et al. (2019):
>
> - Individual-level vs. Group-level
>   - Mouzannar et al. (2019) explicitly focus on _Demographic Parity_ group-level fairness and characterize _qualification profile_, which contains only one value for the whole group.
>   - Our notion involves detailed causal modeling of the individual-level situation changes.
>
> - Underlying latent factor vs. Observed variables only
>   - The _qualification profile_ (Mouzannar et al. 2019) is a summarizing statistics of the _observed_ binary decision, i.e., $v$ in their work, for all agents within that group.
>   - The underlying latent factor, i.e., $H$ in our work, is an unobserved root cause that carries on the impact of current decision on future data distribution.
>
> We also provide additional discussions and note that the suitable modeling choice depends on practical scenario of interest, along with the side note `Re: C1 by Reviewer pPYU` on page 17.
>
> ---
>
> ### **C2:** "Mobility (tiers shifting) is influenced only by social outcomes and interventions. This is markedly false in real life, where prejudices and privileges dictated by protected groups affect mobility."
>
> **R2:** Thank you for your thoughtful comment and trying to go further.
>
> We **agree** that there is still a long way to go for long-term fairness analysis in terms of the modeling of the complicated social mobility.
>
> However, we **respectfully disagree** that tier shifting influenced by _social outcomes_ and _interventions_ is false. For example, if we view the FICO score as tier, the payment history (_"social outcomes"_ in terms of whether or not debt is repaid) and credit information (_"interventions"_ in terms of what kinds of credit are approved and utilized) are two primary driving forces behind the update of the FICO score.
> Please kindly let us know if we misunderstood you.
>
> We hope our detailed causal modeling of decision-distribution interplay can inspire further research into the important and challenging long-term fairness pursuit.
>
> ---
>
> ### **C3:** "The dynamic model of the evolution of tiers, Eq. (3), is also too specific. I put in question the choice of a deterministic evolution, given labels and decisions. The entirety of the analysis hinges on this, as it determines exactly relationships between consecutive time steps, as enumerated case-by-case in the appendix tables."
>
> **R3:** Thank you for carefully considering our model detail. Please kindly notice that there might be some misunderstanding.
>
> You are right that the updating dynamic (Equation 3) is a deterministic mapping. However, Equation 3 does **not** indicate that the entire data generating process itself is deterministic. The reason is that the decision $D_t$ and the outcome $Y_t^{\mathrm{(ori)}}$ can be stochastic. Such stochastic nature is preserved in the mapping from $H_t$ to $H_{t + 1}$ (Equation 3). For the convenience of locating the related discussion, we provide the side note `Re: C3 by Reviewer pPYU` on page 16.
>
> ---
>
> (continuing)

---

> > ### Author Response · Authors · 2022-11-17
> > **Response to Reviewer pPYU (2 / 3)**
> >
> > (continued)
> >
> > ---
> >
> > ### **C4:** "A more specific nitpick I have is why mobility is influenced by non-instantiated labels, that is, why depend on $Y^{\mathrm{(ori)}}_t$ and not on $Y^{\mathrm{(obs)}}_t$? What I did not exhibit should not affect me."
> >
> > **R4:** Thanks for the question. We interpret the non-instantiated label $Y_t^{\mathrm{(ori)}}$ as a potential outcome _had_ the individual received the loan previously. It reflects the actual ability to repay, which is an inherent fact of the individual and is irrelevant to whether or not such fact is exhibited. Therefore, when modeling the tier update, we utilize the non-instantiated but inherent fact $Y^{\mathrm{(ori)}}_t$ instead of the (potentially) observed copy $Y^{\mathrm{(obs)}}_t$. For the convenience of locating the related discussion, we provide the side note `Re: C4 by Reviewer pPYU` on page 3.
> >
> > ---
> >
> > ### **C5:** "[T]he exogenous randomness must incorporate all the past evolution of society, namely because the current tier depends on them. As such, I believe it is not true that $E_t$ are independent for different $t$"
> >
> > **R5:** We totally agree with you that the past evolution of the society should be incorporated in the dynamic modeling, and we model such influence from the past through (components of) $E_t$. In light of your comment, to avoid possible misunderstandings and make sure the model detail is transparent and informative, we have updated our discussion on the role of exogenous randomness $E_t$ in Appendix B.3 along with the side note `Re: C5 by Reviewer pPYU` on page 18.
> >
> > ---
> >
> > ### **C6:** "$\Delta_{\mathrm{STIR}}$ is not defined before it is characterized. I believe that, at time $t$ the definition is $\mathbb{E} \left[\lvert f_{t + 1}(0, E_{t + 1}) - f_{t + 1}(1, E_{t + 1}) \rvert\right] - \mathbb{E} \left[\lvert f_{t}(0, E_{t}) - f_{t}(1, E_{t}) \rvert\right]$. This should be stated first and foremost and the dependence on $t$ should be explicit."
> >
> > **R6:** Thank for pointing this out and sharing your suggestion, which helped make the notation more informative. Following your suggestion, we have updated the statement and also explicitly incorporated the dependence on time step in $\Delta_{\mathrm{STIR}} \rvert_t^{t + 1}$. For the convenience of locating the related statement, we provide the side note `Re: C6 by Reviewer pPYU` on page 5.
> >
> > ---
> >
> > ### **C7:** "$q_t$ is introduced as a conditional density on the pair of tiers (for each attribute). [... This] density is rather relative to the measure of the exogenous variables $E_t$, and should indeed always be written as a function of the value the latter takes ($\epsilon$) as follows: $q_t \big(f_t(0, \epsilon), f_t(1, \epsilon) \mid d, d', y, y' \big)$."
> >
> > **R7:** Thank you for pointing this out and sharing your suggestion, which help prevent potential misunderstandings. We follow your suggestion and have updated the expressions in our manuscript.
> >
> > ---
> >
> > ### **C8:** "Theorem 4.1’s limitations of why we cannot establish tier balance is due to the fact that the only interventions we have to control continuous random variables is a pair of discrete random variables. Theorem 4.2’s perfect estimator making tier balance worse is due to the fact that the dynamics in Assumption 3.6 are polarizing based on labels only."
> >
> > **R8:** Thank you for carefully considering our results and providing this summary. Please kindly notice that there might be some misunderstandings.
> >
> > The quantitative analysis in Theorem 4.1 and Theorem 4.2 boils down to the behavior of the $\Delta_{\mathrm{STIR}} \rvert_t^{t + 1}$, which involves _three_ components: (1) individual-level gap comparison, (2) conditional joint density $q_t$, and (3) combination of $D_t$, $Y_t$ values. We believe by "a pair of discrete random variables" you are referring to component (3). Please kindly notice that the results rely not only on component (3), but also (1) and (2). This is why beyond negative results in Theorems 4.1 - 4.2, we also have a positive result Theorem 4.3 (we believe this is also related to your comment **C9**, to which we respond in **R9**). For the convenience of locating the related discussion, we provide the side note `Re: C8 by Reviewer pPYU` on page 6.
> >
> > ---
> >
> > (continuing)

---

> > > ### Author Response · Authors · 2022-11-17
> > > **Response to Reviewer pPYU (3 / 3)**
> > >
> > > (continued)
> > >
> > > ---
> > >
> > > ### **C9:** "As for Theorem 4.3, there seems to be an issue in the proof. When simplifying $\alpha_Y = 0$, it seems that the values of $\gamma$ (which determine $q_t$) are also set to something specific. If this is true, then it should also be part of the Theorem’s hypothesis."
> > >
> > > **R9:** Thank you for your careful review. Theorem 4.3 results from quantitative analyses on three different components (as in our response **R8**). The value of $\alpha_Y$ is related to component (1). The value of $\gamma$ is related to component (2). The two values are irrelevant to each other, and all hypotheses needed are included when presenting Theorem 4.3. For the convenience of locating the related material, we provide the side note `Re: C9 by Reviewer pPYU` on page 33.
> > >
> > > ---
> > >
> > > ### **C10:** "While we see the polarization caused by the perfect predictor in Fig.2, it is not clear whether the counterfactually fair one is any better, as the tiers do not appear balanced in Fig. 3(c)"
> > >
> > > **R10:** Thank you for considering the connection between our theoretical and empirical results. We would like to refer to our Figure 3(a), where we show the step-by-step trend of tier updates. We can observe that around step 12, the gaps decrease before increasing again, indicating that counterfactually fair predictors do have the potential (not guarantee) to get closer to the long-term fairness goal. For the convenience of locating the related discussion, we provide the side note `Re: C10 by Reviewer pPYU` on page 9.
> > >
> > > ---
> > >
> > > ### **Q11:** "[For Equation (5)] with $\varphi(t+1)\left(\epsilon, d, d^{\prime}, y, y^{\prime}\right)$ precisely the date given in the tables in the appendix, which depend only on the past $\epsilon$ and the decisions and labels $d, d', y, y'$, why not just do
> > >
> > > > $$\Delta_{\operatorname{STIR}}(t)=\sum_{d, d^{\prime}, y, y^{\prime}} P_t\left(d, d^{\prime}, y, y^{\prime}\right) \int q_t\left(f_t(0, \epsilon), f_t(1, \epsilon) \mid d, d^{\prime}, y, y^{\prime}\right) \cdot\left(\left|\varphi(t+1)\left(\epsilon, d, d^{\prime}, y, y^{\prime}\right)-\varphi_t(\epsilon)\right|\right) \mathrm{d} \epsilon$$
> > >
> > > **R11:** Thank you for considering the expression of STIR term in detail and sharing your suggestion. Following your suggestion, we have updated the expression in Equation (5):
> > > $$\begin{equation*}
> > >     \begin{split}
> > >         & \Delta_{\mathrm{STIR}} \rvert_{t}^{t + 1}
> > >         \coloneqq
> > >         \mathbb{E} \left[\lvert
> > >             f_{t + 1}(0, E_{t + 1}) - f_{t + 1}(1, E_{t + 1})
> > >         \rvert\right]
> > >         -
> > >         \mathbb{E} \left[\lvert
> > >             f_{t}(0, E_{t}) - f_{t}(1, E_{t})
> > >         \rvert\right] \\\\
> > >         & = \sum_{d, d', y, y' \in \{0, 1\}}
> > >         P_t (d, d', y, y')
> > >         \cdot
> > >         \int_{\epsilon \in  \mathbb{E}}
> > >         \int_{\xi \in  \mathbb{E}} q_t \big(f_t(0, \epsilon), f_t(1, \epsilon) \mid d, d', y, y' \big) \\\\
> > >         & ~~~~~~
> > >         \cdot \big( \lvert \varphi_{t + 1}(\xi) \rvert - \lvert \varphi_t(\epsilon) \rvert \big)
> > >         \cdot
> > >         \mathbf{1}\\{ \varphi_{t + 1}(\xi) = G_t(f_t, g_t^D, g_t^{Y^{\text{(ori)}}}; d, d', y, y', \epsilon, \alpha_D, \alpha_Y) \\}
> > >         d \xi d \epsilon.
> > >     \end{split}
> > > \end{equation*}$$
> > >
> > > We are a little bit hesitant to reload the function $\varphi(t + 1)$ since $\varphi_{t + 1}(\xi) \coloneqq f_{t + 1}(0, \xi) - f_{t + 1}(1, \xi)$ appears in the appendix tables. Therefore, we use $G_t$ indicate that the function only relies on information at time step $T = t$ within the indicator function. We keep the indicator function to explicitly state that we are keeping track of the individual. Please kindly let us know if you think our updated expression is informative and clear. We also update the derivation detail of STIR in Appendix B.4 along with the side note `Re: Q11 by Reviewer pPYU` on page 20.
> > >
> > > ---
> > >
> > > ### **Q12:** "In Appendix B.2, you talk about tracking individuals. This is related to the previous point, but I am not sure how you are defining individuals, based on what properties or partitioning of the population?"
> > >
> > > **R12:** Thank you for your great question. We characterize "individual" according to the value of the exogenous randomness $E_t$, which signifies the unique characteristics of an individual. By "keeping track of individuals" we are referring to the indicator function in Equation (5), i.e., the $\mathbf{1}\\{ \cdot \\}$ function we presented in response **R11**. For the convenience of locating the related discussion, we provide the side note `Re: Q12 by Reviewer pPYU` on page 20.

---

### Official Review · Reviewer_N3M4 · 2022-11-02

**Confidence:** 3
**Correctness:** 3
**Technical Novelty And Significance:** 3
**Empirical Novelty And Significance:** 2
**Recommendation:** 6

**Clarity, Quality, Novelty And Reproducibility:**

The paper is quite dense. New notations are introduced throughout which makes it hard to follow. The work is heavily dependant on the DAG in figure 1 which models the causal relationships between the different variables. It would be helpful to clearly explain and justify the assumptions made throughout.

Additionally the motivation for the fairness notion could be more intuitively explained. In definition 2.1, the Tier balancing fairness notion is satisfied when H is independent of A. Even with the running example, it is unclear how this translate specifically as a fairness notion with regards to the decision D. The explicit comparison with existing fairness definitions is left to the appendix but it would be good to add a summary in the main section. It would be helpful to explain the definition through the graph and the fairness problem it addresses.

I have not carefully checked the appendix.

**Strength And Weaknesses:**

Strengths:
- The addresses an important and under-explored problem of long term fairness. The modelling of long term fairness as a dynamical systems via a causal DAG seems novel and can potentially lead to further interesting work in this direction.
- Despite the theoretical nature of the paper, the authors make a good effort in using a running example of credit application to illustrate the approach.

Weaknesses:
- Figure 1: the paper relies on the DAG in figure 1 which shows the causal relations between the variables of interest at each time step: H (hidden socio economic status), D (decision), A (sensitive attribute), Y^ori ("groundtruth outcome"), Y^obs (observed outcome), and X (observed features). However, it's not clearly explained what justifies the assumed causal relationship between these variables. For example, why is H_t+1 only updated from ((H_t, Y^(ori)_t, D_t)). If H is socio-economic status, wouldn't it also be dependant on some observed features such as income? Additionally, in paragraph "Decision-making dynamics", it says that "The institution (decision maker) assigns decision D_t to each individual according to the observed features (A_t, X_t,i)  and the outcome record Y^obs". However this is not reflected in the graph. There is no arrow between Y^obs_t and D. The causal relationship between H_t, A_t and X_t are also not clarified. Overall, the assumptions made in section 2.1 and the graph are not clearly justified.
- Y^ori vs Y^obs: Y^ori is defined as the "(unobserved) ground truth label" and Y^obs label. In the graph, Y^ori is a direct cause of Y^obs. I am not sure how to interpret Y^ori. If Y^ori is the potential outcome, then what does the causal arrow between Y^ori and Y_obs mean?
- Fact 3.2: it is unclear to me why why the structural equations for D_t, Y_^ori_t and H_t are only dependent on A and a noise term when this is not reflected in the graph. The authors mention that Fact3.2 is a direct result of representing causal relations with a functional causal model but it would be good to explain more how the structural equations relate to the graph.

**Summary Of The Paper:**


The paper aims to address long term fairness in decision making systems by using a dynamical approach to model the underlying changes over time. The paper proposes a causal DAG that models the connection between the data (background variable H, sensitive attribute A, observed features X) and a decision D at each time step. Given an unobserved variable H_t (for example: socio-economic status) which is a direct cause of the potential outcome Y^ori and X, the goal is to make H_ and A_t independent. A main contribution of the paper is showing that a one step intervention is insufficient to achieve this goal and you need multiple interventions over time and specific conditions to achieve the long-term fairness goal.



**Summary Of The Review:**

I have updated my score to 6 after the rebuttal below. The authors have addressed my concerns.

-----------------

Original summary: This is an interesting paper which addresses an important problem in long term fairness. While the proposal to model the problem as a dynamical problem with a causal DAG and show impossibility results is interesting and useful, the clarity of the paper could be improved. Given the reliance on the DAG, it is hard to evaluate the contribution without a clear justification of the correctness of the graph.

---

> ### Author Response · Authors · 2022-11-17
> **Response to Reviewer N3M4 (1 / 2)**
>
> We are very grateful for your thoughtful, constructive, and insightful comments and suggestions, as well as the time devoted!
> In our revised manuscript, we (1) **condensed and updated notations** to make sure they are informative, rigorous, and concise,
> (2) added **additional discussions related to motivation,  modeling details, and related literature** and reorganize the materials according to reviewer's suggestions thoroughly (main text and appendix),
> and (3) added **additional experimental results** and further improve the connection between our theoretical and empirical results.
> We also included **the list of side notes on page 13** to help locate the related contents.
> Below please see our responses to specific points in the review comments:
>
> ---
>
> ### **C1:** "Why is $H_{t+1}$ only updated from $(H_t, Y^{\mathrm{(ori)}}_t, D_t)$. If $H$ is socio-economic status, wouldn't it also be dependent on some observed features such as income?"
>
> **R1:** Thank you for considering the detail of our model. You are totally right. The latent causal factor $H_{t + 1}$ is dependent with $X_{t + 1, i}$. However, if we follow the causal direction, $H_{t + 1}$ can be determined by $H_t$, $Y_t^{\mathrm{(ori)}}$, and $D_t$.
> We model the update of $H_{t + 1}$ only from its direct causes, instead of from all variables that are dependent with $H_{t + 1}$. In light of your comment, we have added a quick narrative example along with the side note `Re: C1 by Reviewer N3M4` on page 2.
>
> ---
>
> ### **C2:** "Institution assigns decision $D_t$ to each individual according to the observed features $(A_t, X_{t,i})$ and the outcome record $Y_t^{(\mathrm{obs})}$". However, there is no arrow between $Y_t^{(\mathrm{obs})}$ and $D_t$."
>
> **R2:** Thank you for your careful review. You are totally right. $D_t$ is trained on the joint distribution $(A_t, X_{t,i}, Y_t^{(\mathrm{obs})})$. However, when making predictions, $D_t$ _only_ takes $(A_t, X_{t,i})$ as input. Since we are modeling causal relations in data generating processes, we only add an arrow if there is a direct causal relation between variables. Therefore, the data generating process of $D_t$ does **not** involve an arrow between $Y_t^{(\mathrm{obs})}$ and $D_t$. In light of your comment, we have included this discussion in Appendix B.1.1 along with the side note `Re: C2 by Reviewer N3M4` on page 16.
>
> ---
>
> ### **C3:** "The causal relationship between $H_t$, $A_t$ and $X_t$ are also not clarified."
>
> **R3:** Thank you for your comment. In Section 2.1, we present the relationship among variables $\big(H_t, A_t, X_{t,i}, Y_t^{(\mathrm{ori})}, Y_t^{(\mathrm{obs})}\big)$ in the paragraph "Underlying data dynamics". In light of your comment, we add additional modeling details of the involved dynamics and the role of $X_{t,i}$ in Appendix B.1.1 along with the side note `Re: C3 by Reviewer N3M4` on page 16.
>
> ---
>
> ### **C4:** "In the graph, $Y_t^{(\mathrm{ori})}$ is a direct cause of $Y_t^{(\mathrm{obs})}$. I am not sure how to interpret $Y_t^{(\mathrm{ori})}$. If $Y_t^{(\mathrm{ori})}$ is the potential outcome, then what does the causal arrow between $Y_t^{(\mathrm{ori})}$ and $Y_t^{(\mathrm{obs})}$ mean?"
>
> **R4:** Thank you for your insightful question. We interpret $Y_t^{(\mathrm{ori})}$ as a potential outcome, namely, whether or not one would repay the loan were he/she approved the credit at $T = t - 1$ (Section 2.1). There is an arrow between $Y_t^{(\mathrm{ori})}$ and $Y_t^{(\mathrm{obs})}$ because $Y_t^{(\mathrm{obs})}$ is the observed copy of $Y_t^{(\mathrm{ori})}$ controlled by $D_{t - 1}$ (1 for observable and 0 for undefined $Y_t^{(\mathrm{obs})}$). The mathematical expression is summarized in Equation 2. For the convenience of locating related discussion, we provide the side note `Re: C4 by Reviewer N3M4` on page 4.
>
> ---
>
> (continuing)

---

> > ### Author Response · Authors · 2022-11-17
> > **Response to Reviewer N3M4 (2 / 2)**
> >
> > (continued)
> >
> > ---
> >
> > ### **C5:** "Fact 3.2: it is unclear to me why why the structural equations for $D_t$, $Y_t^{(\mathrm{ori})}$ and $H_t$ are only dependent on A and a noise term when this is not reflected in the graph. The authors mention that Fact 3.2 is a direct result of representing causal relations with a functional causal model but it would be good to explain more how the structural equations relate to the graph."
> >
> > **R5:** Thanks for your suggestion. Following your suggestion, we have provided a remark for Fact 3.2 in Appendix B.3 along with a pointer in the footnote. In particular, we present (1) the definition of the functional causal model (FCM), (2) the DAG with exogenous terms explicitly modeled in Figure 4, and (3) details with respect to the application of FCM in Fact 3.2. For the convenience of locating related discussion, we provide the side note `Re: C5 by Reviewer N3M4` on page 18.
> >
> > ---
> >
> > ### **C6:** "It is unclear how [_Tier Balancing_ fairness notion] translates specifically as a fairness notion with regards to the decision $D$."
> >
> > **R6:** Although our _Tier Balancing_ notion is not directly defined in terms of the decisions $D_T$ themselves, the induced consequence in the future $H_{T+K}$ inevitably involves the interplay of _both_ decision-making _and_ the underlying data dynamics along the way. We believe such interplay plays an important and indispensable role in the pursuit of long-term fairness. In light of your comment, we have provided such motivation immediately after the fairness definition, along with the side note `Re: C6 by Reviewer N3M4` on page 3.
> >
> > ---
> >
> > ### **C7:** "The explicit comparison with existing fairness definitions is left to the appendix but it would be good to add a summary in the main section."
> >
> > **R7:** Thank you for your constructive suggestion. We totally agree with you, and that is why we summarize the distinctions between our fairness notion previous notions immediately after introducing our fairness notion in Section 2.2. Due to space limit, we defer more detailed discussions in Appendix A (as you kindly pointed out). For the convenience of locating the related content, we provide the side note `Re: C7 by Reviewer N3M4` on page 3.

---

### Decision · Program_Chairs · 2023-01-20

**Decision:**

Accept: poster

**Justification For Why Not Higher Score:**

The paper does not provide a strong solution to the problem being studied. The modeling assumptions are quite simplistic, the presented analysis is only one step, and the proofs of the main results do not seem to rely on novel or interesting proof techniques. The “impossibility result” from one-step analysis is somewhat expected. The empirical results are also limited, and at least in the original version did not support theoretical analysis.


**Justification For Why Not Lower Score:**

The reviewers agreed that the problem being studied is interesting. Having this paper published may inspire people to build and improve upon the current causal model of how the underlying factors evolve over time and affect fairness.

**Metareview: Summary, Strengths And Weaknesses:**

Thanks again to the authors and reviewers for actively participating in the discussions that will help improve this paper.

The paper considers a problem of long term fairness under dynamic conditions, which the reviewers thought would be of interest to the community. The authors assume a particular model of how the so-called tiers (causal factors affecting the outputs and some features) evolve over time. They then perform a one-step analysis of how decisions affect the tiers, and derive some positive and negative results. The positive results towards achieving long term fairness consider all possible conditions which live in a discrete set due to the modeling assumptions. The proof thus involves systematically going through these conditions.

Mathematical presentation could be substantially improved, bringing in clarity and trust in the derived results. Nearly all of the reviewers had trouble extracting the key messages from the paper: the paper is dense, and hard to follow. The authors could potentially improve the presentation by distilling the key ideas, and perhaps keeping only the necessary parts of the results/proof outlines in the main text. Improvements in the notation, as suggested by some reviewers, would also go a long way.

**Note From Pc:**

if the above contains the word "oral" or "spotlight" please see: "oral" presentation means -> notable-top-5% and "spotlight" means -> notable-top-25%. As stated in our emails, we are disassociating presentation type from AC recommendations

**Summary Of Ac-Reviewer Meeting:**

All of the reviewers engaged in further discussion, and most participated in a virtual call to discuss the paper further. Only WEUm did not participate in the call, but responded to my emails and provided a justification for why they were unavailable.

During the meeting, all of the reviewers shared the following concerns:

(1) the modeling assumptions and independence of the protective attribute. I would encourage you to add a paragraph discussing this as a potential limitation, and perhaps give an explicit example showing that such a model is still interesting (as you did in the rebuttal).
(2) independence assumption at K=0 (just before Section 3, last paragraph). Is that really an assumption? If so, please explain and justify why it is reasonable. Reviewers are quite concerned about this, but also interpret this paragraph differently indicating confusion.
(3) mathematical clarity. This one is harder to address. But, for example, your proof in the appendix, page 34 of the main positive result is really hard to follow, making the reader worry that there may be some errors introduced. In general, the mathematical presentation of the results could be really improved.

After the meeting, all the reviewers who participated updated their reviews reiterating their concerns. The authors responded to the points above and promised to implement various changes improving the clarity. They confirmed that (2) is not an assumption and will reword the paragraph to make it less confusing. Regarding (3), the authors say that they validated the proof using numerical simulations.